# Learning Survival Distributions with Individually Calibrated Asymmetric Laplace Distribution

**Deming Sheng, Ricardo Henao***
{deming.sheng,ricardo.henao}@duke.edu

## Abstract

Survival analysis plays a critical role in modeling time-to-event outcomes across various domains. Although recent advances have focused on improving *predictive accuracy* and *concordance*, fine-grained *calibration* remains comparatively underexplored. In this paper, we propose a survival modeling framework based on the Individually Calibrated Asymmetric Laplace Distribution (ICALD), which unifies *parametric* and *nonparametric* approaches based on the ALD. We begin by revisiting the probabilistic foundation of the widely used *pinball* loss in *quantile regression* and its reparameterization as the *asymmetry form* of the ALD. This reparameterization enables a principled shift to *parametric* modeling while preserving the flexibility of *nonparametric* methods. Furthermore, we show theoretically that ICALD, with the *quantile regression* loss is probably approximately individually calibrated. Then we design an extended ICALD framework that supports both *pre-calibration* and *post-calibration* strategies. Extensive experiments on 14 synthetic and 7 real-world datasets demonstrate that our method achieves competitive performance in terms of *predictive accuracy*, *concordance*, and *calibration*, while outperforming 12 existing baselines including recent *pre-calibration* and *post-calibration* methods.

## 1 Introduction

Survival analysis, modeling the distribution of time-to-event outcomes, has gained popularity in recent years (Lánczky & Győrffy, 2021; Emmerson & Brown, 2021). Survival models can be broadly categorized into *parametric*, *semi-parametric* and *nonparametric* approaches, depending on the strength of the distributional assumptions they make about the underlying time-to-event distribution. *Parametric models* make strong assumptions by specifying a particular probability distribution for event times, such as the exponential (Feigl & Zelen, 1965), Weibull (Scholz & Works, 1996), log-normal (Royston, 2001), asymmetric Laplace (Kotz et al., 2012), or their mixtures (Nagpal et al., 2021). In contrast, *semi-parametric model*, such as the Cox proportional hazards (CPH) model (Cox, 1972), relax these assumptions by modeling the hazard multiplicatively without specifying the baseline hazard. *Nonparametric models*, such as Gradient Boosting Machines (GBM) (Dembek et al., 2014) and Random Survival Forests (RSF) (Ishwaran et al., 2008), do not rely on predefined forms for the survival distribution or the hazard function, and instead estimate survival quantities directly from the data. Alternatively, neural networks have significantly advanced survival modeling in both *(semi-)parametric* and *nonparametric* paradigms. For example, the *parametric* LogNorm-MLE model (Hoseini et al., 2017) improves parameter estimation for survival distributions under the log-normal assumption (Royston, 2001). The *semi-parametric* DeepSurv model (Katzman et al., 2018) leverages neural networks to capture complex nonlinear covariate effects while preserving the multiplicative structure of Cox models (Cox, 1972). *Nonparametric* models such as DeepHit (Lee et al., 2018) and CQRNN (Pearce et al., 2022) utilize neural architectures to directly estimate individualized survival distributions, offering greater expressiveness and scalability than *non-neural* models like GBM (Dembek et al., 2014) and RSF (Ishwaran et al., 2008).

---

*Corresponding author, Duke University.

To assess the performance of survival models, evaluation metrics are typically grouped into three major categories: *predictive accuracy* (Graf et al., 1999), *concordance* (Harrell et al., 1982; Uno et al., 2011), and *calibration* (Haider et al., 2020). *Predictive accuracy* measures how well the estimated survival probabilities or times align with the observed outcomes, and is particularly useful in scenarios where precise event time estimates are critical, such as predicting patient prognosis, forecasting treatment duration, or scheduling follow-up assessments. *Concordance* assesses the model's ability to correctly rank individuals by risk, making it valuable for pairwise comparisons, such as prioritizing patients for treatment. *Calibration* reflects the reliability of the predicted survival probabilities, *i.e.*, if predicted risks are consistent with empirical observations. More specifically, *calibration* can be further assessed at the *average*, *group*, or *individual* level (Gneiting et al., 2007). *Individual calibration* (Villani, 2009) is important for high-stakes decisions at the patient level, such as eligibility for high-risk interventions, while *average* and *group calibration* (Naeini et al., 2015) are more relevant for population-level decisions, such as allocating clinical resources or designing screening strategies.

While most recent works (Katzman et al., 2018; Lee et al., 2018; Pearce et al., 2022) primarily focus on *predictive accuracy*, *concordance*, and coarse-grained notions of *calibration*, our objective is to conduct a more comprehensive evaluation of model performance, with particular emphasis on fine-grained *calibration* (Gneiting et al., 2007; Villani, 2009; Naeini et al., 2015). In addition, we propose the *Individually Calibrated Asymmetric Laplace Distribution* (ICALD) model, which significantly improves the calibration performance. Our contributions are summarized below.

- We propose the ICALD model that synthesizes the complementary advantages of *parametric* and *nonparametric* ALD-based approaches. Our model not only enhances *calibration* and mitigates issues associated with *distribution mismatch* in *parametric* ALD approach (Sheng & Henao, 2025), but also effectively addresses the issues of *discretization* and *quantile crossing* commonly encountered in *nonparametric* methods (Pearce et al., 2022).

- Specifically, ICALD admits two theoretically equivalent loss functions, each of which is provably capable of rendering the model *Probably Approximately Individually Calibrated* (*PAIC*; Zhao et al. 2020). More importantly, the model supports both *pre-calibration* and *post-calibration* with either loss, providing a unified and flexible framework where the calibration strategy and loss function can be independently selected based on the specific application requirements.

- We comprehensively evaluate our method on 14 synthetic and 7 real-world datasets using 7 performance metrics that span *predictive accuracy*, *concordance* and *calibration*. Our method is compared against 9 strong baselines covering a wide spectrum of survival models, including both *(semi-)parametric* and *nonparametric* approaches, as well as both *neural* and *non-neural* architectures. Furthermore, we compare with 1 *pre-calibration* method (X-CAL; Goldstein et al. 2020) and 2 *post-calibration* methods (CSD; Qi et al. 2024a and CiPOT; Qi et al. 2024b) that target *average calibration* of survival distributions. Overall, our method consistently outperforms these baselines, achieving superior performance in terms of *predictive*, *concordance* and *calibration*.

## 2 BACKGROUND

We use capital letters $X$, $Y$, $Q$ to denote random variables, lowercase letters $y$, $q$ to denote fixed values, bold lowercase letters $\mathbf{x}$ to denote vectors, and $\mathcal{X}$, $\mathcal{Y}$, $\mathcal{Q}$ to denote the sets of all possible values they can take.

**Survival Data** A survival dataset $\mathcal{D}$ is composed of a set of triplets $\{(\mathbf{x}_n, y_n, \delta_n)\}_{n=1}^N$, each containing covariates $\mathbf{x}_n \in \mathbb{R}^d$, an observed time $y_n \in \mathbb{R}_+$, and an event indicator $\delta_n \in \{0, 1\}$. Moreover, the observed time is defined as the minimum between the true event time $e_n$ and the censoring time $c_n$, *i.e.*, $y_n = \min(e_n, c_n)$, and the event indicator is defined as $\delta_n = \mathbb{I}(e_n \leq c_n)$, denoting whether the event was *observed* ($\delta_n = 1$) or *censored* ($\delta_n = 0$). In this work, we adopt the common assumption that the event and censoring distributions are conditionally independent given the covariates, *i.e.*, $e \perp\!\!\!\perp c \mid \mathbf{x}$. Moreover, although we focus on right-censored data; less common types of censoring (*e.g.*, left and interval) can also be readily accommodated (Klein & Moeschberger, 2006).

**Asymmetric Laplace Distribution** The Asymmetric Laplace Distribution (ALD) (Kotz et al., 2012) has two common parameterizations. In its *quantile form*, let the random variable $Y \sim \mathcal{AL}(\theta, \sigma, q)$, where $\theta \in \mathbb{R}$ is the location (distribution mode), $\sigma > 0$ is the scale, and $q \in (0, 1)$ is the target quantile. This form is quite useful in quantile regression (Koenker & Bassett Jr, 1978), and has the following probability density function (PDF):

$$f(y; \theta, \sigma, q) = \frac{q(1-q)}{\sigma} \begin{cases} \exp\left(\frac{q}{\sigma}(\theta - y)\right), & y \geq \theta, \\ \exp\left(\frac{1-q}{\sigma}(y - \theta)\right), & y < \theta. \end{cases} \tag{1}$$

Alternatively, in its *asymmetry form*, the ALD is reparameterized as $\mathcal{AL}(\theta, \sigma, \kappa)$, where $\kappa > 0$ denotes the asymmetry parameter, and the latter is related to $q$ through $q = \frac{\kappa^2}{1+\kappa^2}$. Its PDF is:

$$f(y; \theta, \sigma, \kappa) = \frac{\sqrt{2}}{\sigma} \frac{\kappa}{1 + \kappa^2} \begin{cases} \exp\left(\frac{\sqrt{2}\kappa}{\sigma}(\theta - y)\right), & y \geq \theta, \\ \exp\left(\frac{\sqrt{2}}{\sigma\kappa}(y - \theta)\right), & y < \theta. \end{cases} \tag{2}$$

**Quantile Value and Percentage** Given a random variable $Y$ with its conditional cumulative distribution function (CDF) $F_Y(y|\mathbf{x})$, the *quantile value* $y_q$ corresponding to a *quantile percentage* $q \in [0, 1]$ is defined as (Garthwaite et al., 2002):

$$y_q = F_Y^{-1}(q|\mathbf{x}) = \inf\{y \in \mathbb{R} : F_Y(y|\mathbf{x}) \geq q\}, \tag{3}$$

where $y_q$ represents the threshold below which a proportion $q$ of observations $y \in \mathcal{Y}$ lie. The quantile percentage $q$ indicates the probability mass accumulated up to $y_q$, *i.e.*, the fraction of the distribution that falls below this threshold.

**Average Calibration** A CDF predictive model $F_\Phi(y|\mathbf{x})$ parameterized by $\Phi$ is said to be perfectly *averagely calibrated*, if its predicted distribution aligns with the empirical distribution of the target population (represented by a test set). Formally, the model should satisfy (Gneiting et al., 2007):

$$\Pr\left(y \leq F_\Phi^{-1}(q|\mathbf{x})|\mathbf{x} \in \mathcal{X}\right) = q \quad \text{or} \quad \Pr\left(F_\Phi(y|\mathbf{x}) \leq q|\mathbf{x} \in \mathcal{X}\right) = q, \quad \forall q \in [0, 1], \forall y \in \mathcal{Y}. \tag{4}$$

**Group Calibration** A CDF predictive model $F_\Phi(y|\mathbf{x})$ parameterized by $\Phi$ is said to be perfectly *group calibrated* with respect to a collection $\mathcal{S} = \{\mathcal{S}_k\}_{k=1}^K \subset \mathcal{X}$ *predefined* subsets if, for every group $\mathcal{S}_k$, the predicted distribution is consistent with the empirical outcome distribution within that group. Formally, the model should satisfy (Gneiting et al., 2007):

$$\Pr\left(y \leq F_\Phi^{-1}(q|\mathbf{x})|\mathbf{x} \in \mathcal{S}_k\right) = q \text{ or } \Pr\left(F_\Phi(y|\mathbf{x}) \leq q|\mathbf{x} \in \mathcal{S}_k\right) = q, \ \forall q \in [0, 1], \forall y \in \mathcal{Y}, \forall k \in \mathcal{S}. \tag{5}$$

**Individual Calibration** A CDF predictive model parameterized by $\Phi$ is said to be perfectly *individually calibrated* if its predicted conditional cumulative distribution function $F_\Phi(Y|\mathbf{x})$ satisfies, for any given input $\mathbf{x} \in \mathcal{X}$, the following condition (Gneiting et al., 2007):

$$\Pr\left(Y \leq F_\Phi^{-1}(q|\mathbf{x}) \mid X = \mathbf{x}\right) = q \text{ or } \Pr\left(F_\Phi(Y|\mathbf{x}) \leq q|X = \mathbf{x}\right) = q, \ \forall q \in [0, 1], \forall Y \in \mathcal{Y}. \tag{6}$$

**Definition 1** (*Probably Approximately Individually Calibrated (PAIC)* (Zhao et al., 2020))**.** *A model with predictive CDF $F_\Phi(Y|\mathbf{x})$ is said to be $(\epsilon, \delta)$-PAIC if for all $\mathbf{x} \in \mathcal{X}$, $Y \in \mathcal{Y}$, and $q \in [0, 1]$, the following holds:*

$$\Pr\left[\int_0^1 \left|\Pr\left[Y \leq F_\Phi^{-1}(q|\mathbf{x})\right] - q\right| dq \leq \epsilon\right] \text{ or } \Pr\left[\int_0^1 \left|\Pr\left[F_\Phi(Y|\mathbf{x}) \leq q\right] - q\right| dq \leq \epsilon\right] \geq 1 - \delta.$$

We slightly extended the original definition for *PAIC* introduced by Zhao et al. (2020) to also allow an equivalent expression based on the inverse CDF function. Definition 1 is connected to earlier notions of calibration for regression models, including those in Gneiting et al. (2007) and Kuleshov et al. (2018), which formalize approximate individual calibration consistent with Eq. (6).

Note that $F_\Phi(y|\mathbf{x})$ represents the model that directly outputs the conditional CDF value at $y \in \mathcal{Y}$. However, popular models such as CQRNN (Pearce et al., 2022) based on Eq. (1) produce $\tilde{y}_q = F_\Phi^{-1}(q|\mathbf{x})$ for specific values of $q$, and DeepHit (Lee et al., 2018) produces $F_\Phi(y|\mathbf{x})$, but assuming that it is piecewise constant. In contrast, *parametric* models such as the accelerated failure time model (AFT) (Wei, 1992) return distributional parameters which then can be used to obtain $F_\Phi(y|\mathbf{x})$ for $y \in \mathcal{Y}$. Specifically, a *parametric* model for $F_\Phi(y|\mathbf{x})$ based on the ALD (Sheng & Henao, 2025) in Eq. (2) is denoted here as $\{\theta, \sigma, \kappa\} = m_\Phi(\cdot)$.

# 3 Survival Modeling with the ICALD

## 3.1 Parametric and Nonparametric ALD Approaches

Quantile regression methods, such as CQRNN (Pearce et al., 2022), utilize the widely adopted *pinball* (or *check*) loss (Koenker & Bassett Jr, 1978) to estimate conditional quantiles of the response variable. The *pinball* loss for a target value $y$ and a predicted quantile value $\tilde{y}_q = F_\Phi^{-1}(q|\mathbf{x}) = m_{\Phi,q}(\mathbf{x})$, from a model for $q$ is defined as:

$$\mathcal{L}_{\text{pinball}}(y; \Phi, q) = (y - m_{\Phi,q}(\mathbf{x}))(q - \mathbb{I}[m_{\Phi,q}(\mathbf{x}) > y]), \tag{7}$$

which optimizes a weighted absolute deviation objective that asymmetrically penalizes the under- and over-estimation of $y$. This formulation yields a predictive function that statistically separates the $q$-th and $(1 - q)$-th quantiles in a consistent manner. Moreover, the loss for CQRNN accounting for censoring leveraging the Portnoy estimator (Portnoy, 2003) is defined as:

$$\mathcal{L}_{\text{Cqr}}(y; \Phi, q) = \sum_{n \in \mathcal{D}_{\text{O}}} \mathcal{L}_{\text{pinball}}(y; \Phi, q) + \sum_{n \in \mathcal{D}_{\text{C}}} w_n \mathcal{L}_{\text{pinball}}(y; \Phi, q) + (1 - w_n)\mathcal{L}_{\text{pinball}}(y^*; \Phi, q), \tag{8}$$

where $\mathcal{D}_{\text{O}}$ and $\mathcal{D}_{\text{C}}$ are the subsets of the dataset $\mathcal{D} = \mathcal{D}_{\text{O}} \cup \mathcal{D}_{\text{C}}$, corresponding to observed ($\delta = 1$), censored ($\delta = 0$) instances, $y^*$ is a *pseudo* target set to be sufficiently larger than all observed values of $y$ in the dataset, $w_n \in (0, 1)$ is a weight that balances the contribution of the censored and imputed targets and the loss is optimized for a model $m_{\Phi,q}(\mathbf{x})$ defined for a specific value of $q$. In fact, the pinball loss in Eq. (7) can be interpreted as the negative log-likelihood of the *quantile form* of the ALD, parameterized as $\mathcal{AL}(\theta = \tilde{y}_q, \sigma = 1, q)$, up to an additive constant (see Lemma 1 in Appendix A.1 for technical details). This connection provides a probabilistic interpretation of quantile regression and forms the basis for likelihood-based extensions. Building on this foundation, it is possible to extend the *nonparametric* quantile regression into a *parametric* modeling framework by adopting an alternative parameterization of the *asymmetry form* of the ALD, which explicitly models the location, scale, and asymmetry parameters of $\mathcal{AL}(\theta, \sigma, \kappa)$ (Sheng & Henao, 2025). This reformulation enables the transition from pointwise quantile estimation to full conditional distribution estimation, offering a relatively more flexible modeling framework beyond fixed quantile levels. Formally, the loss of the *parametric* model can be summarized as follows:

$$\mathcal{L}_{\text{ALD}}(y; \Phi) = - \sum_{n \in \mathcal{D}_{\text{O}}} \log f_{\text{ALD}}(y_n; m_\Phi(\mathbf{x})) - \sum_{n \in \mathcal{D}_{\text{C}}} \log S_{\text{ALD}}(y_n; m_\Phi(\mathbf{x})), \tag{9}$$

where $\{\theta, \sigma, \kappa\} = m_\Phi(\cdot)$ is a *parametric* model that maps covariates to the parameters of the ALD, and $f_{\text{ALD}}(\cdot)$ and $S_{\text{ALD}}(\cdot)$ denote the PDF and survival function $(1 - F_{\text{ALD}}(\cdot))$ of $\mathcal{AL}(\theta, \sigma, \kappa)$, respectively (see Lemma 2 in Appendix A.2 for technical details).

It should be noted that both the *nonparametric* and *parametric* ALD-based modeling approaches in Eq. (8) and Eq. (9), respectively, come with their own limitations. *Nonparametric* approaches are inherently restrictive, as each model (or head) can only capture a single quantile value $y_q$ specified by the quantile percentage $q$. This form of *discretization*, observed in many other methods, can cause approximation errors when quantile grids are sparse. Although increasing the number of quantiles reduces this error, it requires training multiple models, leading to a fragmented formulation that behaves as a collection of independent ALD models. While this increases the density of estimated quantiles, it also introduces substantial computational overhead and fails to capture global coherence across quantile levels, which in turn gives rise to the *crossing quantiles* issue where higher quantile estimates may fall below lower ones, violating $\tilde{y}_{q_1} \geq \tilde{y}_{q_2}$ for all $q_1 > q_2$ (Bondell et al., 2010). A case study illustrating this behavior can be found in Appendix C.4.

Alternatively, *parametric* approaches based on the ALD offer greater flexibility in computing various distributional summaries, such as mean, median, mode and quantiles, which result from having closed-form expressions for $f_{\text{ALD}}(\cdot)$ and $S_{\text{ALD}}(\cdot)$. It is also computationally efficient and maintains smoothness throughout the estimated distribution, thereby avoiding issues like *discretization* and *crossing quantiles* that commonly arise in *nonparametric* approaches. However, relying on a single ALD to model the entire conditional distribution is also restrictive. Although parametric ALD models typically perform well for central quantiles, their approximation error tends to increase at the distribution tails, leading to degraded performance for extreme-quantile estimation. More critically, *distribution mismatch* can occur in some cases, where the estimated ALD distribution significantly deviates from the ground truth, often manifesting as consistent over- or under-estimation in regions of $\mathcal{Y}$. A case study illustrating this behavior can be found in Appendix C.4.

## 3.2 SURVIVAL MODELING WITH THE INDIVIDUALLY CALIBRATED ALD (ICALD)

Given the strengths and limitations of both *non-parametric* and *parametric* ALD-based survival models, combining them within a unified framework seems like a natural and effective strategy. Specifically, we begin by adopting a *parametric* ALD-based survival model as the backbone, which ensures global continuity and smoothness throughout the estimated distribution. Then, we include an *adapter module* in the backbone which takes $q$ as input to produce refined ALD parameters through $\{\theta, \sigma, \kappa\} = m_\Phi(\mathbf{x}, q)$. The resulting model illus-

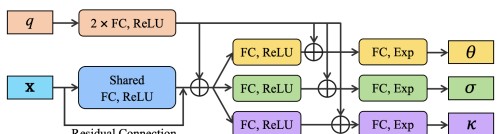

Figure 1: Architecture of the proposed ICALD survival model. Here, $\oplus$ denotes the concatenation operation, and FC refers to a fully connected layer.

trated in Fig. 1 is optimized using the negative log-likelihood loss in Eq. (9) and the quantile regression loss (with censoring) in Eq. (8) for a specific value of $q$ used both as input to the model and the quantile regression loss as follows:

$$\mathcal{L}_{\text{ALD+Cqr}}(y; \Phi) = \mathcal{L}_{\text{ALD}}(y; \Phi) + \lambda \mathcal{L}_{\text{Cqr}}(y; \Phi, q), \tag{10}$$

where $\lambda$ is a hyperparameter that balances the contributions of the two losses (see Appendix C.4 for ablation studies). In practice, to encourage *individualized quantile calibration*, we sample a quantile percentage at random from $q \sim \mathcal{U}(0, 1)$ for each instance $\mathbf{x}$ during training, which in turn is used to refine the prediction of model $m_\Phi(\mathbf{x}, q)$ to minimize $\mathcal{L}_{\text{Cqr}}(y; \Phi, q)$ in Eq. (10) for a specific value of $q$, while also maximizing the likelihood of $y$ over the full ALD distribution implied by the model $\{\theta, \sigma, \kappa\} = m_\Phi(\mathbf{x}, q)$ using $\mathcal{L}_{\text{ALD}}(y; \Phi)$.

Conceptually, this model can be seen as a continuous mixture of ALDs specified as $\int dq\, p(q) f_{\text{ALD}}(y; m_\Phi(\mathbf{x}, q))$, for $p(q) = \mathcal{U}(0, 1)$, in which the backbone captures the general shape of the conditional distribution, while the adapter module drives local adjustments to improve the calibration of the model. Note that marginalizing over $q$ does not yield an ALD distribution and that, in practice, we approximate this mixture at test time by averaging predictions over a finite set of 2,000 samples drawn from $p(q) = \mathcal{U}(0, 1)$. During training, we iterate for up to 2000 epochs with early stopping, which is necessary to prevent overfitting, particularly on datasets with high variance or limited sample size (see Appendix C.4). The properties of the continuous mixture of ALDs are discussed in Appendix A.3. In particular, the CDF of the mixed ALDs admits a closed-form expression, allowing for interpretation via standard scalar summaries (*e.g.*, the mean survival time of the mixed ALDs). Thus, off-the-shelf explanation tools such as SHAP (Lundberg & Lee, 2017) can be applied to these scalar quantities to obtain factor-level attributions for individual predictions; the same analysis can also be aggregated across subgroups to provide cohort-level insights (*e.g.*, how covariate contributions vary across risk strata or time horizons, see a representative visualization and case study in Appendix C.4). In the end, the theoretical foundation supporting the individual calibration ability of the proposed ICALD model is discussed below.

## 3.3 THE ICALD MODEL IS PAIC

By Definition 1, a *quantile regression* model $\tilde{y}_q = F_\Phi^{-1}(q|\mathbf{x}) = m_{\Phi,q}(\mathbf{x})$ trained to minimize the loss $\mathcal{L}_{\text{Cqr}}(y; \Phi, q)$ in Eq. (8), is in principle $(\epsilon, \delta)$-*PAIC*, provided it satisfies the *individual calibration* condition for each outcome $y \in \mathcal{Y}$ and its corresponding covariate $\mathbf{x} \in \mathcal{X}$. However, as also noted in Definition 1, verifying this condition empirically is challenging provided that for each (input) $\mathbf{x}$, the model produces predictions $\tilde{y}_q$ for a fixes set of $q$. This limitation makes it difficult to flexibly estimate the probability $\Pr\left[y \leq F_\Phi^{-1}(q|\mathbf{x})\right]$ across all quantile percentages $q \in [0, 1]$. To overcome this issue, we instead consider evaluating the modified probability $\Pr\left[y \leq F_\Phi^{-1}(q|\mathbf{x}, q)\right]$, where the model $m_\Phi(\mathbf{x}, q)$ trained via Eq. (10) takes as inputs both the covariates $\mathbf{x}$ and quantile percentage $q \sim \mathcal{U}(0, 1)$. This change allows us to assess calibration across quantile percentages using each observed $y$ with $m_\Phi(\mathbf{x}, q)$, under the assumption that it is monotonic in $q$. Note that any continuous but non-monotonic function can be transformed into a monotonic one by sorting its outputs over quantile levels (see Appendix A.5 for more details on monotonicity). This idea draws inspiration from the *reparameterization trick* introduced by Kingma et al. (2013). Under this construction, we extend the notion of *PAIC* to the monotonic setting, resulting in the following definition.

**Definition 2** (*Monotonic Probably Approximately Individually Calibrated (MPAIC)* (Zhao et al., 2020)). *A predictive CDF model $F_\Phi(Y|X, q)$ is said to be $(\epsilon, \delta)$-MPAIC if for all $X \in \mathcal{X}$, $Y \in \mathcal{Y}$, and $q \in [0, 1]$, the following holds:*

$$\Pr\left[\int_0^1 \left|\Pr\left[Y \leq F_\Phi^{-1}(q|X, q)\right] - q\right| dq \leq \epsilon\right] \text{ or } \Pr\left[\int_0^1 \left|\Pr\left[F_\Phi(Y|X, q) \leq q\right] - q\right| dq \leq \epsilon\right] \geq 1 - \delta.$$

Thus, by Definition 2, we can conclude that the ICALD model $m_\Phi(\mathbf{x}, q)$, trained with the loss $\mathcal{L}_{\text{ALD+Cqr}}(y; \Phi)$ in Eq. (10), is $(\epsilon, \delta)$-*MPAIC*. Moreover, with Theorem 1 below (proof is provided in Appendix A.4), this further implies that the ICALD model is also *PAIC*.

**Theorem 1** (*MPAIC is a sufficient (but not necessary) condition for PAIC* (Zhao et al., 2020)). *If a predictive CDF model $F_\Phi$ is $(\epsilon, \delta)$-MPAIC, then for any $\epsilon' > \epsilon$, it is also $\left(\epsilon', \delta\frac{1-\epsilon}{\epsilon'-\epsilon}\right)$-PAIC.*

In addition to using the *quantile regression* $\mathcal{L}_{\text{ALD+Cqr}}$ to improve the individual calibration, we note that an equivalent alternative is to use a *calibration* loss $\mathcal{L}_{\text{Cal}}$, defined directly over the predicted cumulative distribution. Specifically, this *calibration* loss measures the discrepancy between the predicted cumulative probability $F_\Phi(y|\mathbf{x}, q)$ and the target quantile percentage $q$, and is defined as:

$$\mathcal{L}_{\text{Cal}}(y; \Phi, q) = |F_\Phi(y|\mathbf{x}, q) - q|. \tag{11}$$

This loss enforces the calibration condition that, for a given input $\mathbf{x}$, the predicted CDF of *true event time distribution* aligns with the queried quantile percentage $q$, which is essentially equivalent to the *MPAIC* loss in Zhao et al. (2020). Importantly, $\mathcal{L}_{\text{Cal}}$ is evaluated over the joint distribution of all $(\mathbf{x}, y, Q)$ pairs, where $y$ is treated as a realization of the latent *true event time* random variable. Thus, whether an observation is *censored* or *uncensored* does not affect the validity of the assessment, since the modeling target remains the conditional distribution of *true event time*.

In the end, $\mathcal{L}_{\text{Cqr}}$ satisfies the first condition in Definition 2, while $\mathcal{L}_{\text{Cal}}$ enforces the second condition, making them equivalent under the assumption of $m_\Phi(\mathbf{x}, q)$. Hence, the overall training objective for the ICALD model in Eq. (10) can alternatively be formulated as:

$$\mathcal{L}_{\text{ALD+Cal}}(y; \Phi, q) = \mathcal{L}_{\text{ALD}}(y; m_\Phi(\mathbf{x}, q)) + \lambda\mathcal{L}_{\text{Cal}}(y; \Phi, q). \tag{12}$$

Furthermore, by Definition 2 and Theorem 1, we can conclude that the ICALD model $m_\Phi(\mathbf{x}, q)$, trained with the loss $\mathcal{L}_{\text{ALD+Cal}}$, is $(\epsilon, \delta)$-*MPAIC* and also $\left(\epsilon', \delta\frac{1-\epsilon}{\epsilon'-\epsilon}\right)$-*PAIC*.

## 3.4 PRE-CALIBRATION AND POST-CALIBRATION

A potential issue of *asynchronous convergence* may arise in *pre-calibration* models trained with Eq. (10) or Eq. (12). This happens when the log-likelihood and calibration losses converge at different speeds, which although not observed in most datasets, it is an issue in heavier-tailed ones. To address this, we introduce a *warm-up calibration* strategy (see Appendix C.4), where training initially focuses solely on the negative log-likelihood loss before incorporating the calibration loss at a later stage. Alternatively, *post-calibration* offers an even simpler and more effective approach for handling with this issue. As discussed above, the theoretical guarantees of calibration arise from the properties of the *quantile regression* loss $\mathcal{L}_{\text{Cqr}}$ (or *calibration* loss $\mathcal{L}_{\text{Cal}}$) itself. This enables *post-calibration* to be applied as a lightweight post-processing step, without retraining or modifications to the original model architecture. By decoupling the additional loss (*i.e.*, $\mathcal{L}_{\text{Cqr}}$ or $\mathcal{L}_{\text{Cal}}$) from the training dynamics, *post-calibration* also avoids noisy or conflicting gradient signals during early training stages, leading to more stable and reliable calibration.

Referring back to the *pre-calibration* model architecture (denoted as $m_\Phi^{\text{Pre}}$) illustrated in Fig. 1, we can infer that the parameters of the Individually Calibrated Asymmetric Laplace distribution (ICALD) are conditioned on both the input $\mathbf{x}$ and the quantile percentage $q \sim \mathcal{U}(0, 1)$. Therefore, we can utilize a simple *adapter module* (denoted as $m_\Phi^{\text{Post}}$), like in the top part of Fig. 1, that takes $\mathbf{x}$ and $q$ as input and outputs the *post-calibration* adjustment factors $\gamma \in \mathbb{R}^3$ for the ALD parameters estimated by the *base* model (denoted as $m_\Phi^{\text{Base}}$). Effectively, the *pre-calibration* model can be decomposed into a *post-calibration* model and a *base* model with the *quantile regression* loss or *calibration* loss as:

$$m_\Phi^{\text{Pre}}(\mathbf{x}, q) = \{\theta_q^*, \sigma_q^*, \kappa_q^*\} = m_\Phi^{\text{Post}}(\mathbf{x}, q) \odot m_\Phi^{\text{Base}}(\mathbf{x}) = \gamma \odot \{\theta_q, \sigma_q, \kappa_q\}, \tag{13}$$

where $\odot$ denotes element-wise multiplication, and $\theta_q^*, \sigma_q^*, \kappa_q^*$ are the ICALD parameters produced by the *pre-calibration* model. In practice, since both the *quantile regression* loss and the *calibration* loss

are Monte Carlo approximations of their respective theoretical expectations, it is crucial to sample as many quantile percentages $q \sim \mathcal{U}(0,1)$ as possible during training. As shown in Theorem 2 (with proof provided in Appendix A.4), increasing the number of quantile samples improves the approximation quality and enhances the model's *individual calibration* performance.

**Theorem 2** (*Concentration* (Zhao et al., 2020)). *Let $F_\Phi$ be any $(\epsilon, \delta)$-mPAIC predictive CDF model, and let $\{(\mathbf{x}_i, y_i)\}_{i=1}^n \overset{i.i.d.}{\sim} \mathbb{F}_{XY}, \{q_i\}_{i=1}^n \overset{i.i.d.}{\sim} \mathcal{U}(0,1)$. Then, with probability at least $1 - \gamma$, we have:*

$$\frac{1}{n} \sum_{i=1}^n \mathbb{I}\left(|F_\Phi(y_i|\mathbf{x}_i, q_i) - q_i| \geq \epsilon\right) \leq \delta + \sqrt{\frac{-\log \gamma}{2n}}.$$

To this end, we apply several practical strategies to improve both *pre-calibration* and *post-calibration* models. *Deepening the calibration anchor:* We first concatenate the quantile percentage $q$ at each network layer to ensure its influence propagates deeply through the architecture. Then we increase the number of training epochs to allow the model more opportunity to align predictions with the target quantiles. *Widening the calibration anchor:* Rather than using a scalar $q$ as in Zhao et al. (2020), we empirically found that expanding it into a small vector (*e.g.*, 4-dimensional when $n_{\text{hidden}} = 32$) enables richer interactions with learned features and improves the expressiveness of the quantile-conditioned outputs (see Appendix C.4). In general, both *pre-calibration* and *post-calibration* can be implemented with either $\mathcal{L}_{\text{ALD+Cqr}}$ or $\mathcal{L}_{\text{ALD+Cal}}$, offering a unified and flexible framework where the calibration strategy and loss function can be independently selected based on the use case.

**General-purpose post-hoc calibrator** Beyond calibrating our ALD instantiation, the proposed *post-calibration* module could be used as a general-purpose calibrator for other survival models (*e.g.*, RSF (Ishwaran et al., 2008), DeepSurv (Katzman et al., 2018), and DeepHit (Lee et al., 2018)). This is primarily because: ($i$) The adapter module (*i.e.*, the $q$-branch in Section 3.2) is agnostic to the choice of backbone feature extractor. In principle, it can be attached to MLPs, tree-based models, CNNs for image input, and BERT-style encoders for text — all without architectural restrictions. ($ii$) The calibration losses we introduce (*i.e.*, $\mathcal{L}_{\text{Cqr}}$ and $\mathcal{L}_{\text{Cal}}$) only require access to the estimated survival CDF (*i.e.*, $F_\Phi(y|\mathbf{x}, q)$), making them widely applicable as long as such a function can be computed.

However, we do note some practical considerations when adapting our calibrator to other models, such as RSF, DeepSurv, and DeepHit: ($i$) As discussed in Section 3.1 and Appendix C.4, these models are *nonparametric*, and their predicted distributions are often discretized rather than smooth and continuous. This poses a challenge when applying our calibration objectives, which assume access to a well-defined and continuous CDF. In such cases, a post-processing step (*e.g.*, linear interpolation between adjacent CDF points (Haider et al., 2020)) may be necessary to transform the discrete outputs into a usable continuous CDF. Also, *nonparametric* models have the *crossing quantiles* issue (See Section 3.1), thus additional constraints or penalties (*e.g.*, non-crossing loss (Bondell et al., 2010)) might be required to enforce monotonicity. ($ii$) Models like DeepHit already incorporate multiple training objectives (*e.g.*, the likelihood and ranking loss), so directly adding a calibration term may introduce training instability and exacerbate trade-offs between competing losses.

In summary, our framework is theoretically general and readily applies to many *parametric* survival models (*e.g.*, Hoseini et al. 2017; Nagpal et al. 2021; Sheng & Henao 2025), where a smooth CDF is directly available. For *nonparametric* models (*e.g.*, Ishwaran et al. 2008; Dembek et al. 2014; Lee et al. 2018; Katzman et al. 2018; Pearce et al. 2022), extra implementation care may be needed because the output CDF is typically discrete or piecewise-constant.

## 4 EXPERIMENTS

**Datasets** We evaluate our methods on a broad suite of datasets introduced by Pearce et al. (2022). These include two types: *synthetic event data with synthetic censoring* and *real event data with real censoring*. For *synthetic* datasets, inputs $\mathbf{x}$ are sampled uniformly from $\mathcal{U}(0,2)^d$, where $d$ is the number of features, with event times $e$ and censoring times $c$ generated from distinct, parameterized distributions to simulate diverse scenarios. For *real-world* datasets, we consider survival datasets from domains such as healthcare and oncology. Full descriptions for all datasets can be found in Appendix B.1. We follow standard practice by running each experiment with 5 random train/test splits. The source code required to reproduce the experiments presented in this paper is available at: `https://github.com/demingsheng/ICALD`.

Table 1: Performance comparison of our *pre-calibrated* ICALD across 21 datasets, showing the number of cases where it performs **significantly better**, **worse**, or **equal**, respectively. The last two rows report total counts and proportions across all 56 pairwise comparisons.

| Metric | $\mathcal{L}^{Pre}_{ALD+Cal}$ *vs.* $\mathcal{L}_{ALD}$ | | | $\mathcal{L}^{Pre}_{ALD+Cal}$ *vs.* $\mathcal{L}_{Cqr}$ | | | $\mathcal{L}^{Pre}_{ALD+Cqr}$ *vs.* $\mathcal{L}_{ALD}$ | | | $\mathcal{L}^{Pre}_{ALD+Cqr}$ *vs.* $\mathcal{L}_{Cqr}$ | | | $\mathcal{L}^{Pre}_{ALD+Cal}$ *vs.* $\mathcal{L}^{Pre}_{ALD+Cqr}$ | | | $\mathcal{L}^{Pre}_{ALD+Cal}$ *vs.* $\mathcal{L}^{Pre}_{ALD+X\text{-}CAL}$ | | |
|---|---|---|---|---|---|---|---|---|---|---|---|---|---|---|---|---|---|---|
| *Average Calibration* | 8 | 0 | 13 | 5 | 0 | 16 | 4 | 4 | 13 | 2 | 3 | 16 | 13 | 0 | 8 | 7 | 0 | 14 |
| *Group Calibration* | 8 | 0 | 13 | 11 | 0 | 10 | 7 | 2 | 12 | 3 | 0 | 18 | 14 | 0 | 7 | 9 | 0 | 12 |
| *Individual Calibration* | 6 | 0 | 8 | 7 | 1 | 6 | 4 | 6 | 4 | 4 | 5 | 5 | 9 | 0 | 5 | 9 | 3 | 2 |
| **Total** | 22 | 0 | 34 | 23 | 1 | 32 | 15 | 12 | 29 | 9 | 8 | 39 | 36 | 0 | 20 | 25 | 3 | 28 |
| **Proportion (%)** | 39.3 | 0.0 | 60.7 | 41.1 | 1.8 | 57.1 | 26.8 | 21.4 | 51.8 | 16.1 | 14.3 | 69.6 | 64.3 | 0.0 | 35.7 | 44.6 | 5.4 | 50.0 |

Table 2: Performance comparison of $\mathcal{L}^{Post}_{ALD+Cal}$ against three *post-calibrated* ALD baselines (*i.e.*, $\mathcal{L}^{Post}_{ALD+Cqr}$, $\mathcal{L}^{Post}_{ALD+CSD}$, and $\mathcal{L}^{Post}_{ALD+CiPOT}$) as well as its *pre-calibrated* counterpart $\mathcal{L}^{Pre}_{ALD+Cal}$, across 21 datasets. Each triplet reports the number of datasets where $\mathcal{L}^{Post}_{ALD+Cal}$ performs **significantly better**, **worse**, or the **same**, respectively, for each calibration metric. The final two rows show total counts and proportions across 56 pairwise comparisons.

| Metric | $\mathcal{L}^{Post}_{ALD+Cal}$ *vs.* $\mathcal{L}^{Post}_{ALD+Cqr}$ | | | $\mathcal{L}^{Post}_{ALD+Cal}$ *vs.* $\mathcal{L}^{Post}_{ALD+CSD}$ | | | $\mathcal{L}^{Post}_{ALD+Cal}$ *vs.* $\mathcal{L}^{Post}_{ALD+CiPOT}$ | | | $\mathcal{L}^{Post}_{ALD+Cal}$ *vs.* $\mathcal{L}^{Pre}_{ALD+Cal}$ | | |
|---|---|---|---|---|---|---|---|---|---|---|---|---|
| *Average Calibration* | 16 | 1 | 4 | 14 | 5 | 2 | 11 | 3 | 7 | 1 | 1 | 19 |
| *Group Calibration* | 16 | 1 | 4 | 14 | 6 | 1 | 13 | 0 | 8 | 6 | 2 | 13 |
| *Individual Calibration* | 11 | 0 | 3 | 12 | 0 | 2 | 7 | 0 | 7 | 1 | 1 | 12 |
| **Total** | 43 | 2 | 11 | 40 | 11 | 5 | 31 | 3 | 22 | 8 | 4 | 44 |
| **Proportion (%)** | 76.8 | 3.6 | 19.6 | 71.4 | 19.6 | 8.9 | 55.4 | 5.4 | 39.3 | 14.3 | 7.1 | 78.6 |

**Metrics** We evaluate each model using three categories of metrics: *Predictive Accuracy*, *Concordance*, and *Calibration*. For *predictive accuracy*, we report the Mean Absolute Error (MAE) and the Integrated Brier Score (IBS) (Graf et al., 1999), which quantify the accuracy of survival time predictions over time. For *concordance*, we use Harrell's C-Index (Harrell et al., 1982) and Uno's C-Index (Uno et al., 2011) to evaluate the model's ability to correctly rank survival times while accounting for censored observations. For *calibration*, we assess the reliability of survival probability estimates using Expected Calibration Error (ECE) (Naeini et al., 2015) for both *average* and *group calibration*, and the average Wasserstein Distance (Villani, 2009) between predicted and empirical survival distributions to evaluate *individual calibration*. Details can be found in Appendix B.2.

**Baselines** We compare the proposed method against 12 baselines: ALD (Sheng & Henao, 2025), Log-Norm (Hoseini et al., 2017), DSM (Nagpal et al., 2021) (log normal and Weibull), DeepSurv (Katzman et al., 2018), CQRNN (Pearce et al., 2022), DeepHit (Lee et al., 2018), GBM (Dembek et al., 2014), and RSF (Ishwaran et al., 2008), covering a broad spectrum of survival models, including *(semi-)parametric* and *nonparametric* approaches, as well as both *neural* and *non-neural* architectures. Furthermore, we compare to 1 *pre-calibration* method X-CAL (Goldstein et al., 2020) and 2 recent *post-calibration* methods, CSD (Qi et al., 2024a) and CiPOT (Qi et al., 2024b). These baselines represent diverse modeling strategies and provide a comprehensive and principled benchmark for evaluation. A complementary discussion of *Related Work* is provided in Appendix A.6, while detailed descriptions and implementation notes for each baseline are presented in Appendix B.3.

**Pre-Calibration Comparison** We first evaluate the impact of the proposed calibration strategy in the *pre-calibration* setting. Table 1 evaluates the impact of our proposed calibration strategies across three calibration dimensions: *average*, *group*, and *individual calibration*. When assessing the statistical significance of the different metrics (*i.e.*, Tables 1, 2, and 3), we use a Student's $t$ test with $p < 0.05$ considered significant after correction for false discovery rate using Benjamini-Hochberg (Benjamini & Hochberg, 1995). We observe that both ICALD methods (*i.e.*, $\mathcal{L}^{Pre}_{ALD+Cal}$ and $\mathcal{L}^{Pre}_{ALD+Cqr}$) consistently outperform the single ALD baselines (*i.e.*, $\mathcal{L}_{ALD}$ and $\mathcal{L}_{Cqr}$) and $\mathcal{L}^{Pre}_{ALD+X\text{-}CAL}$ in the majority of cases. Among these, $\mathcal{L}^{Pre}_{ALD+Cal}$ demonstrates a clear advantage over $\mathcal{L}^{Pre}_{ALD+Cqr}$ (64.3% wins *vs.* 0% losses), confirming the effectiveness of the calibration loss $\mathcal{L}_{Cal}$ as a principled and consistent training objective. As discussed in Section 3.3, both ICALD models trained with $\mathcal{L}^{Pre}_{ALD+Cal}$ and $\mathcal{L}^{Pre}_{ALD+Cqr}$ are $(\epsilon, \delta)$-*MPAIC* and therefore also $\left(\epsilon', \delta \frac{1-\epsilon}{\epsilon'-\epsilon}\right)$-*PAIC*. Although both are expected to improve calibration, their empirical effectiveness depends on the performance of the additional loss function (*i.e.*, $\mathcal{L}_{Cal}$ and $\mathcal{L}_{Cqr}$). The superior performance of $\mathcal{L}_{Cal}$ over $\mathcal{L}_{Cqr}$ is likely due to issues of the latter when handling censored data effectively. We provide a detailed discussion and analysis of this issue in Appendix C.4.

Table 3: General comparison of $\mathcal{L}^{\text{Post}}_{\text{ALD+Cal}}$ with nine baselines across 21 datasets. Each group of three columns reports the number of datasets where our method performs **significantly better**, **worse**, or the **same**, respectively. The final two rows summarize total counts and proportions across 140 pairwise comparisons.

| Metric | ALD | | | CQRNN | | | LogNorm | | | DeepSurv | | | DSM (Weibull) | | | DSM (LogNorm) | | | DeepHit | | | GBM | | | RSF | | |
|---|---|---|---|---|---|---|---|---|---|---|---|---|---|---|---|---|---|---|---|---|---|---|---|---|---|---|---|
| MAE | 0 | 3 | 18 | 6 | 11 | 4 | 9 | 7 | 5 | 7 | 11 | 3 | 12 | 6 | 3 | 13 | 6 | 2 | 12 | 5 | 4 | 7 | 9 | 5 | 11 | 6 | 4 |
| IBS | 8 | 0 | 13 | 10 | 2 | 9 | 21 | 0 | 0 | 21 | 0 | 0 | 14 | 2 | 5 | 14 | 2 | 5 | 21 | 0 | 0 | 5 | 3 | 13 | 13 | 2 | 6 |
| Harrell's C-Index | 0 | 0 | 21 | 1 | 1 | 19 | 4 | 1 | 16 | 4 | 3 | 14 | 15 | 0 | 6 | 14 | 1 | 6 | 0 | 0 | 21 | 8 | 2 | 11 | 8 | 0 | 13 |
| Uno's C-Index | 0 | 0 | 21 | 1 | 1 | 19 | 4 | 0 | 17 | 5 | 3 | 13 | 15 | 0 | 6 | 12 | 1 | 8 | 0 | 0 | 21 | 9 | 2 | 10 | 10 | 0 | 11 |
| Average Calibration | 7 | 0 | 14 | 13 | 1 | 7 | 11 | 1 | 9 | 3 | 1 | 17 | 16 | 1 | 4 | 15 | 1 | 5 | 15 | 2 | 4 | 12 | 2 | 7 | 8 | 1 | 12 |
| Group Calibration | 8 | 0 | 13 | 20 | 1 | 0 | 12 | 1 | 8 | 10 | 1 | 10 | 18 | 1 | 2 | 19 | 1 | 1 | 14 | 2 | 5 | 12 | 1 | 8 | 15 | 1 | 5 |
| Individual Calibration | 9 | 0 | 5 | 6 | 0 | 8 | 10 | 2 | 2 | 3 | 0 | 11 | 13 | 1 | 0 | 13 | 0 | 1 | 13 | 0 | 1 | 14 | 0 | 0 | 13 | 0 | 1 |
| **Total** | 32 | 3 | 105 | 57 | 17 | 66 | 71 | 12 | 57 | 53 | 19 | 68 | 103 | 11 | 26 | 100 | 12 | 28 | 75 | 9 | 56 | 67 | 19 | 54 | 78 | 10 | 52 |
| **Proportion (%)** | 22.9 | 2.1 | 75.0 | 40.7 | 12.1 | 47.1 | 50.7 | 8.6 | 40.7 | 37.9 | 13.6 | 48.6 | 73.6 | 7.9 | 18.6 | 71.4 | 8.6 | 20.0 | 53.6 | 6.4 | 40.0 | 47.9 | 13.6 | 38.6 | 55.7 | 7.1 | 37.1 |

Fig. 2 further illustrates the *individual calibration* results by highlighting the best and worst improvement cases of $\mathcal{L}^{\text{Pre}}_{\text{ALD+Cal}}$ compared to $\mathcal{L}_{\text{ALD}}$ on the `Norm Linear` dataset. Full comparison for the *pre-calibration* results across all datasets are provided in Appendix C.1. As discussed in Section 3.1, both the original objectives $\mathcal{L}_{\text{ALD}}$ and $\mathcal{L}_{\text{Cqr}}$ exhibit limited performance due to their inherent limitations. In contrast, both $\mathcal{L}^{\text{Pre}}_{\text{ALD+Cal}}$ and $\mathcal{L}^{\text{Pre}}_{\text{ALD+Cqr}}$ significantly improve calibration perfor-

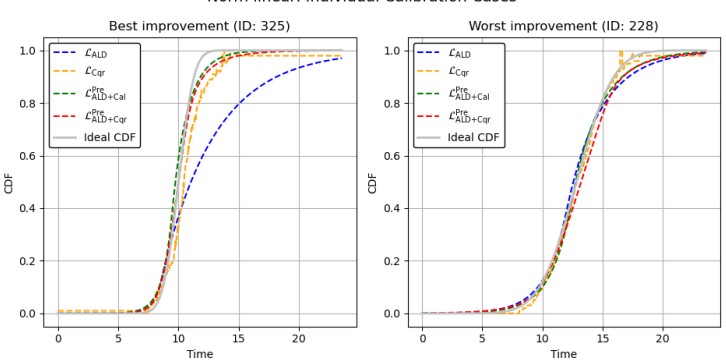

Figure 2: Illustration of the best (Left) and worst (Right) *individual calibration* improvement cases ($\mathcal{L}^{\text{Pre}}_{\text{ALD+Cal}}$ *vs.* $\mathcal{L}_{\text{ALD}}$) achieved by hybrid ALD-based survival models.

mance in most cases, as seen in Fig. 2 (Left). These improvements lead the estimated CDFs to better align with the ideal CDFs, which is the essence of *individual calibration*. Notably, the performance of $\mathcal{L}^{\text{Pre}}_{\text{ALD+Cal}}$ and $\mathcal{L}^{\text{Pre}}_{\text{ALD+Cqr}}$ is comparable in this example, further suggesting that when their loss formulations (*i.e.*, Eq. (10) and Eq. (12)) behave similarly, their calibration outcomes are expected to be similar as well. This is theoretically consistent with the shared *MPAIC* property. However, there are cases where calibration yields only marginal benefits or even slight degradation, like in Fig. 2 (Right), possibly due to the inherent variability across individual samples.

**Post-Calibration Comparison** We then evaluate the impact of the proposed calibration strategy in the *post-calibration* setting. Table 2 summarizes the comparison between the *post-calibrated* ICALD model $\mathcal{L}^{\text{Post}}_{\text{ALD+Cal}}$ and three strong *post-calibration* baselines: $\mathcal{L}^{\text{Post}}_{\text{ALD+Cqr}}$, $\mathcal{L}^{\text{Post}}_{\text{ALD+CSD}}$, and $\mathcal{L}^{\text{Post}}_{\text{ALD+CiPOT}}$. We also include a comparison with the original *pre-calibrated* $\mathcal{L}^{\text{Pre}}_{\text{ALD+Cal}}$ to assess the relative benefit of applying calibration as a post-processing step. Full *post-calibration* results across all datasets are provided in Appendix C.2. Results clearly show that $\mathcal{L}^{\text{Post}}_{\text{ALD+Cal}}$ outperforms all other *post-calibration* baselines in most cases (*i.e.*, **significantly better**: 76.8% over $\mathcal{L}^{\text{Post}}_{\text{ALD+Cqr}}$, 71.4% over $\mathcal{L}^{\text{Post}}_{\text{ALD+CSD}}$, and 55.4% over $\mathcal{L}^{\text{Post}}_{\text{ALD+CiPOT}}$), especially compared to the two recent novel *post-calibration* strategies CSD (Qi et al., 2024a) and CiPOT (Qi et al., 2024b). Furthermore, although $\mathcal{L}^{\text{Post}}_{\text{ALD+Cal}}$ shows only slightly better performance than its *pre-calibrated* counterpart, this marginal gain may be attributed to the issue of *asynchronous convergence* in *pre-calibration* discussed in Section 3.4. In summary, *post-calibration* results reaffirm the robustness of the hybrid ALD model, particularly when using $\mathcal{L}_{\text{Cal}}$.

**General Comparison** We also evaluate and compare our *post-calibrated* ICALD model $\mathcal{L}^{\text{Post}}_{\text{ALD+Cal}}$ against 9 baseline methods across 7 evaluation metrics on 21 datasets. Complete results for all datasets are provided in Appendix C.3. We further provide an interpretability and robustness case study in Appendix C.4, including stratified analyses across time horizons and quantile levels. Table 3 shows that our method consistently delivers strong performance, achieving gains over all baselines in a substantial subset of comparisons.

*ICALD vs. ALD & CQRNN:* Our method achieves significant gains in *individual calibration* (9 **significantly better** and 0 worse over ALD, and 6 **significantly better** and 0 worse over CQRNN), and also improves *average* and *group calibration* on over half of the datasets. This indicates that simply relying on *quantile regression* $\mathcal{L}_{\text{Cqr}}$ or *maximum likelihood* $\mathcal{L}_{\text{ALD}}$ may lead to under-calibration, and that the proposed calibration strategy via $\mathcal{L}_{\text{ALD+Cal}}^{\text{Post}}$ encourages individual-level alignment with ground-truth distributions.

*ICALD vs. (Semi-)Parametric & Mixture Models:* Our model consistently improves both *calibration* and *accuracy* metrics. Specifically, $\mathcal{L}_{\text{ALD+Cal}}^{\text{Post}}$ outperforms the *semi-parametric* model DeepSurv in metrics such as IBS (21 **significantly better** and 0 worse), *average calibration* (15 and 2), and *group calibration* (14 and 2). Moreover, compared to the *parametric mixture* DSM (Weibull) model, our method leads in 73.6% of all comparisons (103 out of 140), especially excelling in MAE, *concordance*, and all forms of *calibration*. This highlights that ICALD provides better generalization than fixed-form *parametric* assumptions or *mixture* modeling approaches.

*ICALD vs. Nonparametric Models:* Our model demonstrates superior *calibration*, especially in *group* and *individual calibration*. ICALD outperforms DeepHit on *group calibration* in 14 of 21 datasets, and achieves **significantly better** *individual calibration* on all 13 datasets. Similar trends are observed for GBM (12/1/8 for *group calibration*) and RSF (15/1/5), confirming that our model delivers better calibrated estimates, despite the strong expressiveness of *ensemble* or *neural nonparametric* baselines.

## 5 CONCLUSION

In this paper, we introduced a novel survival modeling framework ICALD that unifies the strengths of *parametric* and *nonparametric* ALD approaches. By supporting two theoretically equivalent loss functions (*i.e.*, $\mathcal{L}_{\text{ALD+Cal}}$ and $\mathcal{L}_{\text{ALD+Cqr}}$) with formal guarantees for *Probably Approximately Individually Calibrated* (PAIC) learning, ICALD offers a flexible and principled framework for improving calibration in both *pre-* and *post-calibration* settings. Through extensive experiments on 21 benchmark datasets, we demonstrate that ICALD consistently outperforms a wide range of strong baselines, including both traditional and neural survival models, as well as recent *pre-calibration* and *post-calibration* techniques. These results highlight the effectiveness and generalizability of our approach in achieving *accurate*, *concordant*, and *calibrated* survival predictions.

**Limitations** While increasing the number of quantile percentage samples improves the Monte Carlo approximation and calibration quality (Theorem 2), it also prolongs training and may lead to *overfitting*, particularly on heavily skewed datasets (*e.g.*, `LogNorm`). Although we apply *early stopping* to maintain a proper trade-off between calibration and generalization, more principled solutions remain to be explored. In addition, for the joint loss formulations $\mathcal{L}_{\text{ALD+Cal}}^{\text{Pre}}$ and $\mathcal{L}_{\text{ALD+Cqr}}^{\text{Pre}}$, we observe the *synchronous convergence* issue in some cases. To mitigate this, we adopt a *warm-up calibration strategy* that delays the introduction of the additional loss (*i.e.*, $\mathcal{L}_{\text{Cal}}$ and $\mathcal{L}_{\text{Cqr}}$) to encourage alignment between optimization objectives. However, this approach does not fully resolve the convergence mismatch during training. Although both $\mathcal{L}_{\text{ALD+Cal}}$ and $\mathcal{L}_{\text{ALD+Cqr}}$ are theoretically grounded in improving *individual calibration* (Theorem 1), the latter appears more sensitive to censoring. Its reliance on the Portnoy estimator may limit its ability to capture reliable calibration signals under censored conditions (see Appendix C.4). In contrast, $\mathcal{L}_{\text{ALD+Cal}}$ offers a more robust and stable calibration objective for censored survival data. Thus, improving the performance of $\mathcal{L}_{\text{ALD+Cqr}}$ in the presence of censoring remains an open challenge for future work.

Lastly, our empirical evaluation is conducted on a benchmark suite chosen to ensure fair and consistent comparisons with prior work, following the datasets used in Pearce et al. (2022) and Sheng & Henao (2025). While this suite spans diverse distribution families and clinical censoring regimes, it is not exhaustive, and performance may vary across domains and dataset characteristics. Evaluating ICALD on larger, independent, and more diverse benchmarks is therefore an important direction for future work. To partially address concerns about external validity and scalability, we additionally evaluate ICALD on two high-dimensional gene-expression datasets (DBCD; Rosenwald et al. 2002 and DLBCL; Van Houwelingen et al. 2006), as well as four finance and engineering datasets from the PySurvival package[1]. In the end, we provide a dedicated scalability case study in Appendix C.4, where ICALD demonstrates competitive performance, particularly on *calibration* metrics.

---

[1] `https://github.com/square/pysurvival`

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

# A  ANALYTICAL RESULTS

This section presents the theoretical foundations and formal guarantees of the proposed ICALD models and their calibration properties. In Appendix A.1, we show that the widely used *pinball loss* can be interpreted as a special case of the negative log-likelihood of the Asymmetric Laplace Distribution (ALD), providing a probabilistic justification for quantile-based objectives. Appendix A.2 derives the full loss formulation for *parametric* ALD model in the presence of censoring. Appendix A.3 discusses the properties of continuous mixtures of ALDs. Appendix A.4 establishes formal calibration guarantees (Theorem 1) for the ICALD models, demonstrating that training with either $\mathcal{L}_{\text{ALD+Cal}}$ (Equation 12) or $\mathcal{L}_{\text{ALD+Cqr}}$ (Equation 10) yields models that satisfy $(\epsilon, \delta)$-MPAIC and, consequently, $\left(\epsilon', \delta \cdot \frac{1-\epsilon}{\epsilon'-\epsilon}\right)$-*PAIC*. In addition, Appendix A.4 includes Theorem 2, which provides a high-probability generalization bound showing that increasing the number of sampled quantile levels improves the Monte Carlo approximation and strengthens empirical individual calibration. Appendix A.5 provides an extended discussion of the monotonicity constraints, clarifying why monotonicity is required, when it can or cannot be ignored, and how to address nonmonotonic predictions using monotone rearrangement. Also, a discussion of *Related Work* is provided in Appendix A.6.

## A.1  PINBALL LOSS AS A SPECIAL CASE OF THE ALD LIKELIHOOD

**Lemma 1.** *The pinball loss is equivalent to the negative log-likelihood of the Asymmetric Laplace Distribution (ALD) in its* quantile *parameterization* $\mathcal{AL}(\theta = \tilde{y}_q, \sigma = 1, q)$*, up to an additive constant.*

*Proof.* Let a random variable $Y \sim \mathcal{AL}(\theta, \sigma, q)$, where $\theta \in \mathbb{R}$ is the location (the $q$-th quantile, *i.e.,* $y_q$), $\sigma > 0$ is the scale, and $q \in (0, 1)$ is the target quantile. Then, the probability density function (PDF) of the ALD in its *quantile form* is given by:

$$f(y; \theta, \sigma, q) = \frac{q(1-q)}{\sigma} \begin{cases} \exp\left(\frac{q}{\sigma}(\theta - y)\right), & y \geq \theta, \\ \exp\left(\frac{1-q}{\sigma}(y - \theta)\right), & y < \theta. \end{cases} \tag{14}$$

The negative log-likelihood becomes:

$$-\log f(y; \theta, \sigma, q) = \log\left(\frac{\sigma}{q(1-q)}\right) + \begin{cases} \frac{q}{\sigma}(y - \theta), & y \geq \theta, \\ \frac{1-q}{\sigma}(\theta - y), & y < \theta. \end{cases} \tag{15}$$

On the other hand, the *pinball* loss used in *quantile regression* is defined as:

$$\mathcal{L}_{\text{pinball}}(y; \tilde{y}_q) = (y - \tilde{y}_q)(q - \mathbb{I}[\tilde{y}_q > y]) = \begin{cases} q(y - \tilde{y}_q), & y \geq \tilde{y}_q, \\ (1-q)(\tilde{y}_q - y), & y < \tilde{y}_q. \end{cases} \tag{16}$$

Now, if we let $\sigma = 1$ and $\theta = \tilde{y}_q$ be the predicted quantile value. Then, we can conclude:

$$-\log f(y; \tilde{y}_q, 1, q) = \mathcal{L}_{\text{pinball}}(y; \tilde{y}_q) - \log(q(1-q)). \tag{17}$$

This shows that the *pinball* loss is proportional to the negative log-likelihood of the ALD, up to an additive constant. This connection provides a probabilistic interpretation of *quantile regression* and justifies likelihood-based *parameteric* modeling extensions using the ALD. □

## A.2  LOSS FUNCTION FOR THE PARAMETRIC ALD MODEL

**Lemma 2.** *Let* $Y \sim \mathcal{AL}(\theta, \sigma, \kappa)$*, where* $\mathcal{AL}$ *denotes the Asymmetric Laplace Distribution with location parameter* $\theta \in \mathbb{R}$*, scale parameter* $\sigma > 0$*, and asymmetry parameter* $\kappa > 0$*. Suppose a parametric model* $m_\Phi(\mathbf{x}) = \{\theta, \sigma, \kappa\}$ *maps input covariates* $\mathbf{x}$ *to the corresponding ALD parameters. Then, the total loss function over both observed* $\mathcal{D}_O$ *and censored data* $\mathcal{D}_C$ *is given by:*

$$\mathcal{L}_{ALD}(y; \Phi) = - \sum_{n \in \mathcal{D}_O} \log f_{ALD}(y_n; m_\Phi(\mathbf{x}_n)) - \sum_{n \in \mathcal{D}_C} \log S_{ALD}(y_n; m_\Phi(\mathbf{x}_n)), \quad (18)$$

*where $f_{ALD}$ and $S_{ALD} = 1 - F_{ALD}$ denote the probability density function (PDF) and survival function of the ALD, respectively.*

*Proof.* The PDF and CDF are explicitly defined as:

$$f_{\text{ALD}}(y; \theta, \sigma, \kappa) = \frac{\sqrt{2}}{\sigma} \cdot \frac{\kappa}{1 + \kappa^2} \begin{cases} \exp\left(-\frac{\sqrt{2}\kappa}{\sigma}(y - \theta)\right), & y \geq \theta, \\ \exp\left(-\frac{\sqrt{2}}{\sigma\kappa}(\theta - y)\right), & y < \theta. \end{cases} \quad (19)$$

$$F_{\text{ALD}}(y; \theta, \sigma, \kappa) = \begin{cases} 1 - \frac{1}{1+\kappa^2} \exp\left(-\frac{\sqrt{2}\kappa}{\sigma}(y - \theta)\right), & y \geq \theta, \\ \frac{\kappa^2}{1+\kappa^2} \exp\left(-\frac{\sqrt{2}}{\sigma\kappa}(\theta - y)\right), & y < \theta. \end{cases} \quad (20)$$

Accordingly, the negative log-likelihood for observed samples ($\delta = 1$) is:

$$- \sum_{n \in \mathcal{D}_O} \log f_{\text{ALD}}(y_n; m_\Phi(\mathbf{x}_n)) \quad (21)$$

$$= \sum_{n \in \mathcal{D}_O} \left[ \log \sigma_n - \log\left(\frac{\kappa_n}{\kappa_n^2 + 1}\right) + \frac{\sqrt{2}}{\sigma_n} \begin{cases} \kappa_n(y_n - \theta_n), & y_n \geq \theta_n, \\ \frac{1}{\kappa_n}(\theta_n - y_n), & y_n < \theta_n, \end{cases} \right]$$

For censored samples ($\delta = 0$), the loss is derived from the survival function:

$$- \sum_{n \in \mathcal{D}_C} \log S_{\text{ALD}}(y_n; m_\Phi(\mathbf{x}_n)) \quad (22)$$

$$= \sum_{n \in \mathcal{D}_C} \begin{cases} \log(\kappa_n^2 + 1) + \frac{\sqrt{2}}{\sigma_n}\kappa_n(y_n - \theta_n), & y_n \geq \theta_n, \\ \log(\kappa_n^2 + 1) - \log\left[1 + \kappa_n^2\left(1 - \exp\left(-\frac{\sqrt{2}}{\sigma_n\kappa_n}(\theta_n - y_n)\right)\right)\right], & y_n < \theta_n. \end{cases}$$

These formulations enable efficient optimization of ALD-based models for both observed and censored survival data, by leveraging the closed-form expressions of the ALD's PDF and CDF. $\quad \square$

### A.3 The Properties of the Mixture of ALD

Given $Y \sim \mathcal{AL}(\theta, \sigma, \kappa)$, $Y_{\text{Mix}} = \int p(r) f_{\text{ALD}}(y; m_\Phi(\mathbf{x}, r)) dr$ where $r \sim \mathcal{U}(0, 1)$, and $m_\Phi(\mathbf{x}, r) = \{\theta_r, \sigma_r, \kappa_r\}$ are the ALD parameters predicted by the model for each quantile percentage $r$. Here, to avoid confusion with the quantile value $\tilde{y}_q = F_Y^{-1}(q \mid \mathbf{x})$, we use $r$ instead of $q$ to denote the random quantile percentage. We have:

$$\mathbb{E}[Y] = \theta + \frac{\sigma}{\sqrt{2}}\left(\frac{1}{\kappa} - \kappa\right), \quad \text{Var}[Y] = \frac{\sigma^2}{2}\left(\frac{1}{\kappa^2} + \kappa^2\right), \quad (23)$$

$$\mathbb{E}[Y_{\text{Mix}}] = \mathbb{E}_{r \sim \mathcal{U}(0,1)}[\mathbb{E}[Y_r]] = \int_0^1 \left(\theta_r + \frac{\sigma_r}{\sqrt{2}}\left(\frac{1}{\kappa_r} - \kappa_r\right)\right) dr \approx \frac{1}{N} \sum_{i=1}^{N} \left[\theta_i + \frac{\sigma_i}{\sqrt{2}}\left(\frac{1}{\kappa_i} - \kappa_i\right)\right] \quad (24)$$

By the law of total variance, we have:

$$\text{Var}[Y_{\text{Mix}}] = \mathbb{E}_{r \sim \mathcal{U}(0,1)}[\text{Var}[Y_r]] + \text{Var}_r[\mathbb{E}[Y_r]], \quad (25)$$

where the within-component variance is:

$$\mathbb{E}_{r \sim \mathcal{U}(0,1)}[\text{Var}[Y_r]] = \int_0^1 \frac{\sigma_r^2}{2} \left( \frac{1}{\kappa_r^2} + \kappa_r^2 \right) dr \approx \frac{1}{N} \sum_{i=1}^N \left[ \frac{\sigma_i^2}{2} \left( \frac{1}{\kappa_i^2} + \kappa_i^2 \right) \right], \qquad (26)$$

and the between-component variance is:

$$\text{Var}_r[\mathbb{E}[Y_r]] = \frac{1}{N} \sum_{i=1}^N \left( \mathbb{E}[Y_r] - \mathbb{E}[Y_{\text{Mix}}] \right)^2 \qquad (27)$$

Similar to the computation of the mixture mean $\mathbb{E}[Y_{\text{Mix}}]$, we can estimate the quantiles of the mixture ALD model by averaging quantile values $y_q$ sampled over $r \sim \mathcal{U}(0,1)$ from the predicted parameters $\{\theta_r, \sigma_r, \kappa_r\}$. Formally, the mixture quantile $\tilde{y}_q$ is estimated as:

$$y_q = \begin{cases} \theta + \frac{\sigma\kappa}{\sqrt{2}} \log \left[ \frac{1+\kappa^2}{\kappa^2} q \right], & \text{if } q \in \left( 0, \frac{\kappa^2}{1+\kappa^2} \right], \\ \theta - \frac{\sigma}{\sqrt{2}\kappa} \log \left[ (1+\kappa^2)(1-q) \right], & \text{if } q \in \left( \frac{\kappa^2}{1+\kappa^2}, 1 \right). \end{cases} \qquad (28)$$

$$\tilde{y}_q = \mathbb{E}_{r \sim \mathcal{U}(0,1)}[y_{q,r}] \approx \frac{1}{N} \sum_{i=1}^N y_{q,r}. \qquad (29)$$

### A.4 THEORETICAL FOUNDATIONS OF INDIVIDUAL CALIBRATION

In this section, we will show the ICALD model $\{\theta, \sigma, \kappa\} = m_\Phi(\mathbf{x}, q)$, trained with the loss $\mathcal{L}_{\text{ALD+Cal}}$ or $\mathcal{L}_{\text{ALD+Cqr}}$, is $(\epsilon, \delta)$-*MPAIC* and also $\left( \epsilon', \delta \cdot \frac{1-\epsilon}{\epsilon'-\epsilon} \right)$-*PAIC*. Now, we begin by recalling the definitions of *PAIC* and *MPAIC* provided in Definition 1 and Definition 2. These definitions are slight extensions of the original formulation in Zhao et al. (2020), incorporating an equivalent expression based on the inverse CDF. Note that these extended definitions allow us to generalize the original proof in Zhao et al. (2020), which establishes that training a model with the calibration loss $\mathcal{L}_{\text{ALD+Cal}}$ yields *PAIC* and *MPAIC* guarantees. In our case, we show that the same guarantees hold when the model is trained with the equivalent quantile-based loss $\mathcal{L}_{\text{ALD+Cqr}}$.

**Definition 1** (*Probably Approximately Individually Calibrated (PAIC; Zhao et al. 2020)*). *A predictive CDF model $F_\Phi(Y|\mathbf{x})$ is said to be $(\epsilon, \delta)$-PAIC if for all $\mathbf{x} \in \mathcal{X}$, $Y \in \mathcal{Y}$, and $q \in [0,1]$, the following holds:*

$$\Pr \left[ \int_0^1 \left| \Pr \left[ Y \le F_\Phi^{-1}(q|\mathbf{x}) \right] - q \right| dq \le \epsilon \right] \text{ or } \Pr \left[ \int_0^1 \left| \Pr \left[ F_\Phi(Y|\mathbf{x}) \le q \right] - q \right| dq \le \epsilon \right] \ge 1 - \delta.$$

**Definition 2** (*Monotonic Probably Approximately Individually Calibrated (MPAIC; Zhao et al. 2020)*). *A predictive CDF model $F_\Phi(Y|\mathbf{x}, q)$ is said to be $(\epsilon, \delta)$-MPAIC if for all $\mathbf{x} \in \mathcal{X}$, $Y \in \mathcal{Y}$, and $q \in [0,1]$, the following holds:*
$$\Pr \left[ \int_0^1 \left| \Pr \left[ Y \le F_\Phi^{-1}(q|\mathbf{x}, q) \right] - q \right| dq \le \epsilon \right] \text{ or } \Pr \left[ \int_0^1 \left| \Pr \left[ F_\Phi(Y|\mathbf{x}, q) \le q \right] - q \right| dq \le \epsilon \right] \ge 1 - \delta.$$

Then, we will show the proof for Theorem 1.

**Theorem 1** (*MPAIC is a sufficient (but not necessary) condition for PAIC (Zhao et al., 2020)*). *If a predictive CDF model $F_\Phi$ is $(\epsilon, \delta)$-MPAIC, then for any $\epsilon' > \epsilon$, it is also $\left( \epsilon', \delta \cdot \frac{1-\epsilon}{\epsilon'-\epsilon} \right)$-PAIC*

*Proof.* Let $Y \sim F_{Y|\mathbf{x}}$ and $Q \sim \mathcal{U}(0,1)$ be an independent random variable. Define the expected calibration error (ECE) using the 1-Wasserstein distance as:

$$\text{ECE}(F_\Phi) = \int_0^1 |\Pr[F_\Phi(Y|X) \le q] - q| \, dq = d_{W_1} \left( \mathbb{F}_{F_\Phi(Y|X)}, \mathbb{F}_\mathbb{U} \right), \qquad (30)$$

where $\mathbb{F}_{F_\Phi(Y|X)}$ denotes the true CDF of the predicted cumulative probabilities, and $\mathbb{F}_\mathbb{U}$ denotes the CDF of $\mathcal{U}(0,1)$. Intuitively, $d_{W_1}\left(\mathbb{F}_{F_\Phi(Y|X)}, \mathbb{F}_\mathbb{U}\right)$ try to integrate the difference between the curve $q \mapsto \Pr[F_\Phi(Y|X) \leq q]$ and the curve $q \mapsto q$. Then, we can define the calibration error as:

$$\mathrm{err}(\mathbf{x}, y) = d_{W_1}\left(\mathbb{F}_{F_\Phi(y|\mathbf{x},Q)}, \mathbb{F}_\mathbb{U}\right), \mathrm{err}(\mathbf{x}) = d_{W_1}\left(\mathbb{F}_{F_\Phi(Y|\mathbf{x},Q)}, \mathbb{F}_\mathbb{U}\right). \tag{31}$$

**Case 1: Monotonic mapping in $q$.** If $F_\Phi(y|\mathbf{x}, \cdot)$ increases monotonically in $Q$, then we have:

$$\mathrm{err}(\mathbf{x}, y) = \int_0^1 |F_\Phi(y|\mathbf{x}, q) - q|\, dq = \mathbb{E}_{Q\sim\mathcal{U}(0,1)}\left[|F_\Phi(y|\mathbf{x}, Q) - Q|\right]. \tag{32}$$

Let $Z = F_\Phi(y|\mathbf{x}, Q)$, where $Q \sim \mathcal{U}(0,1)$. Then the CDF of $Z$ is given by:

$$\mathbb{F}_Z(z) = \Pr(Z \leq z) = \Pr(F_\Phi(y|x, Q) \leq z). \tag{33}$$

Now, if $F_\Phi(y|x, q)$ is a monotonically nondecreasing continuous function of $q$, then the mapping $q \mapsto F_\Phi(y|x, q)$ is measure-preserving. This implies that:

$$\mathbb{F}_Z(z) = \Pr(Q \leq F_\Phi^{-1}(y|x, z)), \tag{34}$$

and hence,

$$\mathbb{F}_Z^{-1}(q) = F_\Phi(y|x, q), \quad \forall q \in [0, 1]. \tag{35}$$

Let $\mathbb{F}_\mathbb{U}$ denote the CDF of the uniform distribution $\mathcal{U}(0,1)$, that is,

$$\mathbb{F}_\mathbb{U}(u) = \Pr(U \leq u) = u, \quad \text{so} \quad \mathbb{F}_\mathbb{U}^{-1}(q) = q, \quad \forall q \in [0, 1]. \tag{36}$$

According to the property for the 1-Wasserstein distance (Villani et al., 2008) between two distributions $\mu$ and $\nu$ on the real line, if $\mathbb{F}_\mu^{-1}$ and $\mathbb{F}_\nu^{-1}$ are their respective quantile functions (*i.e.*, the inverse function of CDF), then:

$$d_{W_1}(\mu, \nu) = \int_0^1 \left|\mathbb{F}_\mu^{-1}(q) - \mathbb{F}_\nu^{-1}(q)\right|\, dq. \tag{37}$$

Applying this identity to $Z$ and $Q$, we can obtain:

$$d_{W_1}(\mathbb{F}_Z, \mathbb{F}_\mathbb{U}) = \int_0^1 |F_\Phi(y|x, q) - q|\, dq, \tag{38}$$

which is exactly Eq. (32).

**Case 2: General case without monotonicity.** In general, even if monotonicity doesn't hold, the following inequality still applies:

$$\mathrm{err}(\mathbf{x}, y) \leq \mathbb{E}_{Q\sim\mathcal{U}(0,1)}\left[|F_\Phi(y|\mathbf{x}, Q) - Q|\right]. \tag{39}$$

It is derived from the Kantorovich–Rubinstein duality for the 1-Wasserstein distance (Villani et al., 2008):

$$d_{W_1}(\mu, \nu) = \sup_{\|\psi\|_{\mathrm{Lip}} \leq 1}\left(\int \psi\, d\mu - \int \psi\, d\nu\right) = \sup_{\|\psi\|_{\mathrm{Lip}} \leq 1}|\mathbb{E}_\mu[\psi] - \mathbb{E}_\nu[\psi]|, \tag{40}$$

where the supremum is taken over all 1-Lipschitz functions $\psi : \mathbb{R} \to \mathbb{R}$, *i.e.*, functions that satisfy

$$|\psi(x) - \psi(y)| \leq |x - y|, \quad \forall x, y \in \mathbb{R}. \tag{41}$$

Applying this duality to $Z$ and $Q$, and choosing the 1-Lipschitz function $\psi(a) = a$, we can obtain:

$$d_{W_1}(\mathbb{F}_Z, \mathbb{F}_\mathbb{U}) \leq |\mathbb{E}_{Q\sim\mathcal{U}(0,1)}[Z] - \mathbb{E}_{Q\sim\mathcal{U}(0,1)}[Q]| = |\mathbb{E}_{Q\sim\mathcal{U}(0,1)}[Z - Q]| \tag{42}$$

Finally, applying Jensen's inequality ($|\mathbb{E}[A]| \leq \mathbb{E}[|A|]$ over $Q \sim \mathcal{U}(0,1)$) yields:

$$\begin{aligned} d_{W_1}(\mathbb{F}_Z, \mathbb{F}_\mathbb{U}) &\leq |\mathbb{E}_{Q\sim\mathcal{U}(0,1)}[Z - Q]| \\ &\leq \mathbb{E}_{Q\sim\mathcal{U}(0,1)}[|Z - Q|] = \mathbb{E}_{Q\sim\mathcal{U}(0,1)}[|F_\Phi(y|x, Q) - Q|]. \end{aligned} \tag{43}$$

**Importantly, inequality (39) holds even if $F_\Phi(y|x, \cdot)$ is not monotonic.** This is because the dual form of the 1-Wasserstein distance does not require any structural assumptions on the mapping from $Q$ to $Z = F_\Phi(y|x, Q)$. Therefore, inequality (39) remains a valid upper bound in the general case. **On**

**the role of inequality (39)**: Eq. (32) shows when inequality (39) becomes tight (*i.e.* $q \mapsto F_\Phi(y|x, q)$ is monotonically increasing).

Inequality (44) follows from the same reasoning as inequality (39), the application of Kantorovich–Rubinstein duality followed by Jensen's inequality, with an additional expectation over $Y \sim F_{Y|x}$:

$$\text{err}(\mathbf{x}) \leq \mathbb{E}_{Y \sim F_{Y|\mathbf{x}}, Q \sim \mathcal{U}(0,1)} \left[ |F_\Phi(Y|\mathbf{x}, Q) - Q| \right]. \tag{44}$$

**Contradiction argument.** Now suppose, for contradiction, that $F_\Phi$ is not $(\epsilon', \delta')$-PAIC. That is, by Definition 1, we have:

$$\Pr\left[ \text{err}(\mathbf{x}) > \epsilon' \right] > \delta'. \tag{45}$$

If we define the set:

$$\mathcal{S}_b := \left\{ \mathbf{x} \in \mathcal{X}, \mathbb{E}_{Y \sim F_{Y|\mathbf{x}}, Q \sim \mathcal{U}(0,1)} \left[ |F_\Phi(Y|\mathbf{x}, Q) - Q| \right] \geq \epsilon' \right\}, \tag{46}$$

and by the inequality (44), we can know that whenever $err(\mathbf{x}) \geq \epsilon'$ we have $\mathbf{x} \in \mathcal{S}_b$, thus we can conclude:

$$\Pr[\mathbf{X} \in S_b] > \delta'. \tag{47}$$

Then, for any $\epsilon < \epsilon'$ and $\mathbf{x} \in \mathcal{S}_b$, by bounding the expectation, we have:

$$\epsilon' \leq \mathbb{E}_{Y \sim F_{Y|\mathbf{x}}, Q \sim \mathcal{U}(0,1)}[|F_\Phi(Y|\mathbf{x}, Q) - Q|] \tag{48}$$

$$\leq \epsilon \cdot \Pr[|F_\Phi(Y|\mathbf{x}, Q) - Q| < \epsilon] + \Pr[|F_\Phi(Y|\mathbf{x}, Q) - Q| \geq \epsilon], \tag{49}$$

where inequality (49) holds because the absolute deviation term $|F_\Phi(Y|\mathbf{x}, Q) - Q| \in [0, 1]$. Now, letting $p = \Pr[|F_\Phi(Y|\mathbf{x}, Q) - Q| \geq \epsilon]$, we can solve:

$$\epsilon' \leq \epsilon(1 - p) + p \Rightarrow p \geq \frac{\epsilon' - \epsilon}{1 - \epsilon}. \tag{50}$$

Combining this with the bound over $\mathbf{x} \in \mathcal{S}_b$ (*i.e.*, inequality (47)) and applying the law of total probability, we can obtain:

$$\Pr\left[|F_\Phi(Y|\mathbf{x}, Q) - Q| \geq \epsilon\right] = \Pr\left[|F_\Phi(Y|\mathbf{x}, Q) - Q| \geq \epsilon | \mathbf{x} \in \mathcal{S}_b\right] \Pr\left[\mathbf{x} \in \mathcal{S}_b\right] \tag{51}$$

$$+ \Pr\left[|F_\Phi(Y|\mathbf{x}, Q) - Q| \geq \epsilon | \mathbf{x} \notin \mathcal{S}_b\right] \Pr\left[\mathbf{x} \notin \mathcal{S}_b\right] > \frac{\epsilon' - \epsilon}{1 - \epsilon} \cdot \delta'.$$

**Violation of MPAIC.** By Definition 2, $(\epsilon, \delta)$-MPAIC requires: $\Pr[|F_\Phi(Y|\mathbf{x}, Q) - Q| > \epsilon] < \delta$. Thus, equation (51) implies that $F_\Phi$ is not $(\epsilon, \delta' \cdot \frac{\epsilon' - \epsilon}{1 - \epsilon})$-MPAIC.

**Contrapositive and conclusion.** We have shown:

$$\text{Not } (\epsilon', \delta')\text{-PAIC} \Rightarrow \text{Not } (\epsilon, \delta' \cdot \tfrac{\epsilon' - \epsilon}{1 - \epsilon})\text{-MPAIC}, \quad \forall \epsilon' > \epsilon. \tag{52}$$

Taking the contrapositive:

$$(\epsilon, \delta)\text{-MPAIC} \Rightarrow (\epsilon', \delta \cdot \tfrac{1 - \epsilon}{\epsilon' - \epsilon})\text{-PAIC}, \quad \forall \epsilon' > \epsilon. \tag{53}$$

In the end, we can conclude that if $F_\Phi$ is not $(\epsilon', \delta')$-PAIC, then for any $\epsilon < \epsilon'$, it is not $\left(\epsilon, \delta' \cdot \frac{\epsilon' - \epsilon}{1 - \epsilon}\right)$-mPAIC, which is equivalent to the Theorem 1, *i.e.*, if $F_\Phi$ is $(\epsilon, \delta)$-mPAIC, then for any $\epsilon' > \epsilon$, it is also $\left(\epsilon', \delta \cdot \frac{1 - \epsilon}{\epsilon' - \epsilon}\right)$-PAIC. $\square$

Then, we will show the proof for Theorem 2.

**Theorem 2** (*Concentration* (Zhao et al., 2020)). *Let $F_\Phi$ be any $(\epsilon, \delta)$-MPAIC predictive CDF model, and let $(\mathbf{x}_1, y_1), \ldots, (\mathbf{x}_n, y_n) \overset{i.i.d.}{\sim} \mathbb{F}_{XY}$, and $q_1, \ldots, q_n \overset{i.i.d.}{\sim} \mathcal{U}(0, 1)$. Then, with probability at least $1 - \gamma$, we have:*

$$\frac{1}{n} \sum_{i=1}^{n} \mathbb{I}\left( |F_\Phi(y_i \mid \mathbf{x}_i, q_i) - q_i| \geq \epsilon \right) \leq \delta + \sqrt{\frac{-\log \gamma}{2n}}.$$

*Proof.* Define the sequence of Bernoulli random variables: $b_i = \mathbb{I}\left(|F_\Phi(y_i \mid \mathbf{x}_i, q_i) - q_i| \geq \epsilon\right) \in \{0, 1\}$. By the definition of $(\epsilon, \delta)$-*MPAIC*, we know that:

$$\Pr\left(b_i = 1\right) \leq \delta \quad \text{or equivalently} \quad \mathbb{E}[b_i] \leq \delta. \tag{54}$$

Now apply Hoeffding's inequality for bounded *i.i.d.* Bernoulli variables $b_i \in [0, 1]$:

$$\Pr\left(\frac{1}{n}\sum_{i=1}^{n} b_i \geq \delta + \epsilon'\right) \leq e^{-2\epsilon'^2 n}. \tag{55}$$

Set the RHS to $\gamma$, we solve for $\epsilon'$:

$$e^{-2\epsilon'^2 n} = \gamma \quad \Rightarrow \quad \epsilon' = \sqrt{\frac{-\log\gamma}{2n}}. \tag{56}$$

Therefore, with probability at least $1 - \gamma$, we have:

$$\frac{1}{n}\sum_{i=1}^{n} \mathbb{I}\left(|F_\Phi(y_i \mid \mathbf{x}_i, q_i) - q_i| \geq \epsilon\right) \leq \delta + \sqrt{\frac{-\log\gamma}{2n}}. \tag{57}$$

In summary, Theorem 2 reveals a key implication: increasing the number of quantile samples improves the quality of the Monte Carlo approximation, thus enhancing the *individual calibration* performance of the model. □

### A.5 EXTENDED DISCUSSION ABOUT THE MONOTONICITY CONSTRAINTS.

We clarify the role of monotonicity in our framework by answering the following four key questions.

*(1) Why is monotonicity needed?* Monotonicity is required for the first equality in Eq. (32) to hold, *i.e.*, for err($\mathbf{x}$) and err($\mathbf{x}, y$) to attain their maximal value, because the mapping $q \mapsto F_\Phi(y \mid \mathbf{x}, q)$ must be measure-preserving. Without monotonicity, this mapping may fail to be measure-preserving (leading, for example, to crossing quantiles), so the equality can break down. However, it is worth noting that inequality (39) and inequality (44) remain valid upper bounds (Full details can be found in Appendix A.4).

*(2) When can we ignore monotonicity?* During model training, we do not explicitly enforce monotonicity. Inequality (39) and inequality (44) guarantee that the training loss always serves as a valid upper bound on the true calibration error. For nonmonotone mappings, the actual calibration error is even smaller than this bound. Thus, ignoring monotonicity during optimization does not compromise the validity of the learning objective but instead leads to a conservative estimate.

*(3) When can we not ignore monotonicity?* It is important to ensure monotonicity when directly evaluating individual calibration (as in Definition 1 and Definition 2). This ensures that the calculated calibration error accurately reflects the theoretical definition, enabling a fair model comparison.

*(4) What is the solution for nonmonotonic?* Any continuous but nonmonotonic function can be transformed into a monotonic one by applying monotone rearrangement. In practice, we draw a finite number of quantile samples $q_1, \ldots, q_K \sim \mathcal{U}(0, 1)$, compute $f_i = F_\Phi(y \mid \mathbf{x}, q_i)$, and sort these values into a nondecreasing sequence $f_1, \ldots, f_K$, which are then reassigned to $q_1, \ldots, q_K$. This transformation preserves the distributional meaning while ensuring monotonicity, enabling rigorous evaluation of calibration.

### A.6 RELATED WORK

**Survival Models** Classical survival models can be categorized in general into *parametric*, *semiparametric*, and *nonparametric* approaches, depending on the assumptions they make about the underlying time-to-event distribution. *Parametric models* assume that event times follow a certain probability distribution, such as exponential (Feigl & Zelen, 1965), Weibull (Scholz & Works, 1996), log normal (Royston, 2001), Asymmetric Laplace (Kotz et al., 2012), or their mixtures (Nagpal et al., 2021). These methods typically characterize event times using the conditional

probability density function $f(t|\mathbf{x})$ and the corresponding cumulative distribution function $F(t|\mathbf{x})$. *Semi-parametric models*, most notably the Cox proportional hazards model (Cox, 1972), decompose the hazard function into a time-dependent baseline component and a covariate-dependent component, *i.e.*, $h(t|\mathbf{x}) = h_0(t) \exp(\mathbf{x}^\top \beta)$. The baseline hazard $h_0(t)$ is left unspecified and it is estimated nonparametrically, while the covariate effects $\beta$ are modeled parametrically. More recently, neural extensions such as DeepSurv (Katzman et al., 2018) have improved the scalability and expressiveness of Cox models, particularly in high-dimensional settings. *Nonparametric models* avoid explicit distributional assumptions, instead relying on data-driven estimators. Examples include Random Survival Forests (RSF) (Ishwaran et al., 2008), Gradient Boosting Machines (GBM) (Dembek et al., 2014), discrete-time models using categorical likelihoods (*e.g.*, DeepHit (Lee et al., 2018)), quantile regression (*e.g.*, CQRNN (Pearce et al., 2022)), or generative modeling (Chapfuwa et al., 2018).

**Calibration** The notion of calibration has a long history in statistics, with early definitions of *average calibration* (*e.g.*, the Brier score) (Brier, 1950; Murphy, 1973; Dawid, 1984). More recent interest in recalibrating classifiers has surged, especially for deep neural networks (Guo et al., 2017; Lakshminarayanan et al., 2017) Beyond *average calibration*, *group calibration* has been studied for both predefined (Kleinberg et al., 2017) and computationally defined groups (Kearns et al., 2018; Hébert-Johnson et al., 2018). In the context of survival analysis, recent *post hoc* calibration methods (*e.g.*, CSD (Qi et al., 2024a) and CiPOT (Qi et al., 2024b)) have been proposed to improve the average calibration of predicted survival functions by applying conformal prediction techniques and adjusting the estimated curves both *in probability* and *in time*. *Individual calibration*, which assesses the accuracy of predicted risks at the level of each instance, has been explored in fairness-aware learning (Sharifi-Malvajerdi et al., 2019), however, it remains both computationally and statistically challenging. For example, (Foygel Barber et al., 2021) showed that achieving perfect individual calibration with tight confidence intervals is subject to fundamental lower bounds. Moreover, recent work (Zhao et al., 2020) aimed to approximate individual calibration using randomized forecasting, and provided theoretical guarantees under certain assumptions on the model class and the data distribution.

# B EXPERIMENTAL DETAILS

This section provides additional information about the experimental setup. All experiments were implemented using the PyTorch framework. Detailed descriptions of the datasets, evaluation metrics, our method and baseline models, and implementation specifics are provided in Appendix B.1, Appendix B.2, and Appendix B.3, respectively.

**Hardware.** All experiments were conducted on a MacBook Pro equipped with an Apple M3 Pro chip, featuring 12 cores (6 performance and 6 efficiency cores) and 18 GB of memory. All computations were performed on the CPU, as the models predominantly utilized fully connected neural network architectures that did not require GPU acceleration.

## B.1 DATASETS

Our datasets are designed following the settings outlined in Pearce et al. (2022). The first category consists of *synthetic event data with synthetic censoring*. In these datasets, the input features $\mathbf{x}$ are generated uniformly as $\mathbf{x} \sim \mathcal{U}(0, 2)^D$, where $D$ denotes the number of features. The event time $e \sim p(e \mid \mathbf{x})$ and censoring time $c \sim p(c \mid \mathbf{x})$ are drawn from distinct parameterized distributions, with the specific forms of these distributions varying across different dataset configurations. Table 4 summarizes the distributional details of the event and censoring mechanisms.

The other type of dataset comprises *real event data with real censoring*, sourced from various domains and characterized by distinct features, sample sizes, and censoring proportions:

**METABRIC (Molecular Taxonomy of Breast Cancer International Consortium):** Contains genomic and clinical data for breast cancer patients. Includes 9 features, 1523 training samples, and 381 testing samples, with a censoring proportion of 0.42. Retrieved from the DeepSurv Repository.

**WHAS (Worcester Heart Attack Study):** Focuses on predicting survival following acute myocardial infarction. Includes 6 features, 1310 training samples, and 328 testing samples, with a censoring proportion of 0.57. Retrieved from the DeepSurv Repository.

Table 4: Summary of dataset statistics, including the number of features (Feats), training and test set sizes, proportion of censored events (PropCens), and the distributions used for sampling event and censoring times. The coefficient vector $\boldsymbol{\beta}$ is $[0.8, 0.6, 0.4, 0.5, -0.3, 0.2, 0.0, -0.7]$.

| Dataset | Feats | Train size | Test size | PropCens | Variables for event time | Variables for censoring time |
|---------|-------|-----------|-----------|----------|--------------------------|------------------------------|
| *Type 1: Synthetic event data with synthetic censoring* | | | | | | |
| Norm linear | 1 | 500 | 1000 | 0.20 | $\mathcal{N}(2\mathbf{x} + 10, (\mathbf{x} + 1)^2)$ | $\mathcal{N}(4\mathbf{x} + 10, (0.8\mathbf{x} + 0.4)^2)$ |
| Norm nonlinear | 1 | 500 | 1000 | 0.24 | $\mathcal{N}(\mathbf{x}\sin(2\mathbf{x}) + 10, (0.5\mathbf{x} + 0.5)^2)$ | $\mathcal{N}(2\mathbf{x} + 10, 2^2)$ |
| Exponential | 1 | 500 | 1000 | 0.30 | $\mathrm{Exp}(2\mathbf{x} + 4)$ | $\mathrm{Exp}(-3\mathbf{x} + 15)$ |
| Weibull | 1 | 500 | 1000 | 0.22 | $\mathrm{Weibull}(4\mathbf{x}\sin(2(\mathbf{x} - 1)) + 10, 5)$ | $\mathrm{Weibull}(-3\mathbf{x} + 20, 5)$ |
| LogNorm | 1 | 500 | 1000 | 0.21 | $\mathrm{LogNorm}(\mathbf{x} - 1)^2, \mathbf{x}^2$ | $\mathcal{U}(0, 10)$ |
| Norm uniform | 1 | 500 | 1000 | 0.62 | $\mathcal{N}(2\mathbf{x}\cos(2\mathbf{x}) + 13, (\mathbf{x} + 0.5)^2)$ | $\mathcal{U}(0, 18)$ |
| Norm heavy | 4 | 2000 | 1000 | 0.80 | $\mathcal{N}(3\mathbf{x}_0 + \mathbf{x}_1^2 - \mathbf{x}_2^2 + 2\sin(\mathbf{x}_2\mathbf{x}_3) + 6, (\mathbf{x} + 0.5)^2)$ | $\mathcal{U}(0, 12)$ |
| Norm med. | 4 | 2000 | 1000 | 0.49 | — | $\mathcal{U}(0, 12)$ |
| Norm light | 4 | 2000 | 1000 | 0.25 | — | $\mathcal{U}(0, 20)$ |
| Norm same | 4 | 2000 | 1000 | 0.50 | — | Equal to target |
| LogNorm heavy | 8 | 4000 | 1000 | 0.75 | $\mathrm{LogNorm}(\sum_i \boldsymbol{\beta}_i\mathbf{x}_i, 1)/10$ | $\mathcal{U}(0, 0.4)$ |
| LogNorm med. | 8 | 4000 | 1000 | 0.52 | — | $\mathcal{U}(0, 1.0)$ |
| LogNorm light | 8 | 4000 | 1000 | 0.23 | — | $\mathcal{U}(0, 3.5)$ |
| LogNorm same | 8 | 4000 | 1000 | 0.50 | — | Equal to target |
| *Type 2: Real event data with real censoring* | | | | | | |
| METABRIC | 9 | 1523 | 381 | 0.42 | Real | Real |
| WHAS | 6 | 1310 | 328 | 0.57 | Real | Real |
| SUPPORT | 14 | 7098 | 1775 | 0.32 | Real | Real |
| GBSG | 7 | 1785 | 447 | 0.42 | Real | Real |
| TMBImmuno | 3 | 1328 | 332 | 0.49 | Real | Real |
| BreastMSK | 5 | 1467 | 367 | 0.77 | Real | Real |
| LGGGBM | 5 | 510 | 128 | 0.60 | Real | Real |

**SUPPORT (Study to Understand Prognoses Preferences Outcomes and Risks of Treatment):** Provides survival data for critically ill hospitalized patients. Includes 14 features, 7098 training samples, and 1775 testing samples, with a censoring proportion of 0.32. Covariates include demographic information and basic diagnostic data. Retrieved from the DeepSurv Repository.

**GBSG (German Breast Cancer Study Group):** Tracks survival outcomes of breast cancer patients. Includes 7 features, 1785 training samples, and 447 testing samples, with a censoring proportion of 0.42. Retrieved from the DeepSurv Repository.

**TMBImmuno (Tumor Mutational Burden and Immunotherapy):** Predicts survival time for patients with various cancer types using clinical data. Includes 3 features, 1328 training samples, and 332 testing samples, with a censoring proportion of 0.49. Covariates include age, sex, and mutation count. Retrieved from cBioPortal.

**BreastMSK:** Derived from the Memorial Sloan Kettering Cancer Center, this dataset focuses on survival prediction for breast cancer patients using tumor-related information. Includes 5 features, 1467 training samples, and 367 testing samples, with a censoring proportion of 0.77. Retrieved from cBioPortal.

**LGGGBM:** Integrates survival data from low-grade glioma (LGG) and glioblastoma multiforme (GBM), often used for validating models in cancer genomics. Includes 5 features, 510 training samples, and 128 testing samples, with a censoring proportion of 0.60. Retrieved from cBioPortal.

## B.2 METRICS

We evaluate each model using three categories of metrics: *Predictive Accuracy*, *Concordance*, and *Calibration*. For *predictive accuracy*, we report the Mean Absolute Error (MAE) and the Integrated Brier Score (IBS) (Graf et al., 1999), which quantify the accuracy of survival time predictions over time. For *concordance*, we use Harrell's C-Index (Harrell et al., 1982) and Uno's C-Index (Uno et al., 2011) to evaluate the model's ability to correctly rank survival times while accounting for censored observations. For *calibration*, we assess the reliability of survival probability estimates using Expected Calibration Error (ECE) (Naeini et al., 2015) for both *average* and *group calibration*, and the average Wasserstein Distance (Villani, 2009) between predicted and empirical survival distributions to evaluate *individual calibration*. These metrics provide a holistic evaluation framework that effectively captures the *predictive accuracy*, *discriminative* ability, and *calibration* quality of survival models.

**Mean Absolute Error (MAE):**

$$\text{MAE} = \frac{1}{N} \sum_{i=1}^{N} |y_i - \tilde{y}_i|, \tag{58}$$

where $y_i$ is the ground-truth event time, $\tilde{y}_i$ is the model's predicted survival time (*e.g.*, the median value of the estimated survival CDF, *i.e.*, $F_\Phi^{-1}(q = 0.5|\mathbf{x})$), and $N$ is the number of test samples.

**Integrated Brier Score (IBS):**

$$\text{BS}(t) = \frac{1}{N} \sum_{i=1}^{N} \left[ \frac{(1 - \tilde{F}_\Phi(t \mid \mathbf{x}_i))^2 \, \mathbb{I}(y_i \leq t, e_i = 1)}{\tilde{G}(y_i)} + \frac{\tilde{F}_\Phi(t \mid \mathbf{x}_i)^2 \, \mathbb{I}(y_i > t)}{\tilde{G}(t)} \right], \tag{59}$$

$$\text{IBS} = \frac{1}{t_2 - t_1} \int_{t_1}^{t_2} \text{BS}(t) \, dt, \tag{60}$$

where $\tilde{F}_\Phi(t \mid \mathbf{x}_i)$ denotes the predicted cumulative distribution function (CDF) at time $t$, $\tilde{G}(\cdot)$ is the Kaplan–Meier estimator (Kaplan & Meier, 1958) of the censoring distribution, and 100 time points in the integration range $[t_1, t_2]$ are evenly selected from the 0.1 to 0.9 quantiles of the training set's $y$-distribution.

**Harrell's C-Index:**

$$\text{C}_\text{H} = \frac{\sum_{i \neq j} [\mathbb{I}(\phi_i > \phi_j) + 0.5 \cdot \mathbb{I}(\phi_i = \phi_j)] \cdot \mathbb{I}(y_i < y_j)\delta_i}{\sum_{i \neq j} \mathbb{I}(y_i < y_j)\delta_i}, \tag{61}$$

where $\phi_i = \tilde{S}(y_i \mid \mathbf{x}_i) = 1 - \tilde{F}_\Phi(y_i \mid \mathbf{x}_i)$ is the model's risk score. For implementation, we utilize the `concordance_index_censored` function from the `sksurv.metrics` module, as documented in the scikit-survival API.

**Uno's C-Index:**

$$\text{C}_\text{U} = \frac{\sum_{i=1}^{N} \sum_{j=1}^{N} \tilde{G}(y_i)^{-2} [\mathbb{I}(\phi_i > \phi_j) + 0.5 \cdot \mathbb{I}(\phi_i = \phi_j)] \cdot \mathbb{I}(y_i < y_j, y_i < y_\tau)\delta_i}{\sum_{i=1}^{N} \sum_{j=1}^{N} \tilde{G}(y_i)^{-2} \cdot \mathbb{I}(y_i < y_j, y_i < y_\tau)\delta_i}, \tag{62}$$

where $y_\tau$ is the cutoff value for the survival time. For implementation, we use the `concordance_index_ipcw` function from the `sksurv.metrics` module, as documented in the scikit-survival API.

**Average Calibration:** To evaluate *average calibration*, we assess whether the model's predicted cumulative probabilities align with the ideal uniform distribution $\mathcal{U}(0, 1)$. Specifically, the *Expected Calibration Error* (ECE) of a predictive CDF model $F_\Phi$ is defined as the 1-Wasserstein distance between the empirical distribution of the predicted CDF values and the uniform distribution:

$$\text{ECE}(F_\Phi) = \int_0^1 |\Pr[F_\Phi(Y \mid X) \leq q] - q| \, dq = d_{W_1}\left(\mathbb{F}_{F_\Phi(Y|X)}, \mathbb{F}_\mathcal{U}\right), \tag{63}$$

where $Y \sim F_{Y|\mathbf{x}}$, $q \sim \mathcal{U}(0, 1)$, $\mathbb{F}_{F_\Phi(Y|X)}$ denotes the empirical CDF of the predicted cumulative probabilities, and $\mathbb{F}_\mathcal{U}$ denotes the ideal uniform CDF. In practice, we compute $F_\Phi(e_i \mid \mathbf{x}_i)$ for each test instance $(\mathbf{x}_i, y_i)$, where $e_i$ is the true event time. Note that $e_i$ is only observable in synthetic datasets, where the true generative distribution is known.

Predicted CDF values $\tilde{F}_\Phi(e_i \mid \mathbf{x}_i)$ are then sorted to form the empirical distribution, which is compared to uniformly spaced quantile targets $\{q_j\}_{j=1}^{N} \sim \mathcal{U}(0, 1)$. The calibration error is calculated as:

$$\text{ECE} = \frac{1}{N} \sum_{i=1}^{N} \left| \tilde{F}_\Phi(e_i \mid \mathbf{x}_i) - \frac{i}{N} \right|. \tag{64}$$

For real-world datasets with censoring, we replace $e_i$ with the observed time $y_i$, and compute ECE only on uncensored samples ($\delta_i = 1$), resulting in:

$$\text{ECE} = \frac{1}{N_{\text{obs}}} \sum_{i=1}^{N_{\text{obs}}} \left| \tilde{F}_\Phi(y_i \mid \mathbf{x}_i) - \frac{i}{N_{\text{obs}}} \right|, \tag{65}$$

where $N_{\text{obs}}$ is the number of uncensored test samples.

**Group Calibration:** To evaluate *group calibration*, we partition the input space $\mathcal{X}$ into structured subsets (*i.e.*, $\mathcal{S} = \{\mathcal{S}_k\}_{k=1}^K \subset \mathcal{X}$) using combinations of feature dimensions. Let the input $\mathbf{x} \in \mathbb{R}^n$. We define $\binom{n}{2}$ feature pairs, and for each pair $\{x_i, x_j\}$, we form 4 subgroups based on median thresholding:

- (1) $x_i > \text{Median}(x_i)$ and $x_j > \text{Median}(x_j)$
- (2) $x_i > \text{Median}(x_i)$ and $x_j \leq \text{Median}(x_j)$
- (3) $x_i \leq \text{Median}(x_i)$ and $x_j > \text{Median}(x_j)$
- (4) $x_i \leq \text{Median}(x_i)$ and $x_j \leq \text{Median}(x_j)$

This results in $K = 4 \times \binom{n}{2}$ groups in total. Within each group, we compute the ECE as in the average case.

$$\text{ECE}_s = \int_0^1 \left| \Pr\left[ F_\Phi(Y \mid \mathbf{x} \in \mathcal{S}_k) \leq q \right] - q \right| dq. \tag{66}$$

Also, to ensure statistical stability, each group must contain between $\frac{1}{4} \cdot$ size and $\frac{3}{4} \cdot$ size of the full dataset. We then define the group calibration error as the worst (*i.e.*, largest) ECE across all valid groups:

$$\text{GroupECE} = \max_{s \in \mathcal{S}} \text{ECE}_s. \tag{67}$$

**Individual Calibration:** To evaluate *individual calibration*, we compare the predicted cumulative distribution $\tilde{F}_\Phi(y \mid \mathbf{x})$ with the ground-truth CDF $F^*(y \mid \mathbf{x})$ for each individual test input. The discrepancy between these two distributions is measured using the 1-Wasserstein distance:

$$d_{W_1}(\tilde{F}_\Phi, F^*) = \int_0^{1.2 \times y_{\max}} \left| \tilde{F}_\Phi(t \mid \mathbf{x}_i) - F^*(t \mid \mathbf{x}_i) \right| dt, \tag{68}$$

where $\tilde{F}_\Phi(\cdot \mid \mathbf{x}_i)$ is the estimated CDF produced by the model and $F^*(\cdot \mid \mathbf{x}_i)$ is the oracle CDF (ground truth) corresponding to the same input. In practice, we approximate this integral using a discrete grid of 1000 evenly spaced time points $\{t_j\}_{j=1}^{1000} \in [0, 1.2 \times y_{\max}]$, where $y_{\max}$ denotes the maximum observed event time in the test set. Since ground truth distributions are only accessible for synthetic datasets, individual calibration can only be evaluated in synthetic settings, where $F^*(t \mid \mathbf{x}_i)$ is analytically known for each test input.

### B.3 ICALD AND BASELINES

To comprehensively assess the performance of the proposed method (ICALD), we compare it to 11 strong baseline models, summarized in Table 5. These baselines span a spectrum of survival modeling paradigms, including *parametric*, *semi-parametric*, *nonparametric*, and *post-calibration* approaches:

**Parametric models:** ALD (Sheng & Henao, 2025) and LogNorm (Royston, 2001) assume fixed parametric distributions for event times. DSM (Nagpal et al., 2021) extends this to a mixture of parametric families such as Weibull and LogNormal.

**Semi-parametric model:** DeepSurv (Katzman et al., 2018) is a neural extension of the Cox proportional hazards model (Fox & Weisberg, 2002), allowing for non-linear feature representations while maintaining proportional hazard assumptions.

**Nonparametric models:** CQRNN (Pearce et al., 2022) directly estimates quantiles using the *pinball loss* under censoring, while DeepHit (Lee et al., 2018) estimates the full discrete-time survival function via *log-likelihood* and *ranking losses*. Tree-based ensemble models such as GBM (Dembek et al., 2014) and RSF (Ishwaran et al., 2008) are also included, offering *non-neural* alternatives that model complex interactions.

**Pre-calibration methods:** X-CAL (Goldstein et al., 2020) introduces an explicit calibration objective for survival analysis by reformulating the distributional calibration (D-Calibration) metric (Haider et al., 2020) into a differentiable loss, allowing calibration to be optimized jointly with predictive accuracy during model training.

**Post-calibration methods:** CSD (Qi et al., 2024a) and CiPOT (Qi et al., 2024b) are representative *post-calibration* strategies applied after model training to improve alignment between predicted and true distributions.

All neural baselines were trained using either the same network architecture as our method or the default architecture provided by their official repositories, under a consistent optimization protocol to ensure fair comparison. Specifically, the implementations for CQRNN and Log-Norm were adopted from the official CQRNN repository[2], while DeepSurv and DeepHit were adapted from the `pycox.methods` module[3]. For the mixture-based baseline, we employed the Deep Survival Machines (DSM) model from the `auton-survival` library[4], implemented via `auton_survival.models.dsm.DeepSurvivalMachines`. For ensemble-based baselines, we used the official implementations from the `sksurv` library, namely `RandomSurvivalForest` and `GradientBoostingSurvivalAnalysis`, both available in the `ensemble` module[5]. For the *pre-calibration* baseline, we used the official implementation of X-CAL,[6] which introduces an explicit calibration loss for survival analysis. Finally, both the CSD and CiPOT *post-calibration* methods were re-implemented based on their official repository[7].

Table 5: Summary of baselines used for comparison.

| Method | Type | Neural | Description |
|---|---|---|---|
| ALD (Sheng & Henao, 2025) | *Parametric* | ✓ | Assumes event times follow a Asymmetric Laplace distribution (Kotz et al., 2012) |
| LogNorm (Hoseini et al., 2017) | *Parametric* | ✓ | Assumes event times follow a LogNorm (Royston, 2001) distribution |
| DSM (Nagpal et al., 2021) | *Parametric (Mixture)* | ✓ | Mixture of parametric distributions (*e.g.*, LogNorm (Royston, 2001), Weibull (Scholz & Works, 1996)) |
| DeepSurv (Katzman et al., 2018) | *Semi-parametric* | ✓ | Neural extension of Cox proportional hazards model (Fox & Weisberg, 2002) |
| CQRNN (Pearce et al., 2022) | *Non-parametric* | ✓ | Neural censored quantile regression using the *pinball* loss |
| DeepHit (Lee et al., 2018) | *Non-parametric* | ✓ | Predicts survival functions via *log-likelihood* and *ranking* losses |
| GBM (Dembek et al., 2014) | *Non-parametric (Ensemble)* | ✗ | Generalized Boosted Model adapted for survival tasks |
| RSF (Ishwaran et al., 2008) | *Non-parametric (Ensemble)* | ✗ | Random Forests adapted for survival tasks |
| X-CAL (Goldstein et al., 2020) | *Pre-calibration* | ✓ | *Post-calibration* method applied when survival model training |
| CSD (Qi et al., 2024a) | *Post-calibration* | ✗ | *Post-calibration* method applied after survival model training |
| CiPOT (Qi et al., 2024b) | *Post-calibration* | ✗ | *Post-calibration* method applied after survival model training |

**Hyperparameter default settings.** All experiments were repeated across 10 random seeds to ensure robust and reliable results. The hyperparameter settings were as follows:

- **Default Neural Network Architecture:** Fully-connected network with two hidden layers, each consisting of 100 hidden nodes, using ReLU activations.
- **Default Epochs:** 200
- **Default Batch Size:** 128
- **Default Learning Rate:** 0.01
- **Dropout Rate:** 0.1
- **Optimizer:** Adam

---

[2] https://github.com/TeaPearce/Censored_Quantile_Regression_NN
[3] https://github.com/havakv/pycox
[4] https://autonlab.org/auton-survival/models/dsm/index.html
[5] https://scikit-survival.readthedocs.io/en/stable/api/ensemble.html
[6] https://github.com/rajesh-lab/X-CAL
[7] https://github.com/shi-ang/MakeSurvivalCalibratedAgain

- **Batch Norm:** FALSE

**ALD and ICALD (Trained with $\mathcal{L}_{\text{ALD+Cal}}^{\text{Pre}}$, $\mathcal{L}_{\text{ALD+Cqr}}^{\text{Pre}}$, $\mathcal{L}_{\text{ALD+Cal}}^{\text{Post}}$, and $\mathcal{L}_{\text{ALD+Cqr}}^{\text{Post}}$).**

The model architecture for our method (*pre-calibrated* with $\mathcal{L}_{\text{ALD+Cal}}^{\text{Pre}}$ and $\mathcal{L}_{\text{ALD+Cqr}}^{\text{Pre}}$) is illustrated in Fig. 1. Similar to our *pre-calibration* method, we employ a residual connection between the shared feature extractor and the first hidden layer to improve gradient flow and training stability for the base ALD model. Each hidden layer consists of 32 neurons with ReLU activation. To enforce positivity constraints on the ALD parameters, exponential activations are applied to the output heads corresponding to $\theta$, $\sigma$, and $\kappa$. To mitigate overfitting, we randomly hold out 20% of the training set as a validation set and apply early stopping (default epochs for our method is set to 2000) based on validation loss. The key distinction between our *pre-calibration* methods and the original ALD model lies in an additional *adapter module* that incorporates the quantile level $q \sim \mathcal{U}(0, 1)$ as an input. This allows the model to learn quantile-conditioned ALD parameters $\{\theta_q^*, \sigma_q^*, \kappa_q^*\}$, which are crucial for both *quantile regression* and *calibration* losses (see Equation equation 10 and Equation equation 12).

In the *post-calibration* setting (i.e., $\mathcal{L}_{\text{ALD+Cal}}^{\text{Post}}$ and $\mathcal{L}_{\text{ALD+Cqr}}^{\text{Post}}$), we first obtain the ALD parameters from a pre-trained base ALD model. Then, we apply a lightweight *post-calibration* network that takes both the input $\mathbf{x}$ and quantile $q$ using $\mathcal{L}_{\text{ALD+Cal}}^{\text{Post}}$, and $\mathcal{L}_{\text{ALD+Cqr}}^{\text{Post}}$ to output adjustment factors $\gamma \in \mathbb{R}^3$. This *post-calibration* module is implemented as a compact MLP with two hidden layers of 16 units each and ReLU activation. An exponential activation is then applied at the output to ensure positivity of the adjustment factors. These adjustment factors modulate the base parameters to produce calibrated outputs via an element-wise product (see Equation equation 13). This design allows the *post-calibration* module to correct for miscalibration while preserving the base model's learned structure.

Finally, for prediction, we sample 2000 quantile percentages $q \sim \mathcal{U}(0, 1)$ to construct a mixture of ALD. Conceptually, this model can be interpreted as a continuous mixture over quantile-specific ALD components, expressed as:

$$\tilde{f}(y \mid \mathbf{x}) = \int_0^1 f_{\text{ALD}}\left(y; m_\Phi(\mathbf{x}, q)\right) \, dq, \tag{69}$$

where $f_{\text{ALD}}(\cdot; m_\Phi(\mathbf{x}, q))$ is the ALD parameterized by the *post-calibrated* model at quantile $q$. This formulation captures rich distributional information and enables fine-grained calibration by averaging over a wide spectrum of quantile-conditioned predictions.

**CQRNN.** We followed the hyperparameter settings tuned in the original paper (Pearce et al., 2022), where three random splits were used for validation (ensuring no overlap with the random seeds used in the final test runs). The following settings were applied:

- **Weight Decay:** 0.0001
- **Grid Size:** 100
- **Pseudo Value:** $y^* = 1.2 \times \max_i y_i$
- **Dropout Rate:** 0.333

The number of epochs and dropout usage were adjusted based on the dataset type:

- **Synthetic Datasets:**
  - **Norm linear, Norm non-linear, Exponential, Weibull, LogNorm, Norm uniform:** 100 epochs with dropout disabled.
  - **Norm heavy, Norm medium, Norm light, Norm same:** 20 epochs with dropout disabled.
  - **LogNorm heavy, LogNorm medium, LogNorm light, LogNorm same:** 10 epochs with dropout disabled.
- **Real-World Datasets:**
  - **METABRIC:** 20 epochs with dropout disabled.

- **WHAS:** 100 epochs with dropout disabled.
- **SUPPORT:** 10 epochs with dropout disabled.
- **GBSG:** 20 epochs with dropout enabled.
- **TMBImmuno:** 50 epochs with dropout disabled.
- **BreastMSK:** 100 epochs with dropout disabled.
- **LGGGBM:** 50 epochs with dropout enabled.

**LogNorm.** The output dimensions of the default neural network architecture are 2, where the two outputs represent the mean and standard deviation of a Log-Normal distribution. To ensure the standard deviation prediction is always positive and differentiable, the output representing the standard deviation is passed through a `SoftPlus` activation function. We followed the hyperparameter settings tuned in the original paper (Pearce et al., 2022), with a dropout rate of 0.333. The number of epochs and the usage of dropouts were adjusted according to the type of dataset as follows:

- **Synthetic Datasets:** The same settings as described above for **CQRNN**.

- **Real-World Datasets:**

    - **METABRIC:** 10 epochs with dropout disabled.
    - **WHAS:** 50 epochs with dropout disabled.
    - **SUPPORT:** 20 epochs with dropout disabled.
    - **GBSG:** 10 epochs with dropout enabled.
    - **TMBImmuno:** 50 epochs with dropout disabled.
    - **BreastMSK:** 50 epochs with dropout disabled.
    - **LGGGBM:** 20 epochs with dropout enabled.

**DeepSurv.** We adhered to the official hyperparameter settings from the `pycox.methods` module (GitHub Link). Each of the two hidden layers contains 32 hidden nodes. A validation set was created by splitting 20% of the training set. Early stopping was used to terminate training when validation performance stopped improving. Batch normalization was applied.

**DeepHit.** We adhered to the official hyperparameter settings from the `pycox.methods` module (GitHub Link). Each of the two hidden layers contains 32 hidden nodes. A validation set was created by splitting 20% of the training set. Early stopping was used to terminate training when validation performance stopped improving. Batch normalization was applied, with additional settings: `num_durations` = 100, `alpha` = 0.2, and `sigma` = 0.1.

**DSM.** We adopted the Deep Survival Machines (DSM) model from the `auton-survival` library[8] implemented via `auton_survival.models.dsm.DeepSurvivalMachines`. The model was configured with two hidden layers of 32 units each. For the LogNormal variant, the number of mixture components was set to $k = 10$, as increasing $k$ led to performance degradation. For the Weibull variant, we followed the default configuration with $k = 100$ to ensure sufficient capacity. The model was trained using observed event times and indicators, and the final prediction was constructed by evaluating the mixture distribution over a fixed 1000-point time grid to obtain the cumulative distribution function (CDF).

**GBM.** We used the `GradientBoostingSurvivalAnalysis` implementation from the `sksurv.ensemble` module.[9] The model was configured with `n_estimators` = 100, `learning_rate` = 0.01, and `max_depth` = 3.

**RSF.** For the Random Survival Forest, we used the `RandomSurvivalForest` class from `sksurv.ensemble`.[10] We followed the standard configuration with `n_estimators` = 100.

---

[8] https://autonlab.org/auton-survival/models/dsm/index.html
[9] https://scikit-survival.readthedocs.io/en/stable/api/ensemble.html
[10] https://scikit-survival.readthedocs.io/en/stable/api/ensemble.html

# C    ADDITIONAL RESULTS

This section presents additional results to provide a comprehensive evaluation. The full results for *pre-calibration*, *post-calibration*, and *general* performance are provided in Appendix C.1, Appendix C.2, and Appendix C.3, respectively. Case studies are provided in Appendix C.4.

## C.1    PRE-CALIBRATION RESULTS

Table 6 presents the full results for the *pre-calibration* setting. The best performance for each dataset and metric is highlighted in **bold**. Fig. 3 illustrates the best and worst *individual calibration* improvement cases with the *pre-calibration* setting, comparing $\mathcal{L}^{\text{Pre}}_{\text{ALD+Cal}}$ against $\mathcal{L}_{\text{ALD}}$, achieved by the hybrid ALD-based survival model across all synthetic datasets.

Table 6: Full results table on *pre-calibration* for all datasets, methods, and metrics. The values represent the mean ± 1 standard error for the test set over 5 runs.

| Dataset | Method | Average Calibration | Group Calibration | Individual Calibration |
|---|---|---|---|---|
| Norm_linear | ALD ($\mathcal{L}_{\text{ALD}}$) | 0.047 ± 0.006 | 0.079 ± 0.012 | 0.044 ± 0.005 |
| | CQRNN ($\mathcal{L}_{\text{Cqr}}$) | 0.035 ± 0.006 | 0.054 ± 0.005 | **0.018 ± 0.002** |
| | $\mathcal{L}^{\text{Pre}}_{\text{X-CAL}}$ | 0.050 ± 0.007 | 0.076 ± 0.010 | 0.043 ± 0.003 |
| | $\mathcal{L}^{\text{Pre}}_{\text{ALD+Cqr}}$ | 0.017 ± 0.003 | 0.036 ± 0.009 | **0.018 ± 0.002** |
| | $\mathcal{L}^{\text{Pre}}_{\text{ALD+Cal}}$ | **0.016 ± 0.002** | **0.029 ± 0.005** | **0.018 ± 0.002** |
| Norm_nonlinear | ALD ($\mathcal{L}_{\text{ALD}}$) | 0.072 ± 0.010 | 0.119 ± 0.012 | 0.060 ± 0.008 |
| | CQRNN ($\mathcal{L}_{\text{Cqr}}$) | 0.034 ± 0.011 | 0.078 ± 0.011 | 0.029 ± 0.002 |
| | $\mathcal{L}^{\text{Pre}}_{\text{X-CAL}}$ | 0.070 ± 0.007 | 0.112 ± 0.009 | 0.056 ± 0.005 |
| | $\mathcal{L}^{\text{Pre}}_{\text{ALD+Cqr}}$ | 0.034 ± 0.003 | 0.045 ± 0.006 | 0.018 ± 0.001 |
| | $\mathcal{L}^{\text{Pre}}_{\text{ALD+Cal}}$ | **0.023 ± 0.004** | **0.035 ± 0.004** | **0.012 ± 0.001** |
| Norm_uniform | ALD ($\mathcal{L}_{\text{ALD}}$) | 0.095 ± 0.009 | 0.159 ± 0.020 | 0.098 ± 0.020 |
| | CQRNN ($\mathcal{L}_{\text{Cqr}}$) | 0.036 ± 0.009 | 0.113 ± 0.029 | 0.054 ± 0.007 |
| | $\mathcal{L}^{\text{Pre}}_{\text{X-CAL}}$ | 0.078 ± 0.013 | 0.144 ± 0.022 | 0.085 ± 0.020 |
| | $\mathcal{L}^{\text{Pre}}_{\text{ALD+Cqr}}$ | 0.102 ± 0.004 | 0.141 ± 0.009 | 0.092 ± 0.004 |
| | $\mathcal{L}^{\text{Pre}}_{\text{ALD+Cal}}$ | **0.027 ± 0.004** | **0.038 ± 0.006** | **0.018 ± 0.002** |
| Exponential | ALD ($\mathcal{L}_{\text{ALD}}$) | 0.018 ± 0.011 | 0.030 ± 0.014 | 0.016 ± 0.003 |
| | CQRNN ($\mathcal{L}_{\text{Cqr}}$) | 0.030 ± 0.008 | 0.051 ± 0.011 | 0.030 ± 0.003 |
| | $\mathcal{L}^{\text{Pre}}_{\text{X-CAL}}$ | 0.017 ± 0.009 | 0.029 ± 0.011 | **0.013 ± 0.004** |
| | $\mathcal{L}^{\text{Pre}}_{\text{ALD+Cqr}}$ | 0.034 ± 0.007 | 0.041 ± 0.009 | 0.023 ± 0.006 |
| | $\mathcal{L}^{\text{Pre}}_{\text{ALD+Cal}}$ | **0.012 ± 0.005** | **0.020 ± 0.006** | 0.015 ± 0.003 |
| Weibull | ALD ($\mathcal{L}_{\text{ALD}}$) | 0.048 ± 0.009 | 0.067 ± 0.006 | 0.042 ± 0.003 |
| | CQRNN ($\mathcal{L}_{\text{Cqr}}$) | 0.031 ± 0.004 | 0.086 ± 0.021 | 0.040 ± 0.010 |
| | $\mathcal{L}^{\text{Pre}}_{\text{X-CAL}}$ | 0.045 ± 0.008 | 0.061 ± 0.005 | 0.039 ± 0.003 |
| | $\mathcal{L}^{\text{Pre}}_{\text{ALD+Cqr}}$ | 0.031 ± 0.007 | 0.039 ± 0.008 | 0.027 ± 0.002 |
| | $\mathcal{L}^{\text{Pre}}_{\text{ALD+Cal}}$ | **0.019 ± 0.005** | **0.032 ± 0.002** | **0.020 ± 0.002** |
| LogNorm | ALD ($\mathcal{L}_{\text{ALD}}$) | 0.020 ± 0.006 | 0.031 ± 0.012 | **0.128 ± 0.006** |
| | CQRNN ($\mathcal{L}_{\text{Cqr}}$) | 0.031 ± 0.013 | 0.050 ± 0.014 | 0.135 ± 0.008 |
| | $\mathcal{L}^{\text{Pre}}_{\text{X-CAL}}$ | 0.020 ± 0.003 | 0.030 ± 0.010 | 0.130 ± 0.005 |
| | $\mathcal{L}^{\text{Pre}}_{\text{ALD+Cqr}}$ | 0.052 ± 0.005 | 0.058 ± 0.006 | 0.140 ± 0.002 |
| | $\mathcal{L}^{\text{Pre}}_{\text{ALD+Cal}}$ | **0.015 ± 0.003** | **0.021 ± 0.002** | 0.131 ± 0.003 |
| Norm_heavy | ALD ($\mathcal{L}_{\text{ALD}}$) | **0.062 ± 0.009** | 0.113 ± 0.033 | 0.048 ± 0.006 |
| | CQRNN ($\mathcal{L}_{\text{Cqr}}$) | 0.071 ± 0.020 | 0.157 ± 0.023 | **0.032 ± 0.003** |
| | $\mathcal{L}^{\text{Pre}}_{\text{X-CAL}}$ | 0.068 ± 0.015 | 0.118 ± 0.032 | 0.046 ± 0.003 |
| | $\mathcal{L}^{\text{Pre}}_{\text{ALD+Cqr}}$ | 0.132 ± 0.027 | 0.259 ± 0.033 | 0.110 ± 0.005 |
| | $\mathcal{L}^{\text{Pre}}_{\text{ALD+Cal}}$ | 0.063 ± 0.010 | **0.109 ± 0.017** | 0.038 ± 0.003 |
| Norm med. | ALD ($\mathcal{L}_{\text{ALD}}$) | 0.054 ± 0.031 | 0.086 ± 0.026 | 0.028 ± 0.004 |
| | CQRNN ($\mathcal{L}_{\text{Cqr}}$) | 0.050 ± 0.015 | 0.093 ± 0.013 | **0.019 ± 0.001** |
| | $\mathcal{L}^{\text{Pre}}_{\text{X-CAL}}$ | **0.044 ± 0.012** | 0.079 ± 0.009 | 0.025 ± 0.003 |
| | $\mathcal{L}^{\text{Pre}}_{\text{ALD+Cqr}}$ | 0.085 ± 0.003 | 0.111 ± 0.006 | 0.071 ± 0.004 |
| | $\mathcal{L}^{\text{Pre}}_{\text{ALD+Cal}}$ | **0.044 ± 0.004** | **0.076 ± 0.006** | 0.020 ± 0.001 |

| Dataset | Method | Average Calibration | Group Calibration | Individual Calibration |
|---|---|---|---|---|
| Norm light | ALD ($\mathcal{L}_{\text{ALD}}$) | $0.077 \pm 0.034$ | $0.111 \pm 0.027$ | $0.027 \pm 0.005$ |
| | CQRNN ($\mathcal{L}_{\text{Cqr}}$) | $0.036 \pm 0.018$ | $0.083 \pm 0.006$ | $\mathbf{0.015 \pm 0.002}$ |
| | $\mathcal{L}_{\text{X-CAL}}^{\text{Pre}}$ | $0.055 \pm 0.013$ | $0.097 \pm 0.003$ | $0.023 \pm 0.002$ |
| | $\mathcal{L}_{\text{ALD+Cqr}}^{\text{Pre}}$ | $0.048 \pm 0.004$ | $0.068 \pm 0.007$ | $0.037 \pm 0.002$ |
| | $\mathcal{L}_{\text{ALD+Cal}}^{\text{Pre}}$ | $\mathbf{0.032 \pm 0.006}$ | $\mathbf{0.059 \pm 0.005}$ | $0.016 \pm 0.001$ |
| Norm same | ALD ($\mathcal{L}_{\text{ALD}}$) | $0.065 \pm 0.012$ | $0.090 \pm 0.017$ | $0.044 \pm 0.008$ |
| | CQRNN ($\mathcal{L}_{\text{Cqr}}$) | $0.037 \pm 0.008$ | $0.075 \pm 0.008$ | $\mathbf{0.022 \pm 0.004}$ |
| | $\mathcal{L}_{\text{X-CAL}}^{\text{Pre}}$ | $0.062 \pm 0.008$ | $0.088 \pm 0.012$ | $0.043 \pm 0.005$ |
| | $\mathcal{L}_{\text{ALD+Cqr}}^{\text{Pre}}$ | $0.029 \pm 0.008$ | $0.067 \pm 0.014$ | $0.026 \pm 0.003$ |
| | $\mathcal{L}_{\text{ALD+Cal}}^{\text{Pre}}$ | $\mathbf{0.025 \pm 0.003}$ | $\mathbf{0.053 \pm 0.009}$ | $0.023 \pm 0.002$ |
| LogNorm heavy | ALD ($\mathcal{L}_{\text{ALD}}$) | $0.037 \pm 0.029$ | $0.081 \pm 0.043$ | $\mathbf{0.038 \pm 0.008}$ |
| | CQRNN ($\mathcal{L}_{\text{Cqr}}$) | $0.174 \pm 0.008$ | $0.294 \pm 0.014$ | $0.113 \pm 0.011$ |
| | $\mathcal{L}_{\text{X-CAL}}^{\text{Pre}}$ | $0.027 \pm 0.013$ | $0.070 \pm 0.022$ | $0.034 \pm 0.002$ |
| | $\mathcal{L}_{\text{ALD+Cqr}}^{\text{Pre}}$ | $\mathbf{0.024 \pm 0.004}$ | $\mathbf{0.061 \pm 0.006}$ | $0.040 \pm 0.003$ |
| | $\mathcal{L}_{\text{ALD+Cal}}^{\text{Pre}}$ | $0.025 \pm 0.005$ | $0.072 \pm 0.009$ | $0.039 \pm 0.005$ |
| LogNorm med. | ALD ($\mathcal{L}_{\text{ALD}}$) | $0.021 \pm 0.009$ | $0.062 \pm 0.020$ | $0.040 \pm 0.005$ |
| | CQRNN ($\mathcal{L}_{\text{Cqr}}$) | $0.079 \pm 0.012$ | $0.157 \pm 0.018$ | $0.071 \pm 0.007$ |
| | $\mathcal{L}_{\text{X-CAL}}^{\text{Pre}}$ | $0.019 \pm 0.008$ | $0.051 \pm 0.010$ | $\mathbf{0.034 \pm 0.002}$ |
| | $\mathcal{L}_{\text{ALD+Cqr}}^{\text{Pre}}$ | $0.035 \pm 0.008$ | $0.063 \pm 0.006$ | $0.044 \pm 0.002$ |
| | $\mathcal{L}_{\text{ALD+Cal}}^{\text{Pre}}$ | $\mathbf{0.018 \pm 0.006}$ | $\mathbf{0.050 \pm 0.001}$ | $0.035 \pm 0.003$ |
| LogNorm light | ALD ($\mathcal{L}_{\text{ALD}}$) | $0.021 \pm 0.005$ | $0.053 \pm 0.007$ | $0.027 \pm 0.002$ |
| | CQRNN ($\mathcal{L}_{\text{Cqr}}$) | $0.035 \pm 0.007$ | $0.074 \pm 0.012$ | $0.030 \pm 0.002$ |
| | $\mathcal{L}_{\text{X-CAL}}^{\text{Pre}}$ | $0.023 \pm 0.011$ | $0.056 \pm 0.012$ | $0.026 \pm 0.002$ |
| | $\mathcal{L}_{\text{ALD+Cqr}}^{\text{Pre}}$ | $0.043 \pm 0.012$ | $0.073 \pm 0.010$ | $0.037 \pm 0.002$ |
| | $\mathcal{L}_{\text{ALD+Cal}}^{\text{Pre}}$ | $\mathbf{0.017 \pm 0.003}$ | $\mathbf{0.050 \pm 0.009}$ | $\mathbf{0.025 \pm 0.001}$ |
| LogNorm same | ALD ($\mathcal{L}_{\text{ALD}}$) | $0.018 \pm 0.006$ | $0.052 \pm 0.012$ | $0.012 \pm 0.003$ |
| | CQRNN ($\mathcal{L}_{\text{Cqr}}$) | $0.029 \pm 0.008$ | $0.068 \pm 0.004$ | $0.014 \pm 0.001$ |
| | $\mathcal{L}_{\text{X-CAL}}^{\text{Pre}}$ | $0.030 \pm 0.018$ | $0.059 \pm 0.019$ | $0.012 \pm 0.002$ |
| | $\mathcal{L}_{\text{ALD+Cqr}}^{\text{Pre}}$ | $0.035 \pm 0.004$ | $0.067 \pm 0.007$ | $0.011 \pm 0.004$ |
| | $\mathcal{L}_{\text{ALD+Cal}}^{\text{Pre}}$ | $\mathbf{0.014 \pm 0.002}$ | $\mathbf{0.047 \pm 0.003}$ | $\mathbf{0.008 \pm 0.004}$ |
| METABRIC | ALD ($\mathcal{L}_{\text{ALD}}$) | $0.136 \pm 0.013$ | $0.265 \pm 0.020$ | ———— |
| | CQRNN ($\mathcal{L}_{\text{Cqr}}$) | $0.165 \pm 0.001$ | $0.270 \pm 0.010$ | ———— |
| | $\mathcal{L}_{\text{X-CAL}}^{\text{Pre}}$ | $0.135 \pm 0.007$ | $0.265 \pm 0.016$ | ———— |
| | $\mathcal{L}_{\text{ALD+Cqr}}^{\text{Pre}}$ | $0.134 \pm 0.020$ | $0.230 \pm 0.024$ | ———— |
| | $\mathcal{L}_{\text{ALD+Cal}}^{\text{Pre}}$ | $\mathbf{0.100 \pm 0.016}$ | $\mathbf{0.222 \pm 0.018}$ | ———— |
| WHAS | ALD ($\mathcal{L}_{\text{ALD}}$) | $0.103 \pm 0.030$ | $0.290 \pm 0.020$ | ———— |
| | CQRNN ($\mathcal{L}_{\text{Cqr}}$) | $0.144 \pm 0.021$ | $0.348 \pm 0.021$ | ———— |
| | $\mathcal{L}_{\text{X-CAL}}^{\text{Pre}}$ | $0.103 \pm 0.030$ | $0.288 \pm 0.017$ | ———— |
| | $\mathcal{L}_{\text{ALD+Cqr}}^{\text{Pre}}$ | $0.060 \pm 0.019$ | $0.214 \pm 0.019$ | ———— |
| | $\mathcal{L}_{\text{ALD+Cal}}^{\text{Pre}}$ | $\mathbf{0.046 \pm 0.011}$ | $\mathbf{0.172 \pm 0.016}$ | ———— |
| SUPPORT | ALD ($\mathcal{L}_{\text{ALD}}$) | $0.263 \pm 0.008$ | $0.307 \pm 0.015$ | ———— |
| | CQRNN ($\mathcal{L}_{\text{Cqr}}$) | $0.174 \pm 0.005$ | $0.218 \pm 0.004$ | ———— |
| | $\mathcal{L}_{\text{X-CAL}}^{\text{Pre}}$ | $0.257 \pm 0.005$ | $0.301 \pm 0.011$ | ———— |
| | $\mathcal{L}_{\text{ALD+Cqr}}^{\text{Pre}}$ | $0.175 \pm 0.016$ | $0.214 \pm 0.014$ | ———— |
| | $\mathcal{L}_{\text{ALD+Cal}}^{\text{Pre}}$ | $\mathbf{0.164 \pm 0.015}$ | $\mathbf{0.205 \pm 0.014}$ | ———— |
| GBSG | ALD ($\mathcal{L}_{\text{ALD}}$) | $0.201 \pm 0.017$ | $0.295 \pm 0.029$ | ———— |
| | CQRNN ($\mathcal{L}_{\text{Cqr}}$) | $0.204 \pm 0.008$ | $0.315 \pm 0.019$ | ———— |
| | $\mathcal{L}_{\text{X-CAL}}^{\text{Pre}}$ | $0.202 \pm 0.013$ | $0.292 \pm 0.028$ | ———— |
| | $\mathcal{L}_{\text{ALD+Cqr}}^{\text{Pre}}$ | $0.208 \pm 0.020$ | $0.301 \pm 0.009$ | ———— |
| | $\mathcal{L}_{\text{ALD+Cal}}^{\text{Pre}}$ | $\mathbf{0.161 \pm 0.009}$ | $\mathbf{0.272 \pm 0.018}$ | ———— |
| TMBImmuno | ALD ($\mathcal{L}_{\text{ALD}}$) | $\mathbf{0.228 \pm 0.010}$ | $0.275 \pm 0.015$ | ———— |
| | CQRNN ($\mathcal{L}_{\text{Cqr}}$) | $0.229 \pm 0.010$ | $0.286 \pm 0.016$ | ———— |
| | $\mathcal{L}_{\text{X-CAL}}^{\text{Pre}}$ | $0.229 \pm 0.011$ | $0.276 \pm 0.021$ | ———— |
| | $\mathcal{L}_{\text{ALD+Cqr}}^{\text{Pre}}$ | $0.243 \pm 0.009$ | $0.265 \pm 0.013$ | ———— |
| | $\mathcal{L}_{\text{ALD+Cal}}^{\text{Pre}}$ | $0.229 \pm 0.009$ | $\mathbf{0.252 \pm 0.010}$ | ———— |
| BreastMSK | ALD ($\mathcal{L}_{\text{ALD}}$) | $0.272 \pm 0.026$ | $0.286 \pm 0.031$ | ———— |
| | CQRNN ($\mathcal{L}_{\text{Cqr}}$) | $0.289 \pm 0.004$ | $0.310 \pm 0.006$ | ———— |
| | $\mathcal{L}_{\text{X-CAL}}^{\text{Pre}}$ | $0.276 \pm 0.020$ | $0.292 \pm 0.024$ | ———— |
| | $\mathcal{L}_{\text{ALD+Cqr}}^{\text{Pre}}$ | $0.267 \pm 0.017$ | $0.287 \pm 0.011$ | ———— |
| | $\mathcal{L}_{\text{ALD+Cal}}^{\text{Pre}}$ | $\mathbf{0.249 \pm 0.016}$ | $\mathbf{0.270 \pm 0.012}$ | ———— |
| LGGGBM | ALD ($\mathcal{L}_{\text{ALD}}$) | $0.173 \pm 0.050$ | $0.348 \pm 0.034$ | ———— |
| | CQRNN ($\mathcal{L}_{\text{Cqr}}$) | $0.180 \pm 0.024$ | $0.372 \pm 0.021$ | ———— |
| | $\mathcal{L}_{\text{X-CAL}}^{\text{Pre}}$ | $0.165 \pm 0.046$ | $\mathbf{0.334 \pm 0.033}$ | ———— |
| | $\mathcal{L}_{\text{ALD+Cqr}}^{\text{Pre}}$ | $\mathbf{0.133 \pm 0.036}$ | $0.364 \pm 0.032$ | ———— |
| | $\mathcal{L}_{\text{ALD+Cal}}^{\text{Pre}}$ | $0.135 \pm 0.023$ | $0.360 \pm 0.061$ | ———— |

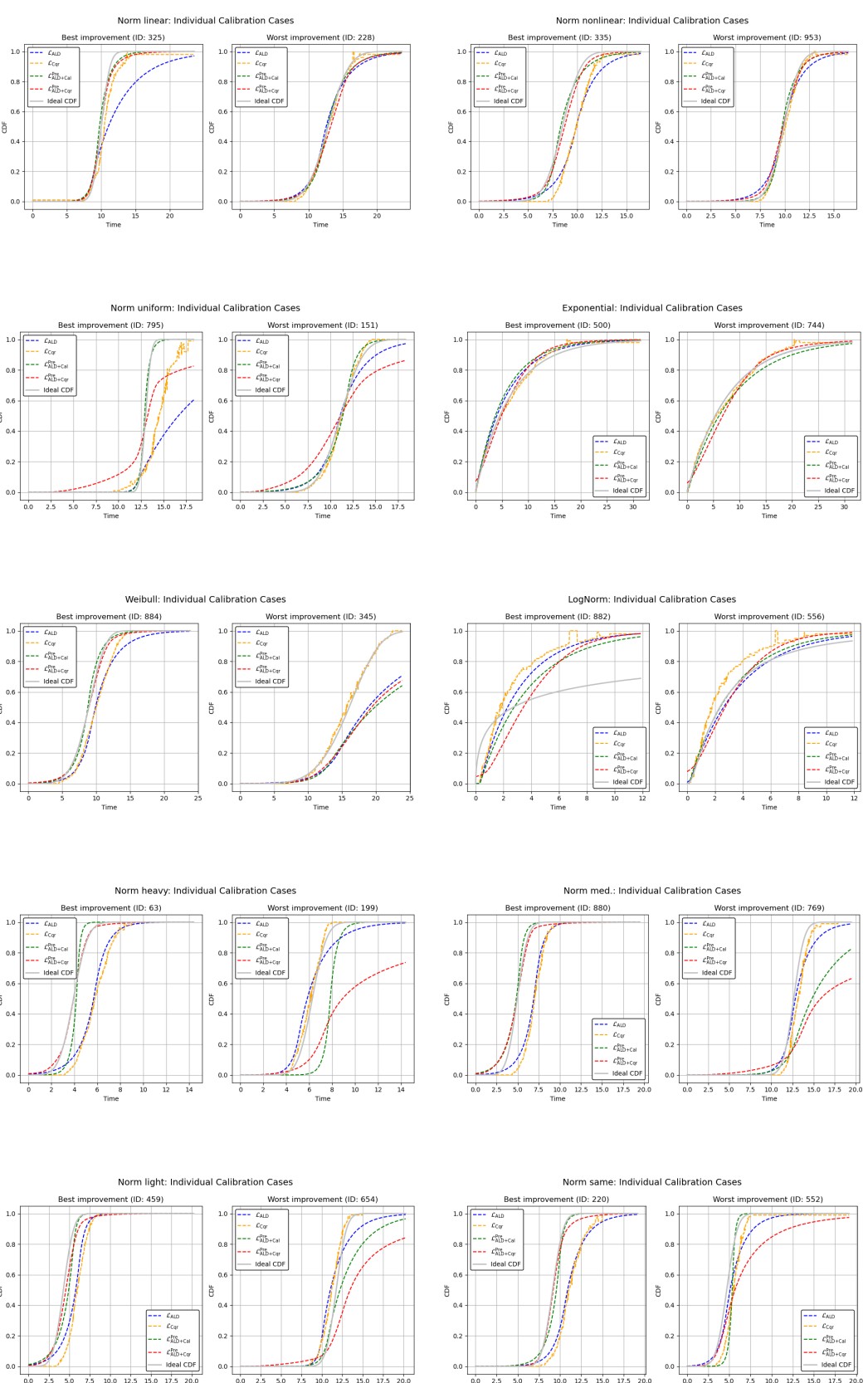

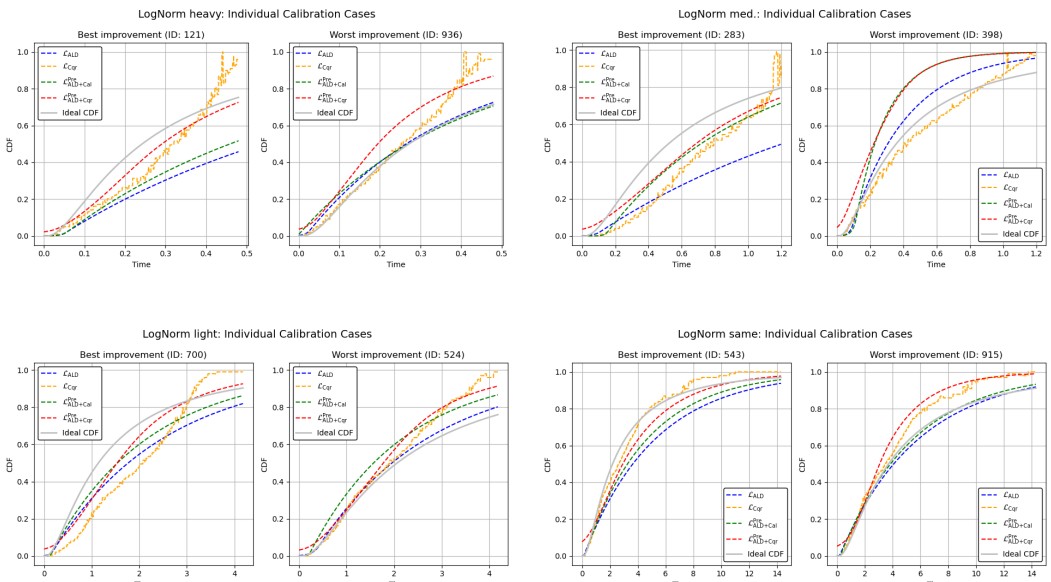

Figure 3: Illustration of the best and worst *individual calibration* improvement cases ($\mathcal{L}^{\text{Pre}}_{\text{ALD+Cal}}$ *vs.* $\mathcal{L}_{\text{ALD}}$) achieved by the hybrid ALD-based survival model across all synthetic datasets.

## C.2 POST-CALIBRATION RESULTS

Table 7 presents the full results for the *post-calibration* setting. The best performance for each dataset and metric is highlighted in **bold**. Fig. 4 illustrates the best and worst *individual calibration* improvement cases with the *post-calibration* setting, comparing $\mathcal{L}_{ALD+Cal}^{Post}$ against $\mathcal{L}_{ALD}$, achieved by the hybrid ALD-based survival model across all synthetic datasets.

Table 7: Full results table on *post-calibration* for all datasets, methods, and metrics. The values represent the mean ± 1 standard error for the test set over 5 runs.

| Dataset | Method | Average Calibration | Group Calibration | Individual Calibration |
|---|---|---|---|---|
| Norm_linear | $\mathcal{L}_{ALD+Cal}^{Pre}$ | **0.016 ± 0.002** | 0.029 ± 0.005 | 0.018 ± 0.002 |
| | $\mathcal{L}_{ALD+Cqr}^{Post}$ | 0.020 ± 0.002 | 0.038 ± 0.006 | 0.020 ± 0.002 |
| | $\mathcal{L}_{ALD+CSD}^{Post}$ | 0.058 ± 0.011 | 0.072 ± 0.017 | 0.058 ± 0.009 |
| | $\mathcal{L}_{ALD+CiPOT}^{Post}$ | 0.047 ± 0.006 | 0.079 ± 0.012 | 0.045 ± 0.005 |
| | $\mathcal{L}_{ALD+Cal}^{Post}$ | 0.018 ± 0.003 | **0.027 ± 0.003** | **0.017 ± 0.001** |
| Norm_nonlinear | $\mathcal{L}_{ALD+Cal}^{Pre}$ | 0.023 ± 0.004 | 0.035 ± 0.004 | **0.012 ± 0.001** |
| | $\mathcal{L}_{ALD+Cqr}^{Post}$ | 0.035 ± 0.004 | 0.047 ± 0.005 | 0.019 ± 0.002 |
| | $\mathcal{L}_{ALD+CSD}^{Post}$ | 0.104 ± 0.006 | 0.138 ± 0.017 | 0.082 ± 0.008 |
| | $\mathcal{L}_{ALD+CiPOT}^{Post}$ | 0.074 ± 0.009 | 0.120 ± 0.011 | 0.061 ± 0.008 |
| | $\mathcal{L}_{ALD+Cal}^{Post}$ | **0.021 ± 0.002** | **0.027 ± 0.003** | **0.012 ± 0.001** |
| Norm uniform | $\mathcal{L}_{ALD+Cal}^{Pre}$ | **0.027 ± 0.004** | 0.038 ± 0.006 | 0.018 ± 0.002 |
| | $\mathcal{L}_{ALD+Cqr}^{Post}$ | 0.105 ± 0.003 | 0.145 ± 0.005 | 0.099 ± 0.003 |
| | $\mathcal{L}_{ALD+CSD}^{Post}$ | 0.145 ± 0.039 | 0.176 ± 0.028 | 0.127 ± 0.018 |
| | $\mathcal{L}_{ALD+CiPOT}^{Post}$ | 0.097 ± 0.008 | 0.160 ± 0.020 | 0.099 ± 0.020 |
| | $\mathcal{L}_{ALD+Cal}^{Post}$ | **0.027 ± 0.002** | **0.031 ± 0.001** | **0.016 ± 0.002** |
| Exponential | $\mathcal{L}_{ALD+Cal}^{Pre}$ | **0.012 ± 0.005** | 0.020 ± 0.006 | 0.015 ± 0.003 |
| | $\mathcal{L}_{ALD+Cqr}^{Post}$ | 0.027 ± 0.008 | 0.038 ± 0.006 | 0.020 ± 0.005 |
| | $\mathcal{L}_{ALD+CSD}^{Post}$ | 0.058 ± 0.036 | 0.064 ± 0.036 | 0.031 ± 0.020 |
| | $\mathcal{L}_{ALD+CiPOT}^{Post}$ | 0.037 ± 0.035 | 0.049 ± 0.034 | 0.026 ± 0.022 |
| | $\mathcal{L}_{ALD+Cal}^{Post}$ | **0.012 ± 0.005** | **0.019 ± 0.006** | **0.011 ± 0.004** |
| Weibull | $\mathcal{L}_{ALD+Cal}^{Pre}$ | **0.019 ± 0.005** | 0.032 ± 0.002 | **0.020 ± 0.002** |
| | $\mathcal{L}_{ALD+Cqr}^{Post}$ | 0.026 ± 0.005 | 0.036 ± 0.007 | 0.026 ± 0.003 |
| | $\mathcal{L}_{ALD+CSD}^{Post}$ | 0.119 ± 0.029 | 0.153 ± 0.045 | 0.054 ± 0.009 |
| | $\mathcal{L}_{ALD+CiPOT}^{Post}$ | 0.063 ± 0.022 | 0.085 ± 0.029 | 0.048 ± 0.006 |
| | $\mathcal{L}_{ALD+Cal}^{Post}$ | 0.021 ± 0.004 | **0.030 ± 0.005** | **0.020 ± 0.003** |
| LogNorm | $\mathcal{L}_{ALD+Cal}^{Pre}$ | 0.015 ± 0.003 | 0.021 ± 0.002 | 0.131 ± 0.003 |
| | $\mathcal{L}_{ALD+Cqr}^{Post}$ | 0.048 ± 0.005 | 0.053 ± 0.005 | 0.138 ± 0.004 |
| | $\mathcal{L}_{ALD+CSD}^{Post}$ | 0.070 ± 0.015 | 0.081 ± 0.014 | 0.141 ± 0.008 |
| | $\mathcal{L}_{ALD+CiPOT}^{Post}$ | 0.021 ± 0.009 | 0.033 ± 0.013 | **0.127 ± 0.007** |
| | $\mathcal{L}_{ALD+Cal}^{Post}$ | **0.014 ± 0.003** | **0.019 ± 0.002** | 0.128 ± 0.004 |
| Norm heavy | $\mathcal{L}_{ALD+Cal}^{Pre}$ | 0.063 ± 0.010 | 0.109 ± 0.017 | 0.038 ± 0.003 |
| | $\mathcal{L}_{ALD+Cqr}^{Post}$ | 0.144 ± 0.009 | 0.223 ± 0.023 | 0.111 ± 0.002 |
| | $\mathcal{L}_{ALD+CSD}^{Post}$ | 0.155 ± 0.047 | 0.217 ± 0.030 | 0.144 ± 0.011 |
| | $\mathcal{L}_{ALD+CiPOT}^{Post}$ | 0.063 ± 0.010 | 0.134 ± 0.036 | 0.048 ± 0.006 |
| | $\mathcal{L}_{ALD+Cal}^{Post}$ | **0.033 ± 0.006** | **0.067 ± 0.003** | **0.030 ± 0.002** |
| Norm med. | $\mathcal{L}_{ALD+Cal}^{Pre}$ | 0.044 ± 0.004 | 0.076 ± 0.006 | 0.020 ± 0.001 |
| | $\mathcal{L}_{ALD+Cqr}^{Post}$ | 0.081 ± 0.004 | 0.107 ± 0.008 | 0.072 ± 0.003 |
| | $\mathcal{L}_{ALD+CSD}^{Post}$ | 0.148 ± 0.048 | 0.180 ± 0.050 | 0.174 ± 0.017 |
| | $\mathcal{L}_{ALD+CiPOT}^{Post}$ | 0.091 ± 0.073 | 0.129 ± 0.070 | 0.034 ± 0.012 |
| | $\mathcal{L}_{ALD+Cal}^{Post}$ | **0.031 ± 0.004** | **0.050 ± 0.004** | **0.017 ± 0.001** |
| Norm light | $\mathcal{L}_{ALD+Cal}^{Pre}$ | 0.032 ± 0.006 | 0.059 ± 0.005 | 0.016 ± 0.001 |
| | $\mathcal{L}_{ALD+Cqr}^{Post}$ | 0.048 ± 0.003 | 0.067 ± 0.004 | 0.037 ± 0.001 |
| | $\mathcal{L}_{ALD+CSD}^{Post}$ | 0.113 ± 0.020 | 0.133 ± 0.027 | 0.194 ± 0.017 |
| | $\mathcal{L}_{ALD+CiPOT}^{Post}$ | 0.074 ± 0.035 | 0.107 ± 0.029 | 0.027 ± 0.005 |
| | $\mathcal{L}_{ALD+Cal}^{Post}$ | **0.028 ± 0.002** | **0.043 ± 0.002** | **0.014 ± 0.001** |
| Norm same | $\mathcal{L}_{ALD+Cal}^{Pre}$ | 0.025 ± 0.003 | 0.053 ± 0.009 | 0.023 ± 0.002 |
| | $\mathcal{L}_{ALD+Cqr}^{Post}$ | 0.027 ± 0.005 | 0.054 ± 0.005 | 0.023 ± 0.002 |
| | $\mathcal{L}_{ALD+CSD}^{Post}$ | 0.201 ± 0.027 | 0.231 ± 0.023 | 0.055 ± 0.018 |
| | $\mathcal{L}_{ALD+CiPOT}^{Post}$ | 0.065 ± 0.012 | 0.096 ± 0.012 | 0.044 ± 0.008 |
| | $\mathcal{L}_{ALD+Cal}^{Post}$ | **0.021 ± 0.004** | **0.043 ± 0.003** | **0.017 ± 0.001** |

| Dataset | Method | Average Calibration | Group Calibration | Individual Calibration |
|---|---|---|---|---|
| LogNorm heavy | $\mathcal{L}^{Pre}_{ALD+Cal}$ | 0.025 ± 0.005 | 0.072 ± 0.009 | **0.039 ± 0.005** |
| | $\mathcal{L}^{Post}_{ALD+Cqr}$ | 0.032 ± 0.001 | **0.056 ± 0.003** | 0.040 ± 0.002 |
| | $\mathcal{L}^{Post}_{ALD+CSD}$ | 0.239 ± 0.006 | 0.256 ± 0.011 | 0.153 ± 0.015 |
| | $\mathcal{L}^{Post}_{ALD+CiPOT}$ | 0.167 ± 0.025 | 0.304 ± 0.025 | 0.038 ± 0.008 |
| | $\mathcal{L}^{Post}_{ALD+Cal}$ | **0.020 ± 0.003** | 0.058 ± 0.008 | 0.041 ± 0.003 |
| LogNorm med. | $\mathcal{L}^{Pre}_{ALD+Cal}$ | 0.018 ± 0.006 | 0.050 ± 0.001 | **0.035 ± 0.003** |
| | $\mathcal{L}^{Post}_{ALD+Cqr}$ | 0.033 ± 0.004 | 0.058 ± 0.008 | 0.046 ± 0.003 |
| | $\mathcal{L}^{Post}_{ALD+CSD}$ | 0.080 ± 0.005 | 0.090 ± 0.006 | 0.073 ± 0.012 |
| | $\mathcal{L}^{Post}_{ALD+CiPOT}$ | 0.060 ± 0.015 | 0.141 ± 0.019 | 0.040 ± 0.005 |
| | $\mathcal{L}^{Post}_{ALD+Cal}$ | **0.014 ± 0.003** | **0.046 ± 0.003** | 0.040 ± 0.002 |
| LogNorm light | $\mathcal{L}^{Pre}_{ALD+Cal}$ | 0.017 ± 0.003 | 0.050 ± 0.009 | **0.025 ± 0.001** |
| | $\mathcal{L}^{Post}_{ALD+Cqr}$ | 0.043 ± 0.004 | 0.072 ± 0.007 | 0.035 ± 0.001 |
| | $\mathcal{L}^{Post}_{ALD+CSD}$ | 0.163 ± 0.020 | 0.169 ± 0.018 | 0.275 ± 0.035 |
| | $\mathcal{L}^{Post}_{ALD+CiPOT}$ | 0.021 ± 0.007 | 0.060 ± 0.010 | 0.027 ± 0.002 |
| | $\mathcal{L}^{Post}_{ALD+Cal}$ | **0.013 ± 0.001** | **0.042 ± 0.003** | 0.030 ± 0.001 |
| LogNorm same | $\mathcal{L}^{Pre}_{ALD+Cal}$ | **0.014 ± 0.002** | 0.047 ± 0.003 | **0.008 ± 0.004** |
| | $\mathcal{L}^{Post}_{ALD+Cqr}$ | 0.031 ± 0.003 | 0.051 ± 0.006 | 0.012 ± 0.005 |
| | $\mathcal{L}^{Post}_{ALD+CSD}$ | 0.291 ± 0.009 | 0.309 ± 0.014 | 0.562 ± 0.075 |
| | $\mathcal{L}^{Post}_{ALD+CiPOT}$ | 0.017 ± 0.006 | 0.059 ± 0.010 | 0.012 ± 0.003 |
| | $\mathcal{L}^{Post}_{ALD+Cal}$ | **0.014 ± 0.002** | **0.042 ± 0.003** | **0.008 ± 0.002** |
| METABRIC | $\mathcal{L}^{Pre}_{ALD+Cal}$ | 0.100 ± 0.016 | 0.222 ± 0.018 | ————— |
| | $\mathcal{L}^{Post}_{ALD+Cqr}$ | 0.145 ± 0.012 | 0.257 ± 0.018 | ————— |
| | $\mathcal{L}^{Post}_{ALD+CSD}$ | **0.095 ± 0.006** | **0.105 ± 0.006** | ————— |
| | $\mathcal{L}^{Post}_{ALD+CiPOT}$ | 0.108 ± 0.009 | 0.229 ± 0.047 | ————— |
| | $\mathcal{L}^{Post}_{ALD+Cal}$ | 0.120 ± 0.015 | 0.242 ± 0.008 | ————— |
| WHAS | $\mathcal{L}^{Pre}_{ALD+Cal}$ | **0.046 ± 0.011** | 0.172 ± 0.016 | ————— |
| | $\mathcal{L}^{Post}_{ALD+Cqr}$ | 0.096 ± 0.017 | 0.287 ± 0.014 | ————— |
| | $\mathcal{L}^{Post}_{ALD+CSD}$ | 0.064 ± 0.008 | **0.138 ± 0.020** | ————— |
| | $\mathcal{L}^{Post}_{ALD+CiPOT}$ | 0.222 ± 0.020 | 0.387 ± 0.011 | ————— |
| | $\mathcal{L}^{Post}_{ALD+Cal}$ | 0.064 ± 0.013 | 0.244 ± 0.010 | ————— |
| SUPPORT | $\mathcal{L}^{Pre}_{ALD+Cal}$ | 0.164 ± 0.015 | 0.205 ± 0.014 | ————— |
| | $\mathcal{L}^{Post}_{ALD+Cqr}$ | 0.215 ± 0.005 | 0.251 ± 0.012 | ————— |
| | $\mathcal{L}^{Post}_{ALD+CSD}$ | **0.102 ± 0.012** | **0.115 ± 0.012** | ————— |
| | $\mathcal{L}^{Post}_{ALD+CiPOT}$ | 0.139 ± 0.011 | 0.324 ± 0.032 | ————— |
| | $\mathcal{L}^{Post}_{ALD+Cal}$ | 0.240 ± 0.003 | 0.283 ± 0.007 | ————— |
| GBSG | $\mathcal{L}^{Pre}_{ALD+Cal}$ | 0.161 ± 0.009 | 0.272 ± 0.018 | ————— |
| | $\mathcal{L}^{Post}_{ALD+Cqr}$ | 0.217 ± 0.008 | 0.304 ± 0.011 | ————— |
| | $\mathcal{L}^{Post}_{ALD+CSD}$ | **0.089 ± 0.002** | **0.104 ± 0.021** | ————— |
| | $\mathcal{L}^{Post}_{ALD+CiPOT}$ | 0.130 ± 0.016 | 0.234 ± 0.032 | ————— |
| | $\mathcal{L}^{Post}_{ALD+Cal}$ | 0.185 ± 0.003 | 0.274 ± 0.009 | ————— |
| TMBImmuno | $\mathcal{L}^{Pre}_{ALD+Cal}$ | 0.229 ± 0.009 | 0.252 ± 0.010 | ————— |
| | $\mathcal{L}^{Post}_{ALD+Cqr}$ | 0.249 ± 0.012 | 0.269 ± 0.010 | ————— |
| | $\mathcal{L}^{Post}_{ALD+CSD}$ | **0.150 ± 0.012** | **0.184 ± 0.021** | ————— |
| | $\mathcal{L}^{Post}_{ALD+CiPOT}$ | 0.157 ± 0.005 | 0.210 ± 0.018 | ————— |
| | $\mathcal{L}^{Post}_{ALD+Cal}$ | 0.214 ± 0.010 | 0.242 ± 0.014 | ————— |
| BreastMSK | $\mathcal{L}^{Pre}_{ALD+Cal}$ | 0.249 ± 0.016 | 0.270 ± 0.012 | ————— |
| | $\mathcal{L}^{Post}_{ALD+Cqr}$ | 0.270 ± 0.014 | 0.287 ± 0.016 | ————— |
| | $\mathcal{L}^{Post}_{ALD+CSD}$ | **0.212 ± 0.012** | **0.225 ± 0.010** | ————— |
| | $\mathcal{L}^{Post}_{ALD+CiPOT}$ | 0.325 ± 0.009 | 0.433 ± 0.079 | ————— |
| | $\mathcal{L}^{Post}_{ALD+Cal}$ | 0.247 ± 0.011 | 0.272 ± 0.010 | ————— |
| LGGGBM | $\mathcal{L}^{Pre}_{ALD+Cal}$ | 0.135 ± 0.023 | 0.360 ± 0.061 | ————— |
| | $\mathcal{L}^{Post}_{ALD+Cqr}$ | 0.152 ± 0.021 | 0.335 ± 0.044 | ————— |
| | $\mathcal{L}^{Post}_{ALD+CSD}$ | 0.351 ± 0.027 | 0.431 ± 0.022 | ————— |
| | $\mathcal{L}^{Post}_{ALD+CiPOT}$ | 0.267 ± 0.021 | 0.415 ± 0.011 | ————— |
| | $\mathcal{L}^{Post}_{ALD+Cal}$ | **0.118 ± 0.023** | **0.257 ± 0.037** | ————— |

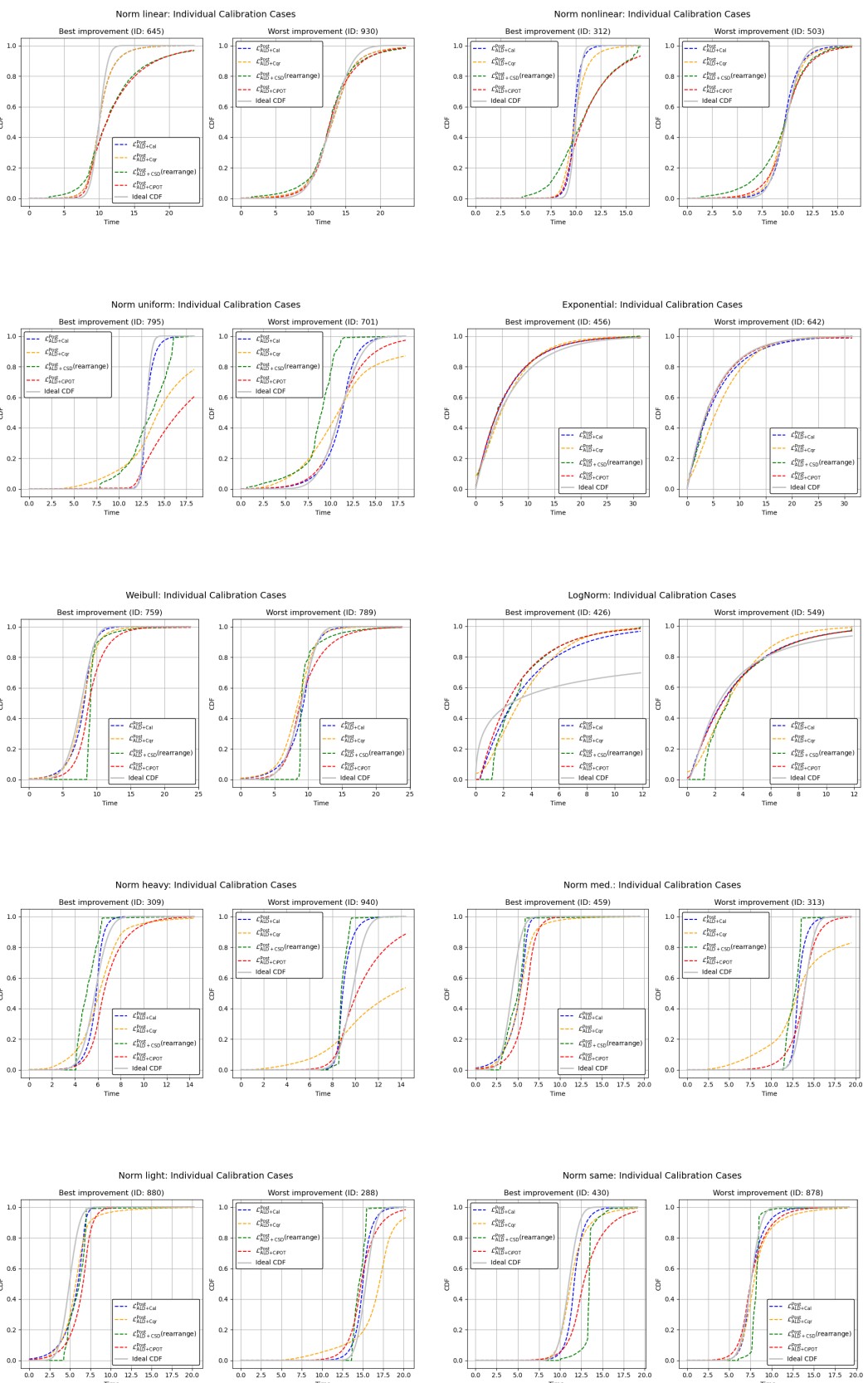

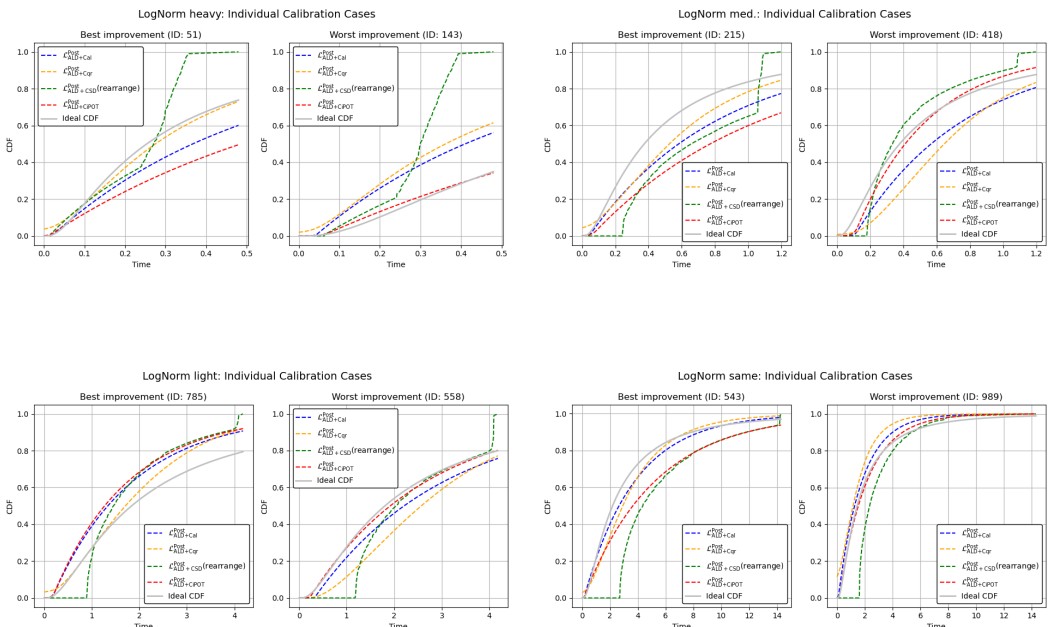

Figure 4: Illustration of the best and worst *individual calibration* improvement cases ($\mathcal{L}_{\text{ALD+Cal}}^{\text{Post}}$ *vs.* $\mathcal{L}_{\text{ALD}}$) achieved by the hybrid ALD-based survival model across all synthetic datasets.

## C.3 OVERALL RESULTS

Table 8 summarizes the full results across 21 datasets, comparing our method with 9 baselines across 7 metrics. The best performance for each dataset and metric is highlighted in **bold**.

Table 8: Full results table for all datasets, methods, and metrics. The values represent the mean ± 1 standard error for the test set over 5 runs.

| Dataset | Methods | MAE | IBS | Harrell's C-index | Uno's C-index | Average Calibration | Group Calibration | Individual Calibration |
|---|---|---|---|---|---|---|---|---|
| Norm_linear | ALD | 0.692 ± 0.124 | 0.285 ± 0.008 | 0.619 ± 0.026 | 0.619 ± 0.022 | 0.047 ± 0.006 | 0.079 ± 0.012 | 0.044 ± 0.005 |
| | CQRNN | **0.217 ± 0.070** | **0.268 ± 0.007** | 0.656 ± 0.011 | 0.650 ± 0.009 | 0.035 ± 0.006 | 0.054 ± 0.005 | 0.018 ± 0.002 |
| | LogNorm | 0.251 ± 0.082 | 0.713 ± 0.007 | 0.657 ± 0.009 | 0.651 ± 0.008 | 0.022 ± 0.007 | 0.040 ± 0.015 | **0.014 ± 0.002** |
| | DeepSurv | 0.248 ± 0.156 | 0.665 ± 0.028 | 0.657 ± 0.008 | 0.651 ± 0.008 | 0.019 ± 0.007 | 0.051 ± 0.022 | 0.016 ± 0.006 |
| | DSM(Weibull) | 1.054 ± 0.034 | 0.322 ± 0.006 | 0.653 ± 0.010 | 0.647 ± 0.009 | 0.091 ± 0.009 | 0.188 ± 0.006 | 0.066 ± 0.002 |
| | DSM(LogNorm) | 1.048 ± 0.031 | 0.323 ± 0.006 | 0.651 ± 0.009 | 0.646 ± 0.008 | 0.090 ± 0.012 | 0.203 ± 0.010 | 0.057 ± 0.003 |
| | DeepHit | 1.666 ± 0.442 | 0.505 ± 0.022 | 0.616 ± 0.030 | 0.607 ± 0.031 | 0.189 ± 0.044 | 0.234 ± 0.061 | 0.151 ± 0.019 |
| | GBM | 0.634 ± 0.022 | 0.305 ± 0.007 | 0.636 ± 0.014 | 0.631 ± 0.012 | **0.017 ± 0.004** | 0.091 ± 0.011 | 0.030 ± 0.001 |
| | RSF | 1.329 ± 0.149 | 0.331 ± 0.007 | 0.581 ± 0.013 | 0.577 ± 0.012 | 0.036 ± 0.002 | 0.053 ± 0.006 | 0.102 ± 0.009 |
| | $\mathcal{L}^{\text{Post}}_{\text{ALD+Cal}}$ | 0.632 ± 0.122 | 0.275 ± 0.005 | **0.661 ± 0.009** | **0.653 ± 0.008** | 0.018 ± 0.003 | **0.027 ± 0.003** | 0.017 ± 0.001 |
| Norm_nonlinear | ALD | 0.607 ± 0.185 | 0.241 ± 0.013 | 0.606 ± 0.030 | 0.575 ± 0.033 | 0.072 ± 0.010 | 0.119 ± 0.012 | 0.060 ± 0.008 |
| | CQRNN | 0.432 ± 0.039 | 0.253 ± 0.036 | 0.594 ± 0.012 | 0.560 ± 0.013 | 0.034 ± 0.011 | 0.078 ± 0.011 | 0.029 ± 0.002 |
| | LogNorm | 0.228 ± 0.049 | 0.564 ± 0.013 | 0.659 ± 0.016 | 0.643 ± 0.014 | 0.066 ± 0.006 | 0.114 ± 0.016 | 0.027 ± 0.005 |
| | DeepSurv | **0.174 ± 0.050** | 0.619 ± 0.013 | 0.669 ± 0.016 | 0.649 ± 0.015 | 0.023 ± 0.007 | 0.065 ± 0.012 | 0.022 ± 0.003 |
| | DSM(Weibull) | 0.469 ± 0.025 | 0.236 ± 0.003 | 0.627 ± 0.022 | 0.605 ± 0.021 | 0.097 ± 0.002 | 0.132 ± 0.004 | 0.067 ± 0.002 |
| | DSM(LogNorm) | 0.497 ± 0.033 | 0.234 ± 0.003 | 0.595 ± 0.021 | 0.563 ± 0.022 | 0.084 ± 0.011 | 0.125 ± 0.013 | 0.055 ± 0.003 |
| | DeepHit | 1.027 ± 0.146 | 0.515 ± 0.042 | 0.610 ± 0.049 | 0.600 ± 0.041 | 0.150 ± 0.114 | 0.189 ± 0.127 | 0.067 ± 0.024 |
| | GBM | 0.314 ± 0.030 | **0.227 ± 0.003** | 0.655 ± 0.021 | 0.638 ± 0.019 | 0.026 ± 0.002 | 0.066 ± 0.006 | 0.027 ± 0.001 |
| | RSF | 0.508 ± 0.051 | 0.243 ± 0.004 | 0.623 ± 0.015 | 0.605 ± 0.014 | 0.043 ± 0.003 | 0.052 ± 0.003 | 0.048 ± 0.003 |
| | $\mathcal{L}^{\text{Post}}_{\text{ALD+Cal}}$ | 0.794 ± 0.184 | 0.261 ± 0.012 | **0.679 ± 0.005** | **0.657 ± 0.006** | **0.021 ± 0.002** | **0.027 ± 0.003** | **0.012 ± 0.001** |
| Norm_uniform | ALD | 2.307 ± 0.664 | 0.049 ± 0.002 | 0.768 ± 0.019 | 0.693 ± 0.024 | 0.095 ± 0.009 | 0.159 ± 0.020 | 0.098 ± 0.020 |
| | CQRNN | 0.690 ± 0.228 | 0.154 ± 0.077 | 0.767 ± 0.022 | 0.680 ± 0.017 | 0.036 ± 0.009 | 0.113 ± 0.029 | 0.054 ± 0.007 |
| | LogNorm | 15.876 ± 3.013 | 0.387 ± 0.014 | 0.576 ± 0.175 | 0.574 ± 0.107 | 0.208 ± 0.003 | 0.234 ± 0.001 | 0.236 ± 0.013 |
| | DeepSurv | **0.573 ± 0.195** | 0.517 ± 0.012 | **0.784 ± 0.010** | **0.704 ± 0.014** | 0.059 ± 0.014 | 0.102 ± 0.016 | 0.064 ± 0.003 |
| | DSM(Weibull) | 1.344 ± 0.012 | 0.062 ± 0.002 | 0.764 ± 0.016 | 0.679 ± 0.018 | 0.074 ± 0.013 | 0.196 ± 0.012 | 0.106 ± 0.002 |
| | DSM(LogNorm) | 1.306 ± 0.016 | 0.062 ± 0.002 | 0.781 ± 0.013 | 0.692 ± 0.021 | 0.113 ± 0.021 | 0.201 ± 0.009 | 0.147 ± 0.018 |
| | DeepHit | 1.353 ± 0.506 | 0.368 ± 0.057 | 0.756 ± 0.025 | 0.691 ± 0.025 | 0.253 ± 0.077 | 0.355 ± 0.113 | 0.147 ± 0.040 |
| | GBM | 1.106 ± 0.121 | 0.058 ± 0.003 | 0.746 ± 0.019 | 0.677 ± 0.012 | 0.032 ± 0.013 | 0.156 ± 0.013 | 0.071 ± 0.004 |
| | RSF | 1.154 ± 0.086 | 0.056 ± 0.002 | 0.657 ± 0.027 | 0.616 ± 0.021 | 0.058 ± 0.010 | 0.072 ± 0.012 | 0.078 ± 0.006 |
| | $\mathcal{L}^{\text{Post}}_{\text{ALD+Cal}}$ | 4.887 ± 3.392 | **0.048 ± 0.003** | 0.773 ± 0.016 | 0.694 ± 0.011 | **0.027 ± 0.002** | **0.031 ± 0.001** | **0.016 ± 0.002** |
| Exponential | ALD | 0.534 ± 0.153 | **0.292 ± 0.004** | **0.564 ± 0.005** | **0.563 ± 0.005** | 0.018 ± 0.011 | 0.030 ± 0.014 | 0.016 ± 0.003 |
| | CQRNN | 1.682 ± 0.316 | **0.292 ± 0.009** | 0.553 ± 0.012 | 0.555 ± 0.008 | 0.030 ± 0.008 | 0.051 ± 0.011 | 0.030 ± 0.003 |
| | LogNorm | 3.220 ± 0.577 | 0.454 ± 0.005 | 0.516 ± 0.041 | 0.518 ± 0.040 | 0.040 ± 0.008 | 0.056 ± 0.008 | 0.054 ± 0.006 |
| | DeepSurv | 2.033 ± 0.184 | 0.480 ± 0.015 | **0.564 ± 0.004** | **0.563 ± 0.004** | 0.019 ± 0.008 | 0.037 ± 0.016 | 0.018 ± 0.006 |
| | DSM(Weibull) | 1.915 ± 0.237 | 0.293 ± 0.004 | 0.563 ± 0.003 | 0.562 ± 0.004 | 0.016 ± 0.006 | 0.062 ± 0.003 | 0.030 ± 0.005 |
| | DSM(LogNorm) | 2.370 ± 0.235 | 0.295 ± 0.004 | 0.556 ± 0.004 | 0.554 ± 0.005 | 0.035 ± 0.008 | 0.062 ± 0.007 | 0.064 ± 0.006 |
| | DeepHit | 1.190 ± 0.267 | 0.469 ± 0.011 | 0.528 ± 0.025 | 0.529 ± 0.023 | 0.085 ± 0.040 | 0.121 ± 0.034 | 0.070 ± 0.040 |
| | GBM | 2.139 ± 0.236 | 0.296 ± 0.005 | 0.537 ± 0.012 | 0.537 ± 0.010 | 0.021 ± 0.005 | 0.045 ± 0.011 | 0.025 ± 0.006 |
| | RSF | 3.507 ± 0.292 | 0.345 ± 0.014 | 0.516 ± 0.016 | 0.515 ± 0.015 | 0.042 ± 0.010 | 0.057 ± 0.015 | 0.137 ± 0.005 |
| | $\mathcal{L}^{\text{Post}}_{\text{ALD+Cal}}$ | **0.500 ± 0.106** | 0.292 ± 0.007 | 0.548 ± 0.004 | 0.550 ± 0.005 | **0.012 ± 0.005** | **0.019 ± 0.006** | **0.011 ± 0.004** |
| Weibull | ALD | 0.642 ± 0.111 | **0.211 ± 0.004** | 0.762 ± 0.010 | 0.758 ± 0.009 | 0.048 ± 0.009 | 0.067 ± 0.006 | 0.042 ± 0.003 |
| | CQRNN | 0.878 ± 0.243 | 0.223 ± 0.007 | 0.752 ± 0.010 | 0.748 ± 0.005 | 0.031 ± 0.004 | 0.086 ± 0.021 | 0.040 ± 0.010 |
| | LogNorm | 0.858 ± 0.113 | 0.838 ± 0.023 | 0.773 ± 0.006 | 0.768 ± 0.007 | 0.043 ± 0.014 | 0.115 ± 0.022 | 0.044 ± 0.005 |
| | DeepSurv | **0.360 ± 0.077** | 0.969 ± 0.021 | **0.774 ± 0.005** | **0.769 ± 0.006** | 0.018 ± 0.009 | **0.029 ± 0.014** | **0.018 ± 0.003** |
| | DSM(Weibull) | 2.647 ± 0.073 | 0.329 ± 0.010 | 0.745 ± 0.003 | 0.743 ± 0.003 | 0.044 ± 0.009 | 0.223 ± 0.012 | 0.127 ± 0.003 |
| | DSM(LogNorm) | 2.590 ± 0.072 | 0.328 ± 0.009 | 0.746 ± 0.004 | 0.744 ± 0.003 | 0.043 ± 0.010 | 0.230 ± 0.018 | 0.129 ± 0.002 |
| | DeepHit | 1.937 ± 0.159 | 0.610 ± 0.033 | 0.770 ± 0.005 | 0.764 ± 0.005 | 0.098 ± 0.018 | 0.195 ± 0.030 | 0.123 ± 0.015 |
| | GBM | 1.430 ± 0.108 | 0.252 ± 0.007 | 0.767 ± 0.006 | 0.762 ± 0.007 | 0.039 ± 0.011 | 0.161 ± 0.016 | 0.063 ± 0.003 |
| | RSF | 1.253 ± 0.149 | 0.233 ± 0.014 | 0.748 ± 0.011 | 0.742 ± 0.012 | 0.047 ± 0.015 | 0.071 ± 0.021 | 0.074 ± 0.008 |
| | $\mathcal{L}^{\text{Post}}_{\text{ALD+Cal}}$ | 0.928 ± 0.348 | 0.212 ± 0.016 | 0.760 ± 0.008 | 0.758 ± 0.006 | 0.021 ± 0.004 | 0.030 ± 0.004 | 0.020 ± 0.003 |
| LogNorm | ALD | 0.277 ± 0.089 | 0.375 ± 0.012 | 0.588 ± 0.020 | 0.584 ± 0.020 | 0.020 ± 0.006 | 0.031 ± 0.012 | 0.128 ± 0.006 |
| | CQRNN | 1.043 ± 0.087 | 0.386 ± 0.014 | 0.584 ± 0.007 | 0.579 ± 0.008 | 0.031 ± 0.013 | 0.050 ± 0.014 | 0.135 ± 0.008 |
| | LogNorm | **0.229 ± 0.037** | 0.647 ± 0.021 | 0.586 ± 0.018 | 0.583 ± 0.018 | **0.013 ± 0.004** | 0.025 ± 0.007 | 0.113 ± 0.002 |
| | DeepSurv | 0.936 ± 0.049 | 0.657 ± 0.033 | 0.587 ± 0.020 | 0.584 ± 0.020 | 0.014 ± 0.003 | 0.030 ± 0.013 | 0.117 ± 0.006 |
| | DSM(Weibull) | 0.768 ± 0.075 | 0.386 ± 0.012 | 0.498 ± 0.013 | 0.498 ± 0.013 | 0.038 ± 0.011 | 0.044 ± 0.011 | 0.122 ± 0.009 |
| | DSM(LogNorm) | 0.948 ± 0.081 | 0.392 ± 0.014 | 0.503 ± 0.007 | 0.503 ± 0.006 | 0.021 ± 0.010 | 0.027 ± 0.010 | **0.110 ± 0.010** |
| | DeepHit | 0.761 ± 0.174 | 0.566 ± 0.027 | 0.529 ± 0.031 | 0.528 ± 0.030 | 0.098 ± 0.048 | 0.103 ± 0.048 | 0.117 ± 0.008 |
| | GBM | 1.037 ± 0.053 | 0.388 ± 0.011 | 0.576 ± 0.020 | 0.573 ± 0.020 | 0.016 ± 0.004 | 0.023 ± 0.004 | 0.127 ± 0.009 |
| | RSF | 1.154 ± 0.072 | 0.440 ± 0.008 | 0.536 ± 0.018 | 0.535 ± 0.017 | 0.040 ± 0.006 | 0.047 ± 0.011 | 0.154 ± 0.009 |
| | $\mathcal{L}^{\text{Post}}_{\text{ALD+Cal}}$ | 0.249 ± 0.067 | **0.366 ± 0.015** | **0.591 ± 0.013** | **0.588 ± 0.014** | 0.014 ± 0.003 | **0.019 ± 0.002** | 0.128 ± 0.004 |
| Norm heavy | ALD | 0.788 ± 0.264 | **0.020 ± 0.001** | **0.916 ± 0.004** | **0.876 ± 0.005** | 0.062 ± 0.009 | 0.113 ± 0.033 | 0.048 ± 0.006 |
| | CQRNN | **0.460 ± 0.045** | 0.476 ± 0.012 | **0.916 ± 0.010** | 0.868 ± 0.017 | 0.071 ± 0.020 | 0.157 ± 0.023 | 0.032 ± 0.003 |
| | LogNorm | 26.184 ± 3.030 | 0.411 ± 0.007 | 0.788 ± 0.040 | 0.704 ± 0.044 | 0.235 ± 0.005 | 0.256 ± 0.009 | 0.261 ± 0.002 |
| | DeepSurv | 1.552 ± 0.147 | 0.559 ± 0.009 | 0.746 ± 0.035 | 0.619 ± 0.036 | 0.070 ± 0.021 | 0.185 ± 0.064 | 0.037 ± 0.003 |
| | DSM(Weibull) | 1.887 ± 0.040 | 0.047 ± 0.003 | 0.830 ± 0.011 | 0.766 ± 0.019 | 0.036 ± 0.009 | 0.240 ± 0.038 | 0.164 ± 0.004 |
| | DSM(LogNorm) | 1.902 ± 0.039 | 0.047 ± 0.003 | 0.786 ± 0.034 | 0.718 ± 0.036 | 0.074 ± 0.009 | 0.238 ± 0.034 | 0.183 ± 0.004 |
| | DeepHit | 0.751 ± 0.093 | 0.493 ± 0.018 | 0.913 ± 0.007 | 0.865 ± 0.010 | 0.067 ± 0.011 | 0.218 ± 0.041 | 0.088 ± 0.010 |
| | GBM | 1.619 ± 0.022 | 0.042 ± 0.003 | 0.870 ± 0.007 | 0.812 ± 0.011 | **0.031 ± 0.009** | 0.208 ± 0.032 | 0.143 ± 0.004 |
| | RSF | 0.619 ± 0.019 | 0.021 ± 0.001 | 0.908 ± 0.009 | 0.853 ± 0.016 | 0.072 ± 0.021 | 0.156 ± 0.063 | 0.070 ± 0.003 |
| | $\mathcal{L}^{\text{Post}}_{\text{ALD+Cal}}$ | 0.809 ± 0.101 | 0.026 ± 0.004 | 0.892 ± 0.013 | 0.826 ± 0.024 | 0.033 ± 0.006 | **0.067 ± 0.003** | **0.030 ± 0.002** |
| Norm med. | ALD | 0.335 ± 0.071 | **0.052 ± 0.005** | **0.888 ± 0.005** | **0.866 ± 0.006** | 0.054 ± 0.031 | 0.086 ± 0.026 | 0.028 ± 0.004 |
| | CQRNN | 0.357 ± 0.014 | 0.498 ± 0.020 | 0.884 ± 0.005 | 0.862 ± 0.004 | 0.050 ± 0.015 | 0.093 ± 0.013 | 0.019 ± 0.001 |
| | LogNorm | 5.524 ± 1.179 | 0.421 ± 0.009 | 0.825 ± 0.028 | 0.792 ± 0.032 | 0.195 ± 0.002 | 0.212 ± 0.007 | 0.231 ± 0.007 |
| | DeepSurv | **0.270 ± 0.039** | 0.726 ± 0.011 | 0.892 ± 0.004 | 0.870 ± 0.004 | 0.029 ± 0.002 | 0.056 ± 0.005 | 0.018 ± 0.002 |
| | DSM(Weibull) | 1.899 ± 0.040 | 0.118 ± 0.006 | 0.755 ± 0.017 | 0.727 ± 0.017 | 0.030 ± 0.003 | 0.235 ± 0.037 | 0.125 ± 0.005 |
| | DSM(LogNorm) | 1.893 ± 0.030 | 0.119 ± 0.006 | 0.685 ± 0.028 | 0.657 ± 0.028 | 0.038 ± 0.003 | 0.214 ± 0.041 | 0.136 ± 0.005 |
| | DeepHit | 0.928 ± 0.069 | 0.574 ± 0.016 | 0.883 ± 0.006 | 0.861 ± 0.005 | 0.122 ± 0.024 | 0.256 ± 0.045 | 0.091 ± 0.004 |
| | GBM | 1.424 ± 0.037 | 0.097 ± 0.005 | 0.857 ± 0.001 | 0.834 ± 0.002 | 0.025 ± 0.004 | 0.210 ± 0.036 | 0.088 ± 0.003 |
| | RSF | 0.433 ± 0.020 | 0.052 ± 0.003 | 0.883 ± 0.004 | 0.860 ± 0.005 | **0.013 ± 0.007** | **0.049 ± 0.014** | 0.035 ± 0.001 |
| | $\mathcal{L}^{\text{Post}}_{\text{ALD+Cal}}$ | 0.624 ± 0.077 | 0.057 ± 0.005 | 0.876 ± 0.005 | 0.852 ± 0.005 | 0.030 ± 0.004 | 0.050 ± 0.004 | **0.017 ± 0.001** |

| Dataset | Methods | MAE | IBS | Harrell's C-index | Uno's C-index | Average Calibration | Group Calibration | Individual Calibration |
|---|---|---|---|---|---|---|---|---|
| Norm light | ALD | 0.374 ± 0.092 | 0.103 ± 0.011 | 0.880 ± 0.003 | 0.872 ± 0.003 | 0.077 ± 0.034 | 0.111 ± 0.027 | 0.027 ± 0.005 |
| | CQRNN | **0.300 ± 0.041** | 0.506 ± 0.015 | 0.878 ± 0.006 | 0.870 ± 0.006 | 0.036 ± 0.018 | 0.083 ± 0.006 | 0.015 ± 0.002 |
| | LogNorm | 1.345 ± 0.393 | 0.551 ± 0.017 | 0.853 ± 0.011 | 0.842 ± 0.013 | 0.185 ± 0.002 | 0.208 ± 0.009 | 0.166 ± 0.011 |
| | DeepSurv | 0.246 ± 0.009 | 0.944 ± 0.016 | **0.882 ± 0.002** | **0.874 ± 0.002** | 0.026 ± 0.004 | 0.056 ± 0.007 | 0.017 ± 0.001 |
| | DSM(Weibull) | 1.904 ± 0.041 | 0.223 ± 0.012 | 0.726 ± 0.018 | 0.716 ± 0.018 | 0.029 ± 0.003 | 0.234 ± 0.038 | 0.124 ± 0.003 |
| | DSM(LogNorm) | 1.906 ± 0.031 | 0.228 ± 0.013 | 0.649 ± 0.021 | 0.639 ± 0.020 | 0.034 ± 0.004 | 0.216 ± 0.039 | 0.128 ± 0.004 |
| | DeepHit | 0.963 ± 0.056 | 0.681 ± 0.024 | 0.876 ± 0.003 | 0.867 ± 0.004 | 0.117 ± 0.033 | 0.251 ± 0.054 | 0.093 ± 0.010 |
| | GBM | 1.318 ± 0.028 | 0.176 ± 0.009 | 0.847 ± 0.004 | 0.839 ± 0.003 | 0.030 ± 0.004 | 0.207 ± 0.033 | 0.080 ± 0.002 |
| | RSF | 0.382 ± 0.013 | **0.099 ± 0.005** | 0.874 ± 0.004 | 0.866 ± 0.004 | **0.014 ± 0.009** | 0.048 ± 0.017 | 0.024 ± 0.001 |
| | $\mathcal{L}^{\text{Post}}_{\text{ALD+Cal}}$ | 0.557 ± 0.102 | 0.106 ± 0.008 | 0.868 ± 0.004 | 0.859 ± 0.004 | 0.028 ± 0.002 | **0.043 ± 0.002** | **0.014 ± 0.001** |
| Norm same | ALD | **0.452 ± 0.137** | **0.071 ± 0.003** | 0.884 ± 0.007 | 0.829 ± 0.019 | 0.065 ± 0.012 | 0.090 ± 0.017 | 0.044 ± 0.008 |
| | CQRNN | 0.334 ± 0.044 | 0.374 ± 0.035 | 0.886 ± 0.004 | 0.841 ± 0.007 | 0.037 ± 0.008 | 0.075 ± 0.008 | 0.022 ± 0.004 |
| | LogNorm | 0.296 ± 0.149 | 0.797 ± 0.011 | **0.894 ± 0.004** | **0.846 ± 0.006** | 0.056 ± 0.043 | 0.091 ± 0.041 | **0.016 ± 0.007** |
| | DeepSurv | 0.255 ± 0.044 | 0.782 ± 0.014 | 0.887 ± 0.004 | 0.832 ± 0.028 | 0.027 ± 0.007 | 0.064 ± 0.013 | 0.016 ± 0.002 |
| | DSM(Weibull) | 2.192 ± 0.060 | 0.176 ± 0.006 | 0.737 ± 0.016 | 0.687 ± 0.012 | 0.125 ± 0.005 | 0.312 ± 0.030 | 0.154 ± 0.003 |
| | DSM(LogNorm) | 2.147 ± 0.064 | 0.177 ± 0.006 | 0.655 ± 0.018 | 0.618 ± 0.015 | 0.120 ± 0.006 | 0.301 ± 0.032 | 0.176 ± 0.004 |
| | DeepHit | 1.274 ± 0.057 | 0.566 ± 0.032 | 0.882 ± 0.003 | 0.826 ± 0.021 | 0.095 ± 0.007 | 0.187 ± 0.046 | 0.088 ± 0.005 |
| | GBM | 1.546 ± 0.077 | 0.141 ± 0.004 | 0.837 ± 0.009 | 0.795 ± 0.006 | 0.088 ± 0.005 | 0.277 ± 0.033 | 0.101 ± 0.004 |
| | RSF | 0.471 ± 0.024 | 0.076 ± 0.003 | 0.874 ± 0.004 | 0.821 ± 0.010 | 0.058 ± 0.012 | 0.100 ± 0.017 | 0.101 ± 0.001 |
| | $\mathcal{L}^{\text{Post}}_{\text{ALD+Cal}}$ | 0.538 ± 0.070 | 0.080 ± 0.002 | 0.875 ± 0.006 | 0.827 ± 0.007 | **0.021 ± 0.004** | **0.043 ± 0.003** | 0.017 ± 0.001 |
| LogNorm heavy | ALD | 0.427 ± 0.383 | **0.096 ± 0.005** | **0.775 ± 0.009** | **0.730 ± 0.019** | 0.037 ± 0.029 | 0.081 ± 0.043 | **0.038 ± 0.008** |
| | CQRNN | 0.738 ± 0.021 | 0.203 ± 0.032 | 0.766 ± 0.007 | 0.722 ± 0.020 | 0.174 ± 0.008 | 0.294 ± 0.014 | 0.113 ± 0.011 |
| | LogNorm | 0.597 ± 0.042 | 0.399 ± 0.010 | 0.619 ± 0.044 | 0.578 ± 0.036 | 0.024 ± 0.013 | 0.149 ± 0.021 | 0.132 ± 0.001 |
| | DeepSurv | 0.835 ± 0.015 | 0.457 ± 0.018 | 0.470 ± 0.031 | 0.445 ± 0.028 | 0.179 ± 0.019 | 0.291 ± 0.033 | 0.047 ± 0.010 |
| | DSM(Weibull) | 0.678 ± 0.018 | 0.124 ± 0.006 | 0.697 ± 0.008 | 0.662 ± 0.002 | 0.103 ± 0.007 | 0.187 ± 0.015 | 0.156 ± 0.003 |
| | DSM(LogNorm) | 0.655 ± 0.015 | 0.126 ± 0.006 | 0.690 ± 0.064 | 0.649 ± 0.054 | 0.135 ± 0.009 | 0.211 ± 0.016 | 0.159 ± 0.003 |
| | DeepHit | 0.721 ± 0.020 | 0.399 ± 0.016 | 0.753 ± 0.015 | 0.708 ± 0.021 | 0.178 ± 0.011 | 0.323 ± 0.013 | 0.184 ± 0.015 |
| | GBM | 0.686 ± 0.016 | 0.119 ± 0.005 | 0.660 ± 0.040 | 0.615 ± 0.032 | 0.145 ± 0.011 | 0.200 ± 0.012 | 0.146 ± 0.002 |
| | RSF | 0.690 ± 0.016 | 0.100 ± 0.004 | 0.706 ± 0.014 | 0.661 ± 0.016 | 0.195 ± 0.009 | 0.272 ± 0.015 | 0.072 ± 0.002 |
| | $\mathcal{L}^{\text{Post}}_{\text{ALD+Cal}}$ | **0.327 ± 0.108** | 0.098 ± 0.005 | 0.762 ± 0.014 | 0.729 ± 0.009 | **0.020 ± 0.003** | **0.058 ± 0.008** | 0.041 ± 0.003 |
| LogNorm med. | ALD | **0.208 ± 0.044** | 0.175 ± 0.004 | 0.746 ± 0.005 | 0.718 ± 0.006 | 0.021 ± 0.009 | 0.062 ± 0.020 | **0.040 ± 0.005** |
| | CQRNN | 0.560 ± 0.035 | 0.206 ± 0.008 | 0.744 ± 0.012 | 0.718 ± 0.013 | 0.079 ± 0.012 | 0.157 ± 0.018 | 0.071 ± 0.007 |
| | LogNorm | 0.458 ± 0.028 | 0.459 ± 0.016 | 0.696 ± 0.013 | 0.671 ± 0.015 | 0.025 ± 0.007 | 0.122 ± 0.014 | 0.109 ± 0.006 |
| | DeepSurv | 0.643 ± 0.031 | 0.539 ± 0.012 | 0.642 ± 0.012 | 0.601 ± 0.014 | 0.070 ± 0.007 | 0.143 ± 0.021 | 0.041 ± 0.005 |
| | DSM(Weibull) | 0.638 ± 0.013 | 0.221 ± 0.006 | 0.668 ± 0.006 | 0.645 ± 0.006 | 0.036 ± 0.004 | 0.158 ± 0.015 | 0.173 ± 0.002 |
| | DSM(LogNorm) | 0.644 ± 0.013 | 0.223 ± 0.007 | 0.719 ± 0.004 | 0.694 ± 0.008 | 0.044 ± 0.006 | 0.167 ± 0.015 | 0.179 ± 0.003 |
| | DeepHit | 0.602 ± 0.022 | 0.423 ± 0.009 | 0.719 ± 0.020 | 0.695 ± 0.017 | 0.062 ± 0.004 | 0.182 ± 0.013 | 0.160 ± 0.004 |
| | GBM | 0.629 ± 0.016 | 0.207 ± 0.007 | 0.708 ± 0.007 | 0.681 ± 0.006 | 0.046 ± 0.009 | 0.133 ± 0.008 | 0.145 ± 0.001 |
| | RSF | 0.501 ± 0.013 | 0.180 ± 0.006 | 0.729 ± 0.004 | 0.700 ± 0.003 | 0.082 ± 0.006 | 0.135 ± 0.009 | 0.081 ± 0.003 |
| | $\mathcal{L}^{\text{Post}}_{\text{ALD+Cal}}$ | 0.257 ± 0.022 | **0.173 ± 0.007** | **0.749 ± 0.009** | **0.721 ± 0.010** | **0.014 ± 0.003** | **0.046 ± 0.003** | **0.040 ± 0.002** |
| LogNorm light | ALD | **0.143 ± 0.026** | **0.306 ± 0.012** | 0.726 ± 0.008 | 0.715 ± 0.010 | 0.021 ± 0.005 | 0.053 ± 0.007 | **0.027 ± 0.002** |
| | CQRNN | 0.415 ± 0.056 | 0.334 ± 0.010 | 0.720 ± 0.009 | 0.709 ± 0.008 | 0.035 ± 0.007 | 0.074 ± 0.012 | 0.030 ± 0.002 |
| | LogNorm | 0.296 ± 0.017 | 0.805 ± 0.026 | 0.712 ± 0.008 | 0.701 ± 0.010 | 0.017 ± 0.004 | 0.073 ± 0.009 | 0.047 ± 0.002 |
| | DeepSurv | 0.401 ± 0.011 | 0.837 ± 0.022 | 0.713 ± 0.009 | 0.698 ± 0.013 | 0.022 ± 0.005 | 0.058 ± 0.005 | 0.032 ± 0.003 |
| | DSM(Weibull) | 0.623 ± 0.012 | 0.385 ± 0.010 | 0.644 ± 0.007 | 0.637 ± 0.008 | 0.031 ± 0.004 | 0.158 ± 0.017 | 0.108 ± 0.000 |
| | DSM(LogNorm) | 0.644 ± 0.013 | 0.388 ± 0.009 | 0.697 ± 0.007 | 0.686 ± 0.009 | 0.019 ± 0.007 | 0.159 ± 0.017 | 0.115 ± 0.001 |
| | DeepHit | 0.588 ± 0.017 | 0.657 ± 0.019 | 0.703 ± 0.008 | 0.692 ± 0.009 | 0.019 ± 0.005 | 0.133 ± 0.014 | 0.089 ± 0.002 |
| | GBM | 0.619 ± 0.016 | 0.366 ± 0.008 | 0.691 ± 0.003 | 0.681 ± 0.005 | 0.017 ± 0.006 | 0.122 ± 0.013 | 0.079 ± 0.001 |
| | RSF | 0.443 ± 0.018 | 0.317 ± 0.006 | 0.715 ± 0.006 | 0.704 ± 0.008 | 0.020 ± 0.003 | 0.054 ± 0.004 | 0.058 ± 0.002 |
| | $\mathcal{L}^{\text{Post}}_{\text{ALD+Cal}}$ | 0.238 ± 0.015 | 0.310 ± 0.007 | **0.727 ± 0.008** | **0.715 ± 0.008** | **0.013 ± 0.001** | **0.042 ± 0.003** | 0.029 ± 0.001 |
| LogNorm same | ALD | **0.156 ± 0.013** | 0.153 ± 0.004 | 0.744 ± 0.007 | 0.698 ± 0.007 | 0.018 ± 0.006 | 0.052 ± 0.012 | 0.012 ± 0.003 |
| | CQRNN | 0.386 ± 0.044 | 0.170 ± 0.007 | 0.747 ± 0.009 | **0.705 ± 0.013** | 0.029 ± 0.008 | 0.068 ± 0.004 | 0.014 ± 0.001 |
| | LogNorm | 0.194 ± 0.010 | 0.532 ± 0.016 | 0.740 ± 0.007 | 0.696 ± 0.006 | 0.024 ± 0.010 | 0.057 ± 0.016 | 0.012 ± 0.003 |
| | DeepSurv | 0.372 ± 0.021 | 0.512 ± 0.007 | **0.745 ± 0.011** | 0.700 ± 0.006 | 0.021 ± 0.003 | 0.050 ± 0.004 | 0.014 ± 0.004 |
| | DSM(Weibull) | 0.601 ± 0.032 | 0.215 ± 0.006 | 0.643 ± 0.011 | 0.614 ± 0.008 | 0.067 ± 0.007 | 0.204 ± 0.004 | 0.042 ± 0.010 |
| | DSM(LogNorm) | 0.612 ± 0.033 | 0.215 ± 0.006 | 0.692 ± 0.020 | 0.656 ± 0.013 | 0.062 ± 0.005 | 0.203 ± 0.007 | 0.057 ± 0.011 |
| | DeepHit | 0.611 ± 0.112 | 0.371 ± 0.013 | 0.617 ± 0.075 | 0.609 ± 0.051 | 0.033 ± 0.016 | 0.141 ± 0.025 | 0.056 ± 0.012 |
| | GBM | 0.580 ± 0.028 | 0.196 ± 0.005 | 0.698 ± 0.011 | 0.661 ± 0.009 | 0.030 ± 0.004 | 0.150 ± 0.003 | 0.035 ± 0.006 |
| | RSF | 0.399 ± 0.015 | 0.166 ± 0.006 | 0.727 ± 0.012 | 0.682 ± 0.009 | 0.050 ± 0.006 | 0.083 ± 0.011 | 0.172 ± 0.009 |
| | $\mathcal{L}^{\text{Post}}_{\text{ALD+Cal}}$ | 0.220 ± 0.004 | **0.149 ± 0.003** | 0.738 ± 0.007 | 0.701 ± 0.005 | **0.014 ± 0.002** | **0.042 ± 0.003** | **0.008 ± 0.002** |
| METABRIC | ALD | 1.520 ± 0.029 | 0.245 ± 0.010 | 0.636 ± 0.016 | 0.637 ± 0.027 | 0.136 ± 0.013 | 0.265 ± 0.020 | ————— |
| | CQRNN | 1.030 ± 0.044 | **0.242 ± 0.013** | 0.634 ± 0.018 | 0.625 ± 0.025 | 0.165 ± 0.001 | 0.270 ± 0.010 | ————— |
| | LogNorm | 1.167 ± 0.030 | 0.597 ± 0.011 | 0.575 ± 0.017 | 0.549 ± 0.033 | 0.172 ± 0.022 | 0.283 ± 0.029 | ————— |
| | DeepSurv | 0.998 ± 0.014 | 0.536 ± 0.025 | 0.643 ± 0.008 | 0.640 ± 0.025 | 0.146 ± 0.008 | 0.275 ± 0.015 | ————— |
| | DSM(Weibull) | 1.026 ± 0.037 | 0.268 ± 0.005 | 0.611 ± 0.016 | 0.602 ± 0.027 | 0.184 ± 0.012 | 0.278 ± 0.026 | ————— |
| | DSM(LogNorm) | 0.985 ± 0.036 | 0.267 ± 0.005 | 0.614 ± 0.016 | 0.583 ± 0.054 | 0.178 ± 0.012 | 0.276 ± 0.025 | ————— |
| | DeepHit | 1.173 ± 0.032 | 0.464 ± 0.007 | 0.558 ± 0.035 | 0.582 ± 0.032 | 0.155 ± 0.011 | 0.250 ± 0.028 | ————— |
| | GBM | **0.957 ± 0.021** | 0.252 ± 0.005 | **0.640 ± 0.015** | **0.648 ± 0.036** | 0.157 ± 0.010 | 0.274 ± 0.019 | ————— |
| | RSF | 1.117 ± 0.043 | 0.245 ± 0.007 | 0.621 ± 0.014 | 0.623 ± 0.028 | 0.142 ± 0.012 | 0.249 ± 0.021 | ————— |
| | $\mathcal{L}^{\text{Post}}_{\text{ALD+Cal}}$ | 1.521 ± 0.127 | 0.260 ± 0.017 | 0.615 ± 0.019 | 0.588 ± 0.061 | **0.120 ± 0.015** | **0.242 ± 0.008** | ————— |
| WHAS | ALD | 3.410 ± 2.095 | 0.139 ± 0.008 | 0.816 ± 0.012 | 0.812 ± 0.011 | 0.103 ± 0.030 | 0.290 ± 0.020 | ————— |
| | CQRNN | 0.891 ± 0.052 | 0.151 ± 0.013 | 0.833 ± 0.013 | 0.826 ± 0.014 | 0.144 ± 0.021 | 0.348 ± 0.021 | ————— |
| | LogNorm | 1.723 ± 0.146 | 0.624 ± 0.014 | 0.614 ± 0.034 | 0.585 ± 0.031 | 0.181 ± 0.024 | 0.264 ± 0.014 | ————— |
| | DeepSurv | 0.880 ± 0.045 | 0.681 ± 0.013 | 0.702 ± 0.013 | 0.634 ± 0.032 | 0.101 ± 0.029 | 0.295 ± 0.018 | ————— |
| | DSM(Weibull) | 1.654 ± 0.053 | 0.208 ± 0.004 | 0.779 ± 0.011 | 0.787 ± 0.012 | 0.247 ± 0.011 | 0.281 ± 0.022 | ————— |
| | DSM(LogNorm) | 1.938 ± 0.068 | 0.212 ± 0.004 | 0.776 ± 0.006 | 0.783 ± 0.015 | 0.245 ± 0.010 | 0.279 ± 0.019 | ————— |
| | DeepHit | 0.906 ± 0.060 | 0.592 ± 0.021 | 0.805 ± 0.016 | 0.805 ± 0.017 | 0.132 ± 0.018 | **0.215 ± 0.032** | ————— |
| | GBM | 1.111 ± 0.075 | 0.166 ± 0.004 | 0.811 ± 0.009 | 0.808 ± 0.012 | 0.187 ± 0.024 | 0.245 ± 0.030 | ————— |
| | RSF | **0.609 ± 0.056** | **0.083 ± 0.008** | **0.864 ± 0.014** | **0.896 ± 0.016** | **0.064 ± 0.018** | 0.258 ± 0.026 | ————— |
| | $\mathcal{L}^{\text{Post}}_{\text{ALD+Cal}}$ | 1.639 ± 0.579 | 0.133 ± 0.011 | 0.828 ± 0.017 | 0.816 ± 0.029 | **0.064 ± 0.013** | 0.244 ± 0.010 | ————— |
| SUPPORT | ALD | 1.116 ± 0.040 | 0.353 ± 0.008 | 0.606 ± 0.009 | 0.608 ± 0.010 | 0.263 ± 0.008 | 0.307 ± 0.015 | ————— |
| | CQRNN | 0.662 ± 0.023 | 0.341 ± 0.009 | 0.609 ± 0.008 | 0.611 ± 0.008 | 0.174 ± 0.005 | 0.218 ± 0.004 | ————— |
| | LogNorm | 1.214 ± 0.073 | 0.766 ± 0.013 | 0.588 ± 0.009 | 0.587 ± 0.009 | 0.216 ± 0.010 | 0.261 ± 0.012 | ————— |
| | DeepSurv | 0.501 ± 0.015 | 0.627 ± 0.007 | 0.598 ± 0.009 | 0.596 ± 0.011 | **0.133 ± 0.005** | 0.180 ± 0.015 | ————— |
| | DSM(Weibull) | 0.573 ± 0.007 | 0.373 ± 0.002 | 0.559 ± 0.010 | 0.563 ± 0.010 | 0.212 ± 0.003 | 0.248 ± 0.007 | ————— |
| | DSM(LogNorm) | 0.506 ± 0.006 | 0.377 ± 0.002 | 0.565 ± 0.008 | 0.566 ± 0.008 | 0.205 ± 0.004 | 0.243 ± 0.007 | ————— |
| | DeepHit | 0.557 ± 0.035 | 0.533 ± 0.006 | 0.578 ± 0.008 | 0.584 ± 0.009 | 0.170 ± 0.011 | 0.210 ± 0.012 | ————— |
| | GBM | **0.425 ± 0.006** | 0.358 ± 0.001 | 0.595 ± 0.007 | 0.599 ± 0.009 | 0.148 ± 0.005 | 0.200 ± 0.010 | ————— |
| | RSF | 0.675 ± 0.026 | **0.338 ± 0.005** | **0.616 ± 0.007** | **0.615 ± 0.009** | 0.139 ± 0.007 | **0.175 ± 0.009** | ————— |
| | $\mathcal{L}^{\text{Post}}_{\text{ALD+Cal}}$ | 1.546 ± 0.195 | 0.405 ± 0.004 | 0.586 ± 0.003 | 0.586 ± 0.003 | 0.240 ± 0.003 | 0.283 ± 0.007 | ————— |
| GBSG | ALD | 1.766 ± 0.171 | **0.273 ± 0.011** | 0.673 ± 0.013 | 0.666 ± 0.012 | 0.201 ± 0.017 | 0.295 ± 0.029 | ————— |
| | CQRNN | 0.917 ± 0.050 | 0.277 ± 0.008 | **0.676 ± 0.014** | **0.667 ± 0.012** | 0.204 ± 0.008 | 0.315 ± 0.019 | ————— |
| | LogNorm | 1.324 ± 0.074 | 0.623 ± 0.018 | 0.638 ± 0.009 | 0.632 ± 0.009 | 0.258 ± 0.020 | 0.318 ± 0.028 | ————— |
| | DeepSurv | **0.708 ± 0.033** | 0.565 ± 0.015 | 0.618 ± 0.019 | 0.610 ± 0.017 | 0.186 ± 0.015 | **0.271 ± 0.027** | ————— |
| | DSM(Weibull) | 1.086 ± 0.031 | 0.306 ± 0.010 | 0.637 ± 0.009 | 0.632 ± 0.009 | 0.233 ± 0.007 | 0.295 ± 0.007 | ————— |
| | DSM(LogNorm) | 1.000 ± 0.030 | 0.305 ± 0.009 | 0.638 ± 0.022 | 0.630 ± 0.019 | 0.229 ± 0.007 | 0.296 ± 0.009 | ————— |
| | DeepHit | 0.795 ± 0.035 | 0.498 ± 0.021 | 0.649 ± 0.015 | 0.644 ± 0.014 | **0.165 ± 0.012** | 0.230 ± 0.010 | ————— |
| | GBM | 0.852 ± 0.035 | 0.287 ± 0.007 | 0.669 ± 0.012 | 0.662 ± 0.011 | 0.201 ± 0.013 | 0.259 ± 0.011 | ————— |
| | RSF | 0.926 ± 0.069 | 0.283 ± 0.010 | 0.653 ± 0.013 | 0.643 ± 0.012 | 0.192 ± 0.015 | 0.283 ± 0.027 | ————— |
| | $\mathcal{L}^{\text{Post}}_{\text{ALD+Cal}}$ | 1.821 ± 0.138 | 0.301 ± 0.017 | 0.655 ± 0.009 | 0.646 ± 0.010 | 0.185 ± 0.003 | 0.274 ± 0.009 | ————— |

| Dataset | Methods | MAE | IBS | Harrell's C-index | Uno's C-index | Average Calibration | Group Calibration | Individual Calibration |
|---|---|---|---|---|---|---|---|---|
| TMBImmuno | ALD | 1.523 ± 0.070 | 0.239 ± 0.005 | 0.559 ± 0.020 | **0.551 ± 0.016** | 0.228 ± 0.010 | 0.275 ± 0.015 | ———— |
| | CQRNN | 0.965 ± 0.026 | 0.246 ± 0.010 | 0.546 ± 0.013 | 0.546 ± 0.021 | 0.229 ± 0.010 | 0.286 ± 0.016 | ———— |
| | LogNorm | 1.747 ± 0.044 | 0.421 ± 0.007 | 0.554 ± 0.018 | 0.543 ± 0.027 | 0.241 ± 0.012 | 0.261 ± 0.026 | ———— |
| | DeepSurv | 0.915 ± 0.023 | 0.390 ± 0.009 | 0.538 ± 0.021 | 0.527 ± 0.013 | **0.196 ± 0.010** | 0.250 ± 0.010 | ———— |
| | DSM(Weibull) | 1.016 ± 0.017 | 0.246 ± 0.006 | 0.547 ± 0.016 | 0.537 ± 0.018 | 0.233 ± 0.009 | 0.249 ± 0.020 | ———— |
| | DSM(LogNorm) | 0.953 ± 0.017 | 0.245 ± 0.006 | 0.511 ± 0.029 | 0.521 ± 0.024 | 0.233 ± 0.009 | 0.251 ± 0.022 | ———— |
| | DeepHit | 1.148 ± 0.134 | 0.398 ± 0.003 | 0.558 ± 0.022 | 0.551 ± 0.024 | 0.239 ± 0.013 | 0.264 ± 0.022 | ———— |
| | GBM | **0.878 ± 0.019** | 0.241 ± 0.007 | 0.573 ± 0.020 | 0.549 ± 0.013 | 0.219 ± 0.009 | 0.244 ± 0.020 | ———— |
| | RSF | 1.656 ± 0.043 | 0.268 ± 0.009 | 0.539 ± 0.017 | 0.530 ± 0.020 | 0.215 ± 0.010 | 0.254 ± 0.022 | ———— |
| | $\mathcal{L}_{\text{ALD+Cal}}^{\text{Post}}$ | 1.611 ± 0.364 | **0.225 ± 0.010** | **0.574 ± 0.010** | 0.546 ± 0.009 | 0.214 ± 0.010 | **0.242 ± 0.014** | ———— |
| BreastMSK | ALD | 2.494 ± 0.234 | 0.085 ± 0.003 | 0.620 ± 0.040 | 0.567 ± 0.052 | 0.272 ± 0.026 | 0.286 ± 0.031 | ———— |
| | CQRNN | 1.571 ± 0.130 | 0.150 ± 0.011 | 0.615 ± 0.055 | 0.567 ± 0.062 | 0.289 ± 0.004 | 0.310 ± 0.006 | ———— |
| | LogNorm | 6.469 ± 0.239 | 0.312 ± 0.018 | 0.602 ± 0.038 | 0.551 ± 0.046 | 0.275 ± 0.011 | 0.298 ± 0.018 | ———— |
| | DeepSurv | 1.627 ± 0.101 | 0.337 ± 0.018 | 0.618 ± 0.038 | 0.563 ± 0.058 | 0.275 ± 0.025 | 0.292 ± 0.031 | ———— |
| | DSM(Weibull) | 1.615 ± 0.071 | 0.097 ± 0.002 | 0.630 ± 0.018 | 0.550 ± 0.026 | 0.300 ± 0.012 | 0.324 ± 0.012 | ———— |
| | DSM(LogNorm) | 1.580 ± 0.078 | 0.095 ± 0.001 | 0.631 ± 0.018 | 0.538 ± 0.017 | 0.296 ± 0.014 | 0.321 ± 0.015 | ———— |
| | DeepHit | **1.517 ± 0.107** | 0.305 ± 0.019 | 0.619 ± 0.036 | 0.542 ± 0.049 | 0.267 ± 0.014 | 0.286 ± 0.018 | ———— |
| | GBM | 1.586 ± 0.052 | 0.092 ± 0.001 | **0.638 ± 0.036** | 0.571 ± 0.028 | 0.283 ± 0.017 | 0.303 ± 0.017 | ———— |
| | RSF | 1.679 ± 0.159 | 0.087 ± 0.003 | 0.630 ± 0.033 | 0.561 ± 0.020 | 0.273 ± 0.024 | 0.306 ± 0.027 | ———— |
| | $\mathcal{L}_{\text{ALD+Cal}}^{\text{Post}}$ | 2.381 ± 0.186 | **0.084 ± 0.006** | 0.623 ± 0.031 | **0.621 ± 0.030** | **0.246 ± 0.011** | **0.272 ± 0.010** | ———— |
| LGGGBM | ALD | 1.255 ± 0.334 | 0.106 ± 0.011 | **0.786 ± 0.020** | 0.728 ± 0.034 | 0.173 ± 0.050 | 0.348 ± 0.034 | ———— |
| | CQRNN | 0.679 ± 0.102 | 0.193 ± 0.015 | 0.784 ± 0.020 | 0.752 ± 0.021 | 0.180 ± 0.024 | 0.372 ± 0.021 | ———— |
| | LogNorm | 1.052 ± 0.097 | 0.399 ± 0.012 | 0.785 ± 0.017 | 0.727 ± 0.045 | 0.180 ± 0.040 | 0.293 ± 0.027 | ———— |
| | DeepSurv | 0.812 ± 0.100 | 0.485 ± 0.024 | 0.722 ± 0.048 | 0.657 ± 0.049 | 0.187 ± 0.050 | 0.362 ± 0.037 | ———— |
| | DSM(Weibull) | 1.073 ± 0.094 | 0.176 ± 0.014 | 0.768 ± 0.026 | 0.727 ± 0.035 | 0.242 ± 0.016 | 0.335 ± 0.009 | ———— |
| | DSM(LogNorm) | 0.989 ± 0.091 | 0.173 ± 0.013 | 0.585 ± 0.041 | 0.618 ± 0.057 | 0.245 ± 0.019 | 0.351 ± 0.016 | ———— |
| | DeepHit | 1.917 ± 0.155 | 0.382 ± 0.026 | 0.772 ± 0.027 | 0.726 ± 0.036 | 0.263 ± 0.024 | 0.317 ± 0.033 | ———— |
| | GBM | **0.639 ± 0.078** | 0.141 ± 0.011 | 0.767 ± 0.010 | 0.731 ± 0.032 | 0.205 ± 0.025 | 0.302 ± 0.026 | ———— |
| | RSF | 0.912 ± 0.255 | 0.115 ± 0.011 | 0.774 ± 0.022 | 0.728 ± 0.023 | 0.190 ± 0.031 | 0.359 ± 0.069 | ———— |
| | $\mathcal{L}_{\text{ALD+Cal}}^{\text{Post}}$ | 0.781 ± 0.181 | **0.103 ± 0.017** | 0.781 ± 0.039 | **0.763 ± 0.047** | **0.118 ± 0.023** | **0.257 ± 0.037** | ———— |

### C.4 CASE STUDIES

**Case Study I: Discretization, Crossing Quantiles, and Distribution Mismatch**

Fig. 5 illustrates a representative example comparing *nonparametric* and *parametric* ALD models. In the *nonparametric* approach (based on the *quantile form*, *i.e.*, $\mathcal{AL}(\theta, \sigma, q)$), only a fixed set of quantile percentages $\{q_i\}_{i=1}^{K}$ are estimated independently. Due to this *discretization*, the cumulative distribution function (CDF) appears as a piecewise curve, with substantial gaps between neighboring quantiles. As shown in the figure, this leads to poor resolution in the distribution tail and visible *quantile crossing*, where estimates at higher quantile levels fall below those at lower ones.

In contrast, the *parametric* ALD-based approach (based on the *asymmetry form*, *i.e.*, $\mathcal{AL}(\theta, \sigma, \kappa)$) provides a continuous, smooth estimate of the conditional distribution. While this results in better global coherence and eliminates quantile crossing, the model can suffer from distribution mismatch, especially in the distribution tails. In this case, the ALD fit systematically underestimates the upper tail, failing to capture the observed data spread. This illustrates the challenge of using a single ALD to model highly skewed or heavy-tailed distributions.

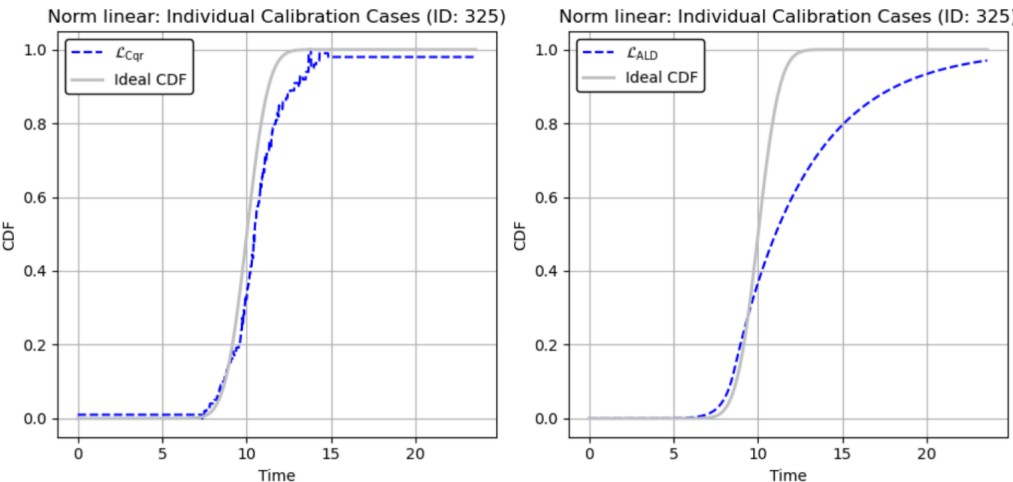

Figure 5: Illustration of limitations in *nonparametric* (left) and *parametric* (right) ALD approaches. Left: *Discretized* and *crossing quantiles* issues from *nonparametric* modeling. Right: *Distribution mismatch* issue in the *parametric* ALD-based model.

**Case Study II: Overfitting and Asynchronous Convergence**

While Theorem 2 demonstrates that increasing the number of quantile samples improves the Monte Carlo approximation and enhances individual calibration, this benefit must be carefully balanced in practice. Sampling a larger number of quantile percentages typically necessitates longer training, but prolonged training can lead to overfitting, particularly on datasets with limited size or high noise (*e.g.*, `LogNorm`-based datasets). As shown in Fig. 6, training for 2000 epochs results in degraded calibration performance due to such overfitting effects. Specifically, while the training loss $\mathcal{L}_{\text{ALD+Cal}}^{\text{Train}}$ continues to decrease, both the test loss $\mathcal{L}_{\text{ALD+Cal}}^{\text{Test}}$ and test negative log-likelihood (NLL) $\mathcal{L}_{\text{ALD}}^{\text{Test}}$ begin to increase steadily after a certain point. This divergence indicates that the model starts to fit noise in the training data, thereby impairing its generalization capability. These observations underscore the importance of incorporating early stopping to maintain a proper trade-off between calibration quality and generalization performance. Fig. 7 presents a case from the `LogNorm_med` dataset that illustrates the issue of *asynchronous convergence* encountered during training with the *pre-calibration* objectives in Equation 10 and Equation 12. This phenomenon typically arises in the early stages of training, when the model has not yet learned a meaningful approximation of the underlying distribution. At this point, applying the additional loss (*i.e.*, $\mathcal{L}_{\text{Cal}}$ or $\mathcal{L}_{\text{Cqr}}$) too early may introduce noisy or conflicting gradient signals that interfere with stable optimization. Specifically, the negative log-likelihood (NLL) loss $\mathcal{L}_{\text{ALD}}$ encourages the model to fit the global structure of the distribution, while the calibration loss enforces local alignment at specific quantile levels. When the distributional

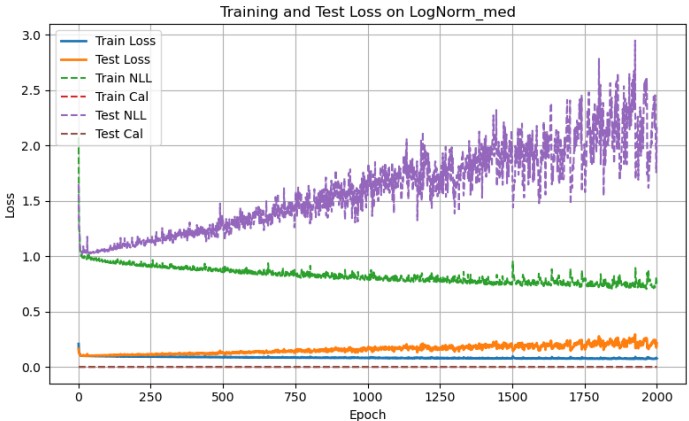

Figure 6: Training dynamics on the `LogNorm_med` dataset with 2000 training epochs. *Overfitting* is evident in the increasing test negative log-likelihood (NLL) $\mathcal{L}_{\text{ALD}}$ and test loss curves, despite the stability of CAL loss ($\mathcal{L}_{\text{Cal}}$), highlighting the risk of prolonged training under joint loss objectives.

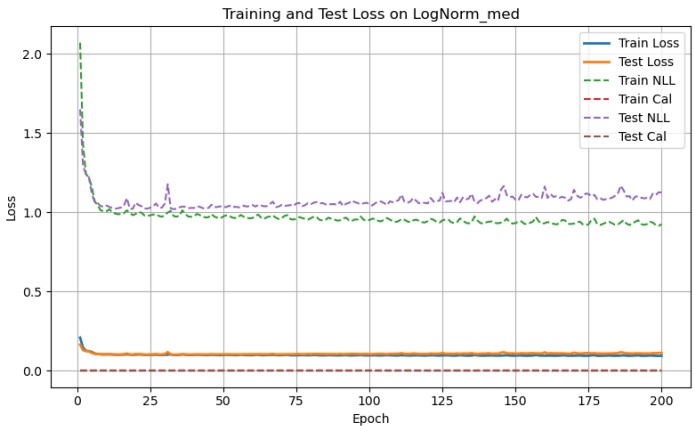

(a) *Pre-calibration* without *warm-up calibration*.

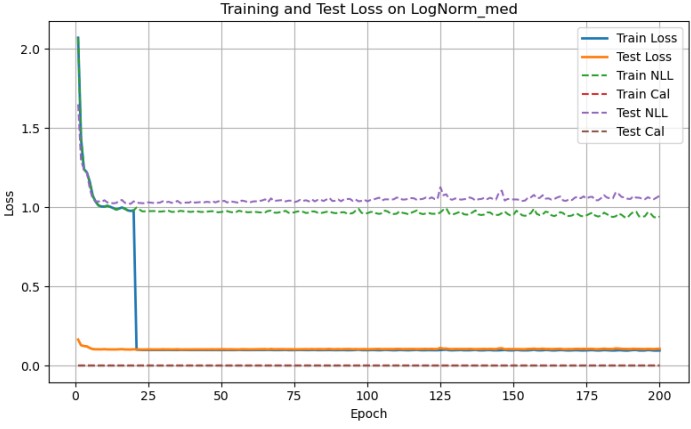

(b) *Pre-calibration* with *warm-up calibration*.

Figure 7: Comparison of *pre-calibration* training dynamics on the `LogNorm_med` dataset, with and without the proposed *warm-up calibration* strategy.

parameters are still unstable, such localized supervision may act more as noise than constructive guidance, ultimately impeding convergence.

To address this, we adopt the proposed *warm-up calibration strategy*, wherein training initially focuses solely on the NLL loss $\mathcal{L}_{\text{ALD}}$. The additional loss (*i.e.*, $\mathcal{L}_{\text{Cal}}$ or $\mathcal{L}_{\text{Cqr}}$) is then gradually incorporated after a fixed number of epochs, allowing the model to first establish a stable approximation of the distribution. As shown in Fig. 7(b), this strategy can stabilize training dynamics and improve calibration consistency. Notably, the final test NLL is lower when using *warm-up calibration strategy* compared to direct *pre-calibration*, demonstrating improved generalization and more effective distribution fitting.

In contrast, the *post-calibration* ICALD models entirely bypass the issue of asynchronous convergence by applying calibration as a post-processing step after the base model has been trained. This decoupled approach yields a more stable and consistent calibration effect, free from the gradient conflicts introduced by joint training losses. As shown in Table 2 and Table 7, *post-calibration* ICALD models consistently achieve strong *calibration* performance across all metrics. These results underscore the practical advantage of *post-calibration* in improving reliability, particularly on challenging datasets like `LogNorm`, which are more susceptible to instability under joint-loss training schemes.

**Case Study III: Interpretability Case Study via SHAP**

We provide an interpretability case study to address the concern that introducing the quantile variable $q$ may hinder factor-level insights. Although ICALD increases modeling flexibility through the continuous mixture mechanism, we can interpret predictions through scalar functionals of the mixed distribution (*e.g.*, the mixed-ALD mean survival time), and then apply standard explanation tools such as SHAP (Lundberg & Lee, 2017) to these scalar targets.

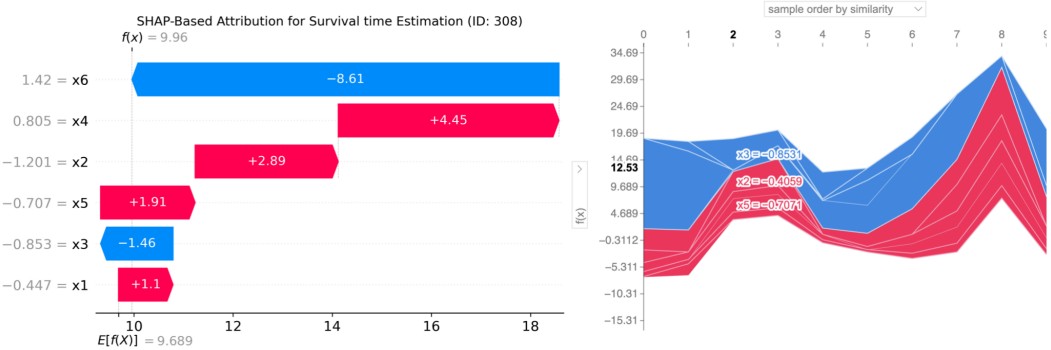

Figure 8: **SHAP-based interpretability for ICALD.** *Left:* Individual-level SHAP waterfall explanation for a representative test instance (ID: 308) on the predicted survival-time summary (mixed-ALD mean). *Right:* Cohort-style visualization (samples ordered by similarity) illustrating how feature contributions vary across individuals, enabling subgroup and dynamic analyses.

**Individual-level explanation.** Figure 8 (left) shows a SHAP waterfall plot for a representative test subject (ID: 308), where the explained target is the predicted survival time summary (here, the mixed-ALD mean). Starting from the population baseline $E[f(X)] \approx 9.689$, the model prediction is $f(x) \approx 9.96$. Feature $x_6$ exerts the largest negative contribution ($-8.61$), indicating that this covariate strongly shifts the prediction towards shorter survival. This effect is partially offset by several positive contributors, notably $x_4$ ($+4.45$), $x_2$ ($+2.89$), and $x_5$ ($+1.91$), with smaller adjustments from $x_3$ ($-1.46$) and $x_1$ ($+1.10$). Overall, the explanation reveals a clear "push–pull" structure: despite a dominant risk-increasing factor ($x_6$), compensating protective factors ($x_4, x_2, x_5$) recover the final prediction close to (and slightly above) the baseline. This illustrates that the model remains interpretable at the factor level even with the additional quantile input.

**Cohort-level and dynamic insights.** Beyond a single prediction, we can aggregate SHAP explanations across a cohort to obtain population-level summaries. For example, we can compute the mean absolute SHAP value per feature to quantify global importance, and we can stratify the test set by predicted risk (*e.g.*, low/medium/high risk groups defined by the predicted median survival time)

to examine how the ranking of covariate contributions changes across subpopulations. Moreover, because the mixed-ALD CDF admits a closed-form expression (Appendix A.3), we can interpret time-dependent targets such as $S(t|\mathbf{x})$ at multiple horizons; repeating SHAP on $S(t|\mathbf{x})$ for a grid of $t$ yields dynamic visualizations that reveal how covariate effects evolve from short-term to long-term risk. Figure 8 (right) provides an example of such a cohort-style visualization for samples ordered by similarity, illustrating that the dominant contributors (*e.g.*. $x_2$, $x_3$, $x_5$ in this example) can vary in magnitude and direction between nearby individuals, allowing nuanced subgroup analyzes.

**Case Study IV: Calibration Performance with $\mathcal{L}^{\mathbf{Pre}}_{\mathbf{ALD+Cal}}$ *vs.* $\mathcal{L}^{\mathbf{Pre}}_{\mathbf{ALD+Cqr}}$**

Although both $\mathcal{L}^{\mathrm{Pre}}_{\mathrm{ALD+Cal}}$ and $\mathcal{L}^{\mathrm{Pre}}_{\mathrm{ALD+Cqr}}$ are theoretically grounded in improving *individual calibration* (see Definition 1, Definition 2, and Theorem 1), their empirical performance differs markedly, likely as a result of how they incorporate censored data and define their loss objectives (see Table 9).

Table 9: Pairwise comparison of calibration performance between $\mathcal{L}^{\mathrm{Pre}}_{\mathrm{ALD+Cal}}$ and $\mathcal{L}^{\mathrm{Pre}}_{\mathrm{ALD+Cqr}}$ across 21 datasets. The two sub-columns reflect settings with and without censored data. Each group reports the number of datasets where $\mathcal{L}^{\mathrm{Pre}}_{\mathrm{ALD+Cal}}$ performs **better**, **worse**, or the **same**. The final two rows show total counts and proportions across 56 pairwise comparisons.

| Metric | Train with censored data | | | Train without censored data | | |
|---|---|---|---|---|---|---|
| *Average Calibration* | 13 | 0 | 8 | 1 | 1 | 19 |
| *Group Calibration* | 14 | 0 | 7 | 6 | 2 | 13 |
| *Individual Calibration* | 9 | 0 | 5 | 1 | 1 | 12 |
| **Total** | 36 | 0 | 20 | 8 | 4 | 44 |
| **Proportion (%)** | 64.3 | 0.0 | 35.7 | 14.3 | 7.1 | 78.6 |

In the left half of Table 9, we compare the two methods under the standard setting where censored data is retained during training, and the quantile regression loss is modified using the Portnoy estimator (Portnoy, 2003), as defined in Equation equation 10. Under this setting, $\mathcal{L}^{\mathrm{Pre}}_{\mathrm{ALD+Cal}}$ exhibits a clear advantage: it outperforms $\mathcal{L}^{\mathrm{Pre}}_{\mathrm{ALD+Cqr}}$ in 64.3% of all comparisons and never underperforms.

To assess whether this performance gap stems from the presence of censoring or the estimator itself, we repeat the comparison using the same datasets but exclude all censored samples during training (right half of Table 9). In this censored-free scenario, the two methods perform comparably in 78.6% of the cases, with only marginal advantages observed on either side. This indicates that, in the absence of censoring, $\mathcal{L}^{\mathrm{Pre}}_{\mathrm{ALD+Cal}}$ and $\mathcal{L}^{\mathrm{Pre}}_{\mathrm{ALD+Cqr}}$ behave similarly in terms of *calibration*, and suggests that the Portnoy-adjusted loss in $\mathcal{L}^{\mathrm{Pre}}_{\mathrm{ALD+Cqr}}$ may introduce bias or instability when censoring is present.

In summary, these results highlight the sensitivity of $\mathcal{L}^{\mathrm{Pre}}_{\mathrm{ALD+Cqr}}$ to the censoring mechanism. Its reliance on the Portnoy estimator may limit its effectiveness in capturing calibration signals under censored conditions. In contrast, $\mathcal{L}^{\mathrm{Pre}}_{\mathrm{ALD+Cal}}$ appears to offer a more robust and stable calibration objective when applied to censored survival data. Notably, a similar trend is observed as well in the *post-calibration* setting.

**Case Study V: Sensitivity to the loss weight $\lambda$ and warm-up length $L$**

**Setup.** We study the sensitivity of ICALD ($\mathcal{L}_{\text{ALD+Cal}}^{\text{Pre}}$) to the calibration loss weight $\lambda \in \{0.1, 0.3, 0.5, 0.7, 0.9\}$ and the warm-up length $L \in \{50, 100, 200, 400\}$ (with the default $L=200$). In Tables 10 and 11, each cell reports the percentage of datasets for which the setting on the left outperforms its comparator. Percentages are *not statistically significant* by Student's $t$-test.

Table 10: Comparison across different $\lambda$ values (default $\lambda=0.1$). Each column shows the percentage of datasets where $\lambda=0.1$ outperforms the comparator $\lambda \in \{0.3, 0.5, 0.7, 0.9\}$ for each metric (higher is better for the comparison rate).

| Metric | 0.1 vs 0.3 | 0.1 vs 0.5 | 0.1 vs 0.7 | 0.1 vs 0.9 |
|---|---|---|---|---|
| MAE | 52.4% | 61.9% | 66.7% | 57.1% |
| IBS | 61.9% | 81.0% | 76.2% | 66.7% |
| Harrell's C-Index | 76.2% | 76.2% | 76.2% | 71.4% |
| Uno's C-Index | 71.4% | 81.0% | 76.2% | 66.7% |
| Average Calibration | 57.1% | 57.1% | 57.1% | 47.6% |
| Group Calibration | 42.9% | 33.3% | 47.6% | 47.6% |
| Individual Calibration | 50.0% | 57.1% | 57.1% | 50.0% |
| **Total** | 59.3% | 64.3% | 65.7% | 58.6% |

Table 11: Comparison across different warm-up lengths (default $L=200$). Each column shows the percentage of datasets where $L=200$ outperforms the comparator $L \in \{50, 100, 400\}$ for each metric (higher is better for the comparison rate).

| Metric | 200 vs 50 | 200 vs 100 | 200 vs 400 |
|---|---|---|---|
| MAE | 57.1% | 57.1% | 57.1% |
| IBS | 52.4% | 52.4% | 52.4% |
| Harrell's C-Index | 57.1% | 57.1% | 61.9% |
| Uno's C-Index | 52.4% | 52.4% | 57.1% |
| Average Calibration | 66.7% | 57.1% | 66.7% |
| Group Calibration | 66.7% | 47.6% | 66.7% |
| Individual Calibration | 71.4% | 57.1% | 50.0% |
| **Total** | 60.5% | 54.4% | 58.8% |

From Tables 10 and 11, we can draw the following key observations:

1. **Overall robustness.** ICALD is generally robust to variations in both $\lambda$ and $L$. While differences are not statistically significant, the average trends (over five runs per dataset) show mild variations.

2. **Effect of $\lambda$.** As $\lambda$ increases, calibration-oriented metrics tend to improve, suggesting stronger distributional calibration at the cost of slight trade-offs in MAE, IBS, and C-indices.

3. **Effect of warm-up length $L$.** Changing $L$ has limited impact on overall performance. Metrics remain stable across $L \in \{50, 100, 200, 400\}$, likely because the calibration loss continues contributing during the post-calibration phase.

**Case Study VI: Impact of the $q$-dimensionality in pre-calibration**

**Setup.** To enhance the expressiveness of the calibration anchor, we explore different choices of the $q$-dimensionality in the pre-calibration model $\mathcal{L}_{\text{ALD+Cal}}^{\text{Pre}}$. Specifically, we consider $d \in \{1, 2, 4, 8\}$ and evaluate whether increasing the dimension of $q$ improves calibration performance. In Table 12, each cell reports the number of datasets where $d = 4$ is significantly better / worse / the same as its comparator, with statistical significance assessed by Student's $t$-test ($p < 0.05$, after FDR correction).

Table 12 shows that increasing the dimension of $q$ (*e.g.*, $d = 2, 4$) generally leads to improved calibration performance. More specifically,

Table 12: Comparison under different $q$ dimensions in pre-calibration. Each cell shows the number of datasets where $d = 4$ is significantly better / worse / the same as the comparison.

| Metric | $d = 4$ vs $d = 1$ | $d = 4$ vs $d = 2$ | $d = 4$ vs $d = 8$ |
|---|---|---|---|
| Average Calibration | 1 / 0 / 20 | 0 / 0 / 21 | 2 / 0 / 19 |
| Group Calibration | 1 / 0 / 20 | 0 / 0 / 21 | 2 / 0 / 19 |
| Individual Calibration | 0 / 0 / 14 | 0 / 0 / 14 | 2 / 0 / 12 |
| **Total** | 2 / 0 / 54 | 0 / 0 / 56 | 6 / 0 / 50 |
| **Proportion (%)** | 3.6 / 0.0 / 96.4 | 0.0 / 0.0 / 100.0 | 10.7 / 0.0 / 89.3 |

1. **For $d = 4$ vs. $d = 1$.** The most significant improvements are observed on the `SUPPORT` dataset (where the feature dimension is 14), suggesting that higher $q$-dimensionality is particularly beneficial when the input features are high-dimensional.

2. **For $d = 4$ vs. $d = 8$.** Significant improvements are found for the `Exponential` and `LogNorm` datasets, both with $x$ of dimension 1. This indicates that increasing $q$-dimensionality beyond a certain point does not necessarily yield additional benefits and may even degrade performance in lower-dimensional settings.

3. **For $d = 4$ vs. $d = 2$.** Although there are no statistically significant differences, the average metric (mean over five runs) is slightly better for $d = 4$ across 21 datasets.

In summary, our results suggest that increasing the $q$-dimensionality to a moderate level (*e.g.*, $d = 2, 4$) generally improves calibration, especially for datasets with higher feature dimensionality.

**Case Study VII: Granular Analysis of Calibration and Accuracy**

We performed an additional analysis on the synthetic `Gaussian_uniform` dataset to assess how calibration and accuracy behave across different time horizons and quantile levels. We stratified test instances by the CDF of the true event time $F_T(t)$ into four equal-probability bins, $[0, 0.25)$, $[0.25, 0.5)$, $[0.5, 0.75)$, and $[0.75, 1]$, which correspond to early, mid-early, mid-late and late event times. The results are shown in Figures 9 and 10, where error bars indicate one standard deviation

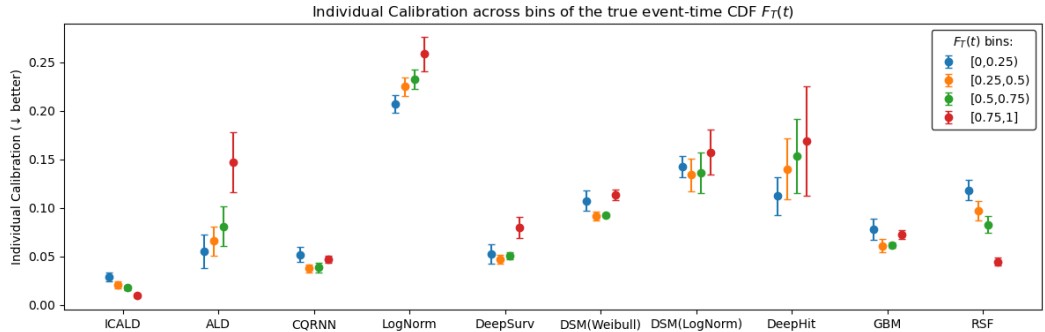

Figure 9: Individual Calibration across bins of the true event-time CDF $F_T(t)$.

over 5 independent runs. From these results, we observe that: ($i$) ICALD consistently achieves the lowest individual calibration error across all $F_T(t)$ bins while maintaining competitive or superior accuracy (IBS) in each bin; ($ii$) ICALD exhibits very small standard deviations across runs, indicating that its performance is highly robust; and ($iii$) ICALD performs particularly well in the late-event regime $F_T(t) \in [0.75, 1]$, providing both better calibration and better accuracy than competing methods.

Overall, this stratified analysis confirms that ICALD maintains strong calibration and accuracy over extended time horizons and across different quantile levels.

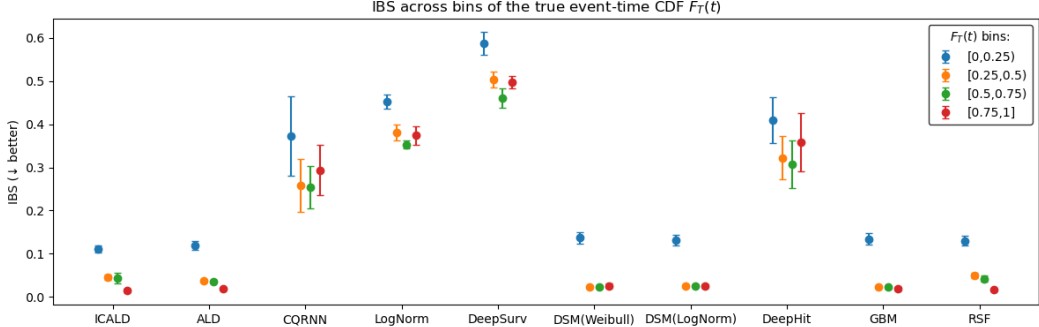

Figure 10: IBS across bins of the true event-time CDF $F_T(t)$.

**Case Study VIII: Scalability and Cross-Domain Evaluation**

A common concern for calibration-focused survival models is whether their reliability gains persist in *challenging* regimes that depart from the standard clinical benchmarks—in particular, (*i*) *high-dimensional* settings with limited sample sizes (large $d$, small $n$), and (*ii*) *non-medical* domains where censoring patterns and feature distributions can differ substantially from healthcare data. To partially address these external validity and scalability concerns, we conduct additional experiments on two widely used high-dimensional gene-expression datasets and four datasets from finance and engineering domains.

**High-dimensional gene-expression datasets.** We consider: (*i*) DBCD (Dutch Breast Cancer Dataset, Rosenwald et al. 2002), which contains $n = 295$ patients with $d = 4919$ gene-expression features and a censoring rate of $73.2\%$; and (*ii*) DLBCL (Diffuse Large B-cell Lymphoma, Van Houwelingen et al. 2006), which contains $n = 240$ patients with $d = 7399$ gene-expression features and a censoring rate of $42.5\%$. These datasets are representative of a difficult regime for survival modeling: the feature dimension is orders of magnitude larger than the sample size, while censoring can be substantial, making both accurate distribution fitting and reliable uncertainty quantification nontrivial. As shown in Table 13, $\mathcal{L}_{\text{ALD+Cal}}^{\text{Post}}$ achieves competitive or superior overall performance on both datasets, with particularly strong results on *calibration* (*average* and *group calibration*) while maintaining competitive *predictive accuracy* (MAE/IBS) and *concordance*. This suggests that the post-calibrated ICALD framework remains effective even when the backbone must operate under severe $d \gg n$ constraints.

**Beyond the medical domain: finance and engineering.** To further evaluate generalizability, we additionally test on four datasets from the PySurvival package (finance and engineering applications). Table 14 reports the full results. Across these datasets, $\mathcal{L}_{\text{ALD+Cal}}^{\text{Post}}$ continues to deliver competitive or superior performance, and the advantage is again most consistent on *calibration* metrics. Notably, this behavior holds even when strong ensemble-based or neural baselines achieve high *concordance*, indicating that ICALD can improve the *reliability* of predicted survival distributions without sacrificing overall ranking performance.

Overall, these additional experiments support the practical robustness of ICALD under (*i*) high-dimensional, small-sample regimes and (*ii*) cross-domain shifts, and reinforce our central claim that the proposed calibration mechanism yields reliable survival distribution estimates in settings beyond the standard clinical benchmarks.

Table 13: Full results table for the high-dimensional datasets, methods, and metrics. The values represent the mean ± 1 standard error for the test set over 5 runs.

| Dataset | Methods | MAE | IBS | Harrell's C-index | Uno's C-index | Average Calibration | Group Calibration |
|---|---|---|---|---|---|---|---|
| DBCD | ALD | 0.782 ± 0.162 | 0.188 ± 0.035 | 0.648 ± 0.114 | 0.663 ± 0.054 | 0.239 ± 0.021 | 0.689 ± 0.236 |
| | CQRNN | 1.539 ± 0.281 | 0.480 ± 0.044 | 0.688 ± 0.062 | 0.628 ± 0.051 | 0.324 ± 0.074 | 0.698 ± 0.128 |
| | LogNorm | 1.129 ± 0.137 | 0.343 ± 0.063 | 0.681 ± 0.028 | 0.668 ± 0.086 | 0.193 ± 0.046 | 0.608 ± 0.135 |
| | DeepSurv | 1.128 ± 0.154 | 0.503 ± 0.057 | 0.572 ± 0.051 | 0.541 ± 0.070 | 0.426 ± 0.045 | 0.556 ± 0.089 |
| | DSM(Weibull) | 3.263 ± 0.349 | 0.143 ± 0.020 | 0.676 ± 0.051 | 0.662 ± 0.074 | 0.351 ± 0.021 | 0.499 ± 0.002 |
| | DSM(LogNorm) | 3.341 ± 0.195 | 0.150 ± 0.021 | 0.688 ± 0.053 | 0.647 ± 0.084 | 0.370 ± 0.014 | 0.498 ± 0.002 |
| | DeepHit | 1.129 ± 0.137 | 0.343 ± 0.063 | 0.681 ± 0.028 | 0.668 ± 0.086 | 0.193 ± 0.046 | 0.608 ± 0.135 |
| | GBM | 2.949 ± 0.198 | 0.146 ± 0.018 | 0.583 ± 0.046 | 0.594 ± 0.060 | 0.343 ± 0.031 | 0.513 ± 0.035 |
| | RSF | 2.225 ± 0.547 | 0.133 ± 0.019 | 0.660 ± 0.081 | 0.635 ± 0.073 | 0.321 ± 0.044 | 0.503 ± 0.022 |
| | $\mathcal{L}^{Post}_{ALD+Cal}$ | **0.774 ± 0.142** | **0.129 ± 0.033** | **0.695 ± 0.062** | **0.707 ± 0.063** | **0.185 ± 0.060** | **0.474 ± 0.128** |
| DLBCL | ALD | 1.161 ± 0.903 | 0.466 ± 0.057 | 0.589 ± 0.034 | 0.533 ± 0.075 | 0.215 ± 0.039 | 0.419 ± 0.040 |
| | CQRNN | 1.192 ± 0.060 | 0.711 ± 0.077 | 0.607 ± 0.010 | 0.545 ± 0.127 | 0.233 ± 0.083 | 0.444 ± 0.036 |
| | LogNorm | 2.894 ± 0.983 | 0.812 ± 0.079 | 0.554 ± 0.071 | 0.620 ± 0.106 | 0.239 ± 0.043 | 0.435 ± 0.020 |
| | DeepSurv | 1.082 ± 0.489 | 0.743 ± 0.089 | 0.546 ± 0.022 | 0.579 ± 0.084 | 0.254 ± 0.071 | 0.442 ± 0.039 |
| | DSM(Weibull) | 1.038 ± 0.078 | 0.367 ± 0.033 | 0.601 ± 0.059 | 0.532 ± 0.138 | 0.259 ± 0.024 | 0.373 ± 0.018 |
| | DSM(LogNorm) | **0.911 ± 0.069** | 0.361 ± 0.033 | 0.592 ± 0.035 | 0.526 ± 0.146 | 0.248 ± 0.029 | 0.372 ± 0.032 |
| | DeepHit | 1.129 ± 0.137 | 0.343 ± 0.063 | **0.681 ± 0.028** | **0.668 ± 0.086** | 0.193 ± 0.046 | 0.608 ± 0.135 |
| | GBM | 0.587 ± 0.142 | 0.336 ± 0.039 | 0.597 ± 0.034 | 0.584 ± 0.031 | 0.187 ± 0.030 | 0.345 ± 0.038 |
| | RSF | 0.925 ± 0.310 | 0.345 ± 0.047 | 0.607 ± 0.064 | 0.625 ± 0.110 | 0.188 ± 0.034 | 0.325 ± 0.041 |
| | $\mathcal{L}^{Post}_{ALD+Cal}$ | 1.059 ± 0.057 | **0.246 ± 0.037** | 0.605 ± 0.038 | 0.585 ± 0.119 | **0.138 ± 0.031** | **0.284 ± 0.040** |

Table 14: Full results table for the finance and engineering datasets, methods, and metrics. The values represent the mean ± 1 standard error for the test set over 5 runs.

| Dataset | Methods | MAE | IBS | Harrell's C-index | Uno's C-index | Average Calibration | Group Calibration |
|---|---|---|---|---|---|---|---|
| Churn | ALD | 0.404 ± 0.023 | 0.063 ± 0.009 | **0.875 ± 0.009** | **0.911 ± 0.009** | 0.064 ± 0.015 | 0.135 ± 0.051 |
| | CQRNN | 0.397 ± 0.021 | 0.505 ± 0.019 | 0.873 ± 0.013 | 0.902 ± 0.017 | 0.077 ± 0.014 | 0.157 ± 0.046 |
| | LogNorm | 0.480 ± 0.031 | 0.793 ± 0.047 | 0.828 ± 0.011 | 0.829 ± 0.032 | 0.147 ± 0.012 | 0.211 ± 0.024 |
| | DeepSurv | 0.397 ± 0.025 | 0.713 ± 0.033 | 0.703 ± 0.038 | 0.652 ± 0.036 | 0.053 ± 0.009 | **0.099 ± 0.035** |
| | DSM(Weibull) | 1.257 ± 0.048 | 0.237 ± 0.022 | 0.722 ± 0.015 | 0.702 ± 0.043 | 0.274 ± 0.009 | 0.304 ± 0.012 |
| | DSM(LogNorm) | 1.211 ± 0.050 | 0.237 ± 0.021 | 0.593 ± 0.040 | 0.610 ± 0.036 | 0.280 ± 0.008 | 0.315 ± 0.007 |
| | DeepHit | 0.429 ± 0.019 | 0.833 ± 0.041 | 0.845 ± 0.018 | 0.877 ± 0.011 | 0.079 ± 0.012 | 0.114 ± 0.012 |
| | GBM | 0.978 ± 0.075 | 0.180 ± 0.016 | 0.818 ± 0.020 | 0.816 ± 0.027 | 0.183 ± 0.008 | 0.221 ± 0.011 |
| | RSF | 0.513 ± 0.047 | 0.092 ± 0.016 | 0.853 ± 0.016 | 0.833 ± 0.036 | 0.062 ± 0.011 | 0.164 ± 0.044 |
| | $\mathcal{L}^{Post}_{ALD+Cal}$ | **0.337 ± 0.181** | **0.041 ± 0.018** | 0.860 ± 0.009 | 0.879 ± 0.023 | **0.048 ± 0.016** | 0.119 ± 0.033 |
| CreditRisk | ALD | 0.782 ± 0.087 | 0.092 ± 0.009 | **0.722 ± 0.020** | 0.704 ± 0.035 | 0.159 ± 0.036 | 0.338 ± 0.017 |
| | CQRNN | 1.043 ± 0.065 | 0.255 ± 0.030 | 0.691 ± 0.043 | 0.693 ± 0.045 | 0.208 ± 0.031 | 0.417 ± 0.022 |
| | LogNorm | **0.720 ± 0.065** | 0.466 ± 0.022 | 0.660 ± 0.040 | 0.610 ± 0.041 | 0.197 ± 0.018 | 0.304 ± 0.080 |
| | DeepSurv | 1.194 ± 0.067 | 0.392 ± 0.020 | 0.660 ± 0.053 | 0.618 ± 0.039 | 0.158 ± 0.028 | 0.337 ± 0.036 |
| | DSM(Weibull) | 1.273 ± 0.075 | 0.107 ± 0.012 | 0.564 ± 0.019 | 0.512 ± 0.031 | 0.219 ± 0.019 | 0.374 ± 0.056 |
| | DSM(LogNorm) | 1.315 ± 0.086 | 0.106 ± 0.012 | 0.563 ± 0.009 | 0.604 ± 0.038 | 0.225 ± 0.018 | 0.376 ± 0.056 |
| | DeepHit | 1.081 ± 0.132 | 0.310 ± 0.016 | 0.698 ± 0.023 | 0.659 ± 0.064 | 0.142 ± 0.022 | 0.278 ± 0.030 |
| | GBM | 1.416 ± 0.067 | 0.101 ± 0.013 | 0.680 ± 0.031 | 0.649 ± 0.034 | 0.201 ± 0.021 | 0.336 ± 0.015 |
| | RSF | 1.363 ± 0.133 | 0.090 ± 0.013 | 0.706 ± 0.012 | **0.709 ± 0.014** | 0.172 ± 0.034 | 0.336 ± 0.031 |
| | $\mathcal{L}^{Post}_{ALD+Cal}$ | 1.339 ± 0.547 | **0.064 ± 0.011** | 0.699 ± 0.020 | 0.638 ± 0.021 | **0.137 ± 0.020** | 0.234 ± 0.071 |
| EmployeeAttrition | ALD | 0.184 ± 0.014 | **0.018 ± 0.002** | 0.933 ± 0.008 | 0.923 ± 0.006 | 0.111 ± 0.026 | 0.196 ± 0.044 |
| | CQRNN | 0.308 ± 0.028 | 0.384 ± 0.013 | 0.946 ± 0.007 | 0.936 ± 0.006 | 0.071 ± 0.033 | 0.181 ± 0.048 |
| | LogNorm | 2.446 ± 0.400 | 0.458 ± 0.009 | 0.915 ± 0.010 | 0.852 ± 0.012 | 0.289 ± 0.013 | 0.339 ± 0.014 |
| | DeepSurv | 0.278 ± 0.024 | 0.500 ± 0.008 | 0.795 ± 0.029 | 0.760 ± 0.026 | 0.131 ± 0.007 | 0.175 ± 0.017 |
| | DSM(Weibull) | 0.707 ± 0.066 | 0.048 ± 0.004 | 0.886 ± 0.013 | 0.815 ± 0.009 | 0.210 ± 0.016 | 0.291 ± 0.008 |
| | DSM(LogNorm) | 0.729 ± 0.070 | 0.051 ± 0.004 | 0.891 ± 0.010 | 0.789 ± 0.008 | 0.235 ± 0.016 | 0.323 ± 0.011 |
| | DeepHit | 0.288 ± 0.039 | 0.502 ± 0.006 | 0.736 ± 0.115 | 0.695 ± 0.121 | 0.126 ± 0.018 | 0.304 ± 0.046 |
| | GBM | 1.041 ± 0.015 | 0.058 ± 0.004 | 0.885 ± 0.006 | 0.824 ± 0.007 | 0.210 ± 0.002 | 0.316 ± 0.001 |
| | RSF | 0.250 ± 0.032 | 0.019 ± 0.001 | 0.944 ± 0.007 | **0.943 ± 0.005** | 0.368 ± 0.003 | 0.449 ± 0.010 |
| | $\mathcal{L}^{Post}_{ALD+Cal}$ | **0.175 ± 0.015** | **0.018 ± 0.003** | **0.954 ± 0.007** | 0.927 ± 0.004 | **0.056 ± 0.055** | 0.167 ± 0.071 |
| Maintenance | ALD | 0.079 ± 0.016 | 0.018 ± 0.002 | 0.957 ± 0.012 | 0.946 ± 0.021 | 0.204 ± 0.021 | 0.246 ± 0.019 |
| | CQRNN | 0.033 ± 0.004 | 0.065 ± 0.021 | 0.978 ± 0.003 | 0.976 ± 0.004 | **0.053 ± 0.012** | **0.133 ± 0.026** |
| | LogNorm | 1.129 ± 0.137 | 0.343 ± 0.063 | 0.681 ± 0.028 | 0.668 ± 0.086 | 0.193 ± 0.046 | 0.608 ± 0.135 |
| | DeepSurv | 0.019 ± 0.014 | 0.667 ± 0.016 | 0.975 ± 0.019 | 0.976 ± 0.019 | 0.168 ± 0.017 | 0.220 ± 0.011 |
| | DSM(Weibull) | 0.315 ± 0.015 | 0.045 ± 0.003 | 0.675 ± 0.026 | 0.656 ± 0.031 | 0.084 ± 0.010 | 0.304 ± 0.056 |
| | DSM(LogNorm) | 0.336 ± 0.017 | 0.046 ± 0.003 | 0.568 ± 0.026 | 0.555 ± 0.029 | 0.087 ± 0.011 | 0.313 ± 0.067 |
| | DeepHit | 0.023 ± 0.001 | 0.689 ± 0.017 | **0.994 ± 0.003** | **0.993 ± 0.003** | 0.226 ± 0.015 | 0.463 ± 0.017 |
| | GBM | 0.270 ± 0.018 | 0.035 ± 0.003 | 0.824 ± 0.024 | 0.838 ± 0.023 | 0.064 ± 0.010 | 0.313 ± 0.026 |
| | RSF | **0.013 ± 0.001** | **0.012 ± 0.000** | 0.992 ± 0.003 | 0.992 ± 0.003 | 0.365 ± 0.011 | 0.435 ± 0.007 |
| | $\mathcal{L}^{Post}_{ALD+Cal}$ | 0.044 ± 0.020 | 0.013 ± 0.004 | 0.969 ± 0.010 | 0.960 ± 0.018 | 0.075 ± 0.009 | 0.173 ± 0.039 |

