# OpenReview forum: "Learning Survival Distributions with Individually Calibrated Asymmetric Laplace Distribution"
_ICLR.cc/2026/Conference — ICLR 2026 Poster_

### Official Review · Reviewer_zNTr · 2025-10-30

**Soundness:** 2
**Presentation:** 2
**Contribution:** 1
**Rating:** 2
**Confidence:** 3

**Summary:**

The paper proposes a survival analysis framework that augments the most recent asymmetric Laplace distribution-based model (Sheng & Henao, 2025) with calibration-oriented loss functions. The proposed model is demonstrated to have a profitable property of monotonic PAIC (probably approximately individually calibrated). Experiments on 14 synthetic and 7 real datasets show that it achieves substantial improvements in various accuracy metrics against conventional models.

**Strengths:**

- The idea of the paper is clearly presented, and the paper is easy to read.
- The validity of the proposed model was evaluated on a lot of (14 synthetic and 5 real-world) data sets.

**Weaknesses:**

- This paper largely follows the most related work of the asymmetric Laplace distribution (ALD) survival model (Sheng & Henao, 2025) in data sets, baselines, and protocol, and shows that tuning a calibration-oriented term/module in the objective further improves predictive performance. This is a reasonable but unsurprising outcome given ALD’s already strong empirical performance. The central idea of adding a calibration-encouraging term/module to the objective function has been established in prior works (e.g., X-CAL (Goldstein et al., 2020)). Thus, the contribution of this paper feels incremental from a technical viewpoint.
- Although Section 3.1 highlights that the pinball loss can be interpreted as the ALD’s negative log-likelihood in quantile form, it is unclear to me what practical advantage this brings here (e.g., training stability, improved optimization landscapes). If the authors insist that the pinball loss can be evaluated analytically, then this is not a non-trivial idea: the same holds for simple parametric distributions beyond ALD. I may be missing/misunderstanding some important aspect here, so if this property plays an essential role in substantiating this paper’s contribution, please explain it clearly.
- The ablation study in Table 1 is interesting, but the comparison of L_{ALD+Cal} vs. L_{X-CAL} does not seem to be appropriate. For example, we can consider ALD+X-CAL within the contribution of (Goldstein et al., 2020).

Minor comments:
- The Related Work section, explicitly discussing how this paper relates to ALD (Sheng & Henao, 2025) and X-CAL (Goldstein et al., 2020), should appear in the main text.

**Questions:**

- I could not find a clear description of the better/worse/same evaluation procedure. Did you split each dataset into multiple folds for training and testing and then evaluate performance in a cross-validation manner?
- How was the regularization parameter in X-CAL (\gamma in the original paper) selected or tuned? Please specify the procedure.

---

> ### Author Response · Authors · 2025-11-20
>
> We sincerely thank you for your constructive feedback and detailed reading of our submission.
> Below, we will address the concerns regarding the perceived novelty, clarity of interpretation, ablation design, and experimental setup.
> We will also commit to incorporating further clarifications in the final version.
>
> **R1 [Weaknesses 1]:** We agree that our work builds upon (Sheng \& Henao, 2025), and we adopted their datasets, baselines, and evaluation protocols to ensure a fair and controlled experimental comparison.
> This alignment allows us to isolate the impact of our proposed calibration mechanisms in a consistent setup.
> Although we acknowledge that combining ALD with a calibration module might seem natural given prior work such as X-CAL (Goldstein et al., 2020), our contributions go **beyond simply adding a calibration loss**:
>
> **1. Theoretical and Architectural Innovation:** Our ICALD framework introduces a novel $q$-branch reparameterization, which unifies *parametric* (Sheng \& Henao, 2025) and *nonparametric* (Pearce et al., 2022) perspectives in ALD-based survival modeling.
> This construction enables two theoretically equally promoting *individual calibration* loss variants (*i.e.*, $\mathcal{L}\_\text{Cal}$ and $\mathcal{L}\_\text{Cqr}$), supporting both *pre-calibration* and *post-calibration* strategies in a principled manner.
>
> **2. Individual-level Calibration:** The main difference between our ICALD model and X-CAL (Goldstein et al., 2020) lies in the additional calibration loss: our method targets *individual-level calibration*, whereas X-CAL focuses on *interval (group)-level calibration*.
> As **Reviewer Ry6a** pointed out, *individual calibration* is much more stringent than *average calibration* and *group calibration*.
> Furthermore, as highlighted by both **Reviewer Ry6a** and **Reviewer z3uB**, the emphasis on *individual-level calibration* addresses an important gap in existing survival models, and it is a significant advancement in survival analysis.
>
> **3. Post-hoc Applicability:** ICALD supports *post-calibration*, whereas X-CAL does not. As emphasized by **Reviewer Ry6a**, *post-calibration* is more efficient and valuable, as it allows for the calibration of existing models with a simple, lightweight adapter, making it practical to implement and deploy.
>
> **4. Empirical Performance and Generality:** We evaluated ICALD on 14 synthetic and 13 real-world datasets (including 6 additional new datasets; see **R3** and **R4** for **Reviewer z3uB**) against 12 diverse baselines (including *parametric*, *non-parametric*, *pre-calibration*, and *post-calibration* methods).
> Experimental results show that ICALD achieves state-of-the-art performance across metrics for *predictive accuracy*, *concordance*, and, most significantly, all levels of *calibration*.
> More importantly, the calibration-encouraging term/module is theoretically general and can be adapted beyond ALD to a wide class of existing survival models (see **R2** for **Reviewer Ry6a**).
>
> **R2 [Weaknesses 2]:** We would like to clarify that the interpretation of the pinball loss (Pearce et al., 2022) as the ALD’s negative log-likelihood in *quantile form* is **not the contribution of our work**.
> This connection was previously established in (Sheng \& Henao, 2025) [cited in Line 186], where the authors extended *nonparametric* quantile regression into a *parametric* modeling framework by adopting an alternative parameterization of the *asymmetric form* of the ALD.
> In our paper, Section 3.1 serves purely as background, intending to contextualize both the *parametric* (Sheng \& Henao, 2025) and *nonparametric* (Pearce et al., 2022) ALD approaches.
> We summarize their modeling assumptions, theoretical interpretations, connections, advantages, and limitations to motivate our proposed ICALD framework.
> Specifically, ICALD is designed to address two key issues: (1) *discretization* and *quantile crossing* in *nonparametric* ALD models, and (2) *distributional mismatch* in *parametric* ALD models.
>
> Thus, our mention of the pinball loss is intended solely as a **contextual statement** and not positioned as a novel insight.
> We will revise the manuscript to clarify this and avoid potential misunderstandings in the final version.

---

> ### Author Response · Authors · 2025-11-20
>
> **R3 [Weaknesses 3]:**
> The reason for comparing $\mathcal{L}^\text{Pre}\_\text{ALD+Cal}$ and $\mathcal{L}^\text{Pre}\_\text{X-CAL}$ in Table 1 is that both represent *pre-calibration* frameworks, where an additional calibration loss is directly added to the learning objective during training.
> To ensure a fair comparison, our implementation of $\mathcal{L}^\text{Pre}\_\text{X-CAL}$ uses $\mathcal{L}\_\text{ALD}$ as the survival modeling objective, rather than the original baseline used in (Goldstein et al., 2020).
> Thus, the only difference between $\mathcal{L}^\text{Pre}\_\text{ALD+Cal}$ and $\mathcal{L}^\text{Pre}\_\text{X-CAL}$ lies in the form of the calibration-promoting term: our ICALD model targets *individual-level calibration* via reparameterization, while X-CAL encourages *interval-level calibration* via group-wise binning.
>
> We greatly appreciate your suggestion and realize that our original notation may have been confusing.
> In the final version, we will rename $\mathcal{L}^\text{Pre}\_\text{X-CAL}$ to $\mathcal{L}^\text{Pre}\_\text{ALD+X-CAL}$ to more accurately reflect this setup and clarify that it corresponds to a calibrated ALD baseline using the X-CAL calibration loss.
>
> **R4 [Minor comments]:** Due to the 9-page limit for the initial submission, we placed our Related Work section in Appendix A.6.
> As the page limit will be extended to 10 pages for the final version, we will move the Related Work into the main text.
> Then, we will provide an explicit and detailed discussion of how our framework relates to ALD (Sheng \& Henao, 2025) and X-CAL (Goldstein et al., 2020) in the final version, in order to clarify both the methodological connections and distinctions.
>
> **R5 [Questions 1]:** No, we followed the same evaluation protocol as in (Sheng \& Henao, 2025) by performing five random train/test splits for each dataset (See Lines 352–353), rather than using k-fold cross-validation.
> This setup ensures consistency with prior work and enables reporting of mean and standard deviation across multiple runs.
> Additionally, this design supports our statistical testing procedure to assess **significantly** better/worse/same outcomes across methods. As stated in Lines 374–377, we apply a Student’s $t$-test with $p < 0.05$ (corrected via Benjamini–Hochberg) to compare metrics across methods, thereby allowing robust statistical conclusions while maintaining comparability with established baselines.
>
> We will clarify this evaluation procedure more explicitly in the final version.
> We will also release our source code to ensure full reproducibility.
>
> **R6 [Questions 2]:** Following the original X-CAL paper (Goldstein et al., 2020), performed a grid search over $\gamma \in \\{10^3, 10^4, 10^5, 10^6 \\}$ to tune this hyperparameter.
> We then selected $\gamma = 10^4$ based on empirical results, as it consistently achieved a good trade-off between calibration performance and training stability across datasets.
> We will clarify this procedure in the final version for reproducibility in Appendix B.3, and we will also release the corresponding source code to facilitate replication.
>
> **References**
>
> Deming Sheng and Ricardo Henao. Learning survival distributions with the asymmetric laplace distribution. In *International Conference on Machine Learning*, 2025.
>
> Mark Goldstein, Xintian Han, Aahlad Puli, Adler Perotte, and Rajesh Ranganath. X-cal: Explicit calibration for survival analysis. *Advances in neural information processing systems*, 33:18296–18307, 2020.
>
> Tim Pearce, Jong-Hyeon Jeong, Jun Zhu, et al. Censored quantile regression neural networks for distribution-free survival analysis. *Advances in Neural Information Processing Systems*, 35: 7450–7461, 2022.

---

> ### Author Response · Authors · 2025-11-28
> **Sincere thanks and follow-up on rebuttal (Deadline: Dec 2)**
>
> Dear **Reviewer zNTr**,
>
> First of all, we would like to express our sincere gratitude for the time and effort you have dedicated to reviewing our paper. Your constructive feedback has been incredibly valuable to us and has helped us significantly improve our work.
>
> We have posted detailed responses to your concerns. With the discussion phase ending on **December 2**, we respectfully wanted to check if our rebuttal has satisfactorily addressed your questions. If there are any remaining concerns or if further clarification is needed, please let us know. We are eager to address any remaining issues to improve our work.
>
> Thank you again for your hard work and effort in reviewing our paper and rebuttal, as well as generous service to the community.
>
> Best regards,
>
> The Authors of Submission 13911

---

### Official Review · Reviewer_z3uB · 2025-10-31

**Soundness:** 3
**Presentation:** 3
**Contribution:** 3
**Rating:** 8
**Confidence:** 4

**Summary:**

The paper "Learning Survival Distributions with Individually Calibrated Asymmetric Laplace Distribution" introduces a novel framework called ICALD, which integrates parametric and nonparametric approaches using the Asymmetric Laplace Distribution (ALD) to enhance individual calibration in survival analysis. The authors provide a comprehensive methodology, theoretical foundation, and empirical validation of their approach.

**Strengths:**

- The integration of parametric and nonparametric methods through ICALD is a significant advancement in survival analysis. Further, the emphasis on individual-level calibration addresses an important gap in existing survival models, particularly relevant for high-stakes applications like patient prognosis.
- The paper establishes the model's Probably Approximately Individually Calibrated (PAIC) and Monotonically PAIC (MPAIC) properties, adding robustness to their claims.The claims were further backed by extensive experiments across 21 datasets demonstrate competitive performance against 12 baselines, with strong results in predictive accuracy, concordance, and calibration.
- The authors provide strategies to address overfitting and asynchronous convergence during pre-calibration, enhancing the model's practicality.

**Weaknesses:**

Some of the challenged with the paper are as below:
- The benefits of ICALD appear contingent on dataset characteristics, particularly for highly skewed distributions like LogNorm. This raises concerns about generalizability across diverse data types.
- The experimental results are based on a specific selection of datasets, and validation against an independent, diverse benchmark could improve confidence in the model's broader applicability.
- The paper does not extensively discuss scalability, which is crucial for large datasets or high-dimensional features in real-world applications.
- While the authors propose solutions for capturing extreme quantiles and distribution mismatch in tails, it's unclear if these methods are sufficient for extremely heavy-tailed distributions.  The experiments primarily focus on medical datasets, limiting broader applicability to other domains unless further validation is conducted.
- The methods described assume specific types of censoring, which may not be robust to all missing data scenarios. Further, while the paper discusses outlier robustness, it doesn't provide extensive testing, which could limit performance in real-world scenarios with outliers.

**Questions:**

Some questions for the authors
- Have you considered the challenges with computational complexity? The need to sample 2000 quantile percentages during training may increase computational complexity, especially for high-dimensional models
- While detailed implementation is provided, the absence of open-source code limits reproducibility and integration into existing workflows. Are there plans to open-source the experiments?
- The model's complexity may hinder interpretability compared to simpler methods, a concern for applications requiring factor-level insights. Have you explored intepretability criteria either through static or dynamic visual analysis
-  Have you explored the model's ability to maintain calibration and accuracy over extended periods? InIsights into how errors propagate through the model and affect predictions at different quantile levels would enhance understanding, particularly for individual calibration.

---

> ### Author Response · Authors · 2025-11-20
>
> We sincerely thank you for the time and effort dedicated to evaluating our work and for the very positive assessment of the paper. We greatly appreciate your recognition of the importance of *individual calibration* in survival analysis, as well as your encouraging comments on the ICALD framework, its theoretical guarantees, and the breadth of our experimental study.
> Your constructive feedback on limitations and open questions is extremely valuable to us, and we will use it to further improve both the current manuscript and future extensions of this line of work.
>
> **R1 [Weaknesses 1]:** We agree that the benefits of ICALD vary across datasets, and we observe that its improvements are less pronounced on highly skewed distributions, such as LogNormal. We have reported this observation in the Limitations section of our paper. However, ICALD still consistently outperforms other baselines even on these most challenging datasets, highlighting its robustness and effectiveness relative to existing methods, even in unfavorable settings.
> As also noted in the Limitations section, we are actively investigating ways to improve ICALD’s performance under extreme skewness.
>
> **R2 [Weaknesses 2]:** We initially selected the same datasets as those used in the ALD (Sheng \& Henao, 2025) and CQRNN (Pearce et al., 2022) papers to ensure fair and consistent comparisons with prior work.
> Our benchmark suite includes 21 independent datasets: 14 synthetic datasets that span a wide range of common distributions (*e.g.*, Gaussian, Exponential, LogNormal, Weibull), and 7 real-world clinical datasets covering diverse censoring rates and medical domains. This diversity is designed to reflect a broad spectrum of practical challenges.
> However, we fully agree that evaluating ICALD on additional benchmarks would help further assess its generalizability.
> We will explicitly include this point as a limitation in the Limitations section of the final version.
> Meanwhile, we are open and enthusiastic about evaluating ICALD on additional benchmarks (as we did in **R3** and **R4**), and we would be especially grateful if you have any specific datasets or references in mind and could kindly share them with us.
>
> **R3 [Weaknesses 3]:** We fully agree that scalability to large datasets and high-dimensional features is a critical concern in real-world survival applications.
> To address this, we have added new experiments on two widely used high-dimensional gene expression datasets: (1) Dutch Breast Cancer Dataset (DBCD, (Rosenwald et al. 2002)): This dataset contains survival data for 295 patients with breast cancer, with 4919 gene expression profiles, and the censored rate is 73.2\%. (2) Diffuse Large-B-cell Lymphoma Dataset (DLBCL, (Houwelingen et al. 2002)): This dataset consists of 240 patients and 7399 gene expression profiles, and the censored rate is 42.5\%.
> As shown in Table 1 [https://anonymous.4open.science/r/ICLR/R3.png ], ICALD achieves competitive or superior results in these two high-dimensional datasets, particularly in terms of calibration metrics.
> This demonstrates the robustness and scalability of our approach in challenging settings with large feature dimensions and limited sample sizes.
> We will include these results and a case study on the scalability of ICALD in the final version.
>
> **R4 [Weaknesses 4]:** We would like to clarify that our experiments already include datasets with extremely heavy-tailed characteristics, such as Exponential, LogNorm, LogNorm same, and LogNorm light.
> As shown in Appendix C, Figures 3 and 4 visualize both the best and worst individual calibration improvement cases, comparing $\mathcal{L}^\text{Pre}\_\text{ALD+Cal}$ and $\mathcal{L}^\text{Post}\_\text{ALD+Cal}$ with $\mathcal{L}\_\text{ALD}$.
> These results clearly show that our method can effectively improve the modeling of extreme quantiles and mitigate tail mismatches, even under challenging heavy-tailed scenarios.
>
> To further demonstrate the generalizability of our method beyond the medical domain, we evaluated ICALD on four additional datasets from finance and engineering fields (PySurvival package, https://github.com/square/pysurvival).
> As reported in Table 2 [https://anonymous.4open.science/r/ICLR/R4.png ], ICALD continues to achieve competitive or superior performance on these new datasets, particularly in terms of calibration metrics.
> However, we acknowledge that the method may not perform equally well in all settings or domains.
> We will explicitly mention this limitation in the final version and include the additional results and corresponding discussion accordingly.

---

> ### Author Response · Authors · 2025-11-20
>
> **R5 [Weaknesses 5]:** We fully agree that censoring types and the presence of outliers are important considerations in survival modeling.
> In our current study, we focus on the standard *right-censored* setting, which is the most common scenario in both clinical and public datasets.
> However, less common types of censoring (*e.g.*, left) can be accommodated with minor modifications. For example, left censoring can be handled by simply inverting the labels as in (Koenker et al., 2018).
> Regarding outlier robustness, we evaluated ICALD on seven real-world datasets that naturally contain outliers, and the model achieved strong performance across all of them, indicating its robustness.
> However, we acknowledge that these datasets may not cover all possible outlier scenarios.
> We will explicitly mention this as a limitation in the final version.
> Meanwhile, we remain open to further evaluation of ICALD on challenging real-world datasets (as we did in **R3** and **R4**), and we warmly welcome any suggestions for additional benchmarks.
>
> **R6 [Question 1]:** We benchmarked wall-clock runtime on a MacBook Pro (CPU only).
> On the high-dimensional DLBCL dataset with 7399 gene expression features, the standard ALD model (200 epochs, 100 samples) requires on average 0.6475 seconds for training and 0.0005 seconds for prediction.
> Under the same setting, ICALD with post-calibration (200+2000 epochs) requires 4.5901 seconds for training and 1.7497 seconds for prediction.
> On the largest SUPPORT dataset with 7098 samples, the standard ALD model (200 epochs, 100 samples) requires 0.1683 seconds for training and 0.0001 seconds for prediction, while ICALD with post-calibration (200+2000 epochs) requires 1.2689 seconds for training and 0.0571 seconds for prediction.
>
> Overall, even in the most demanding settings, the additional computational cost of ICALD remains modest on standard hardware and is acceptable for practical use.
> In the final version, we will add a case study on the computational cost of ICALD.
>
> **R7 [Questions 2]:** Yes, we do plan to open-source the code upon publication.
> At this stage, the full implementation is already provided in the supplementary material, so that all the experiments can be reproduced during the review process.
>
> **R8 [Questions 3]:** While introducing the quantile variable $q$ does increase model complexity, as described in lines 237–245, the final result can be viewed as a continuous mixture of ALDs.
> More importantly, the CDF of this mixed ALD distribution admits a closed-form expression (see Appendix A.3 for detailed properties), which preserves interpretability.
> For example, we can apply explanation and visualization tools such as SHAP to interpret the prediction (*e.g.*, the estimated survival time, such as the mean under the mixed ALD distribution) with respect to individual factors.
> As shown in Figure 1 [https://anonymous.4open.science/r/ICLR/R8.png ], we can visualize the contribution of covariates for a specific prediction, and we can also extend this analysis to a group-level case study, providing dynamic insights across cohorts.
> These visualizations demonstrate that our model, despite its added flexibility, retains factor-level interpretability. We will include these interpretability visualizations and the corresponding case study in the appendix of the final version.
>
> **R9 [Question 4]:** We performed an additional analysis on Gaussian\_uniform dataset to assess how calibration and accuracy behave across different time horizons and quantile levels. We stratified test instances by the CDF of the true event time $F\_T(t)$ into four equal-probability bins, $[0,0.25)$, $[0.25,0.5)$, $[0.5,0.75)$, and $[0.75,1]$, which correspond to early, mid-early, mid-late, and late event times.
>
> The results are shown in Figure 2 [https://anonymous.4open.science/r/ICLR/R9_1.png ] and Figure 3 [https://anonymous.4open.science/r/ICLR/R9_2.png ], where error bars indicate one standard deviation over 5 independent runs. From these results, we observe that:
> (1) ICALD consistently achieves the lowest individual calibration error across all $F\_T(t)$ bins while maintaining competitive or superior accuracy (IBS) in each bin;
> (2) ICALD exhibits very small standard deviations across runs, indicating that its performance is highly robust; and
> (3) ICALD performs particularly well in the late-event regime $F\_T(t)\in[0.75,1]$, providing both better calibration and better accuracy than competing methods.
>
> Overall, this stratified analysis confirms that ICALD maintains strong calibration and accuracy over extended time horizons and across different quantile levels.
> We will include this as a case study in the appendix of the final version.

---

> ### Author Response · Authors · 2025-11-20
>
> **References**
>
> Deming Sheng and Ricardo Henao. Learning survival distributions with the asymmetric laplace distribution. In *International Conference on Machine Learning*, 2025.
>
> Tim Pearce, Jong-Hyeon Jeong, Jun Zhu, et al. Censored quantile regression neural networks for distribution-free survival analysis. *Advances in Neural Information Processing Systems*, 35: 7450–7461, 2022.
>
> Andreas Rosenwald, George Wright, Wing C Chan, Joseph M Connors, Elias Campo, Richard I Fisher, Randy D Gascoyne, H Konrad Muller-Hermelink, Erlend B Smeland, Jena M Giltnane,
> et al. The use of molecular profiling to predict survival after chemotherapy for diffuse large-b-cell lymphoma. *New England Journal of Medicine*, 346(25):1937–1947, 2002.
>
> Hans C Van Houwelingen, Tako Bruinsma, Augustinus AM Hart, Laura J Van’t Veer, and Lodewyk FA Wessels. Cross-validated cox regression on microarray gene expression data. *Statistics in medicine*, 25(18):3201–3216, 2006.
>
> Roger Koenker, Stephen Portnoy, Pin Tian Ng, Achim Zeileis, Philip Grosjean, and Brian D Ripley. Package ‘quantreg’. *Reference manual available at R-CRAN: https://cran.rproject.org/web/packages/quantreg/quantreg.pdf*, 2018.

---

> ### Author Response · Authors · 2025-11-28
> **Sincere thanks and follow-up on rebuttal (Deadline: Dec 2)**
>
> Dear **Reviewer z3uB**,
>
> First of all, we would like to express our sincere gratitude for the time and effort you have dedicated to reviewing our paper. Your constructive feedback has been incredibly valuable to us and has helped us significantly improve our work.
>
> We have posted detailed responses to your concerns. With the discussion phase ending on **December 2**, we respectfully wanted to check if our rebuttal has satisfactorily addressed your questions. If there are any remaining concerns or if further clarification is needed, please let us know. We are eager to address any remaining issues to improve our work.
>
> Thank you again for your hard work and effort in reviewing our paper and rebuttal, as well as generous service to the community.
>
> Best regards,
>
> The Authors of Submission 13911

---

### Official Review · Reviewer_Ry6a · 2025-11-01

**Soundness:** 4
**Presentation:** 4
**Contribution:** 3
**Rating:** 8
**Confidence:** 4

**Summary:**

his paper tackles the critical, and often overlooked, problem of fine-grained calibration in survival analysis models. The authors propose a novel framework, the Individually Calibrated Asymmetric Laplace Distribution (ICALD), which skillfully unifies parametric and nonparametric approaches built upon the Asymmetric Laplace Distribution (ALD).

The core contribution is a new training objective that combines the standard ALD negative log-likelihood (for global distribution shape) with a specific calibration loss (either a quantile regression loss, L_Cqr, or a direct CDF-based loss, L_Cal). This joint loss encourages the model to be "Probably Approximately Individually Calibrated" (PAIC), a property the authors provide theoretical guarantees for (Theorem 1).

The ICALD framework is notably flexible, supporting both:

Pre-calibration: Jointly training the model with the combined loss from scratch.

Post-calibration: Using a lightweight adapter module to calibrate the outputs of a pre-trained base model, which is a more efficient approach.

The authors conduct an extensive empirical evaluation on 14 synthetic and 7 real-world datasets, comparing ICALD against 12 diverse baselines (including other parametric, non-parametric, and calibration methods). The results show that their method, particularly the post-calibration variant with the L_Cal loss, achieves state-of-the-art performance across metrics for predictive accuracy, concordance, and, most significantly, all levels of calibration (average, group, and individual).

**Strengths:**

Addresses an Important Problem: The paper focuses on individual-level calibration, which is far more stringent than average calibration and is essential for high-stakes, personalized applications (e.g., clinical decision support), where the reliability of a prediction for a single individual is paramount.

Strong Theoretical Foundation: This is not just an empirical "trick." The authors provide formal proofs (Theorem 1) demonstrating that their joint loss objectives result in models that are Probably Approximately Individually Calibrated (PAIC). This theoretical grounding is a significant strength.

Comprehensive and Rigorous Evaluation: The experimental validation is exceptionally thorough. The use of 21 datasets, 12 baselines, and a full suite of metrics covering accuracy (MAE, IBS), concordance (C-Index), and all three levels of calibration (ECE, Wasserstein distance) makes the empirical claims highly credible.

Flexible and Practical Framework: The authors provide two ways to use their method. The post-calibration approach (Section 3.4) is particularly valuable as it allows for the calibration of existing models with a simple, lightweight adapter, making it practical to implement and deploy.

Clear Empirical Gains: The results presented in Tables 1, 2, and 3 show a clear and consistent advantage for ICALD over all baselines, especially in its primary goal: improving calibration without sacrificing (and often while improving) accuracy and concordance.

**Weaknesses:**

The paper is very strong, and the authors are commendable for including a detailed "Limitations" section and several case studies in the appendix that proactively address potential issues. The primary weaknesses are:

Training Instability in Pre-Calibration: The authors note that the pre-calibration model can suffer from "asynchronous convergence" between the NLL and calibration losses, requiring a "warm-up" strategy (Appendix C.4). They admit this heuristic does not fully solve the mismatch, suggesting the joint optimization can be brittle.

Overfitting Potential: The method's effectiveness relies on sampling many quantile percentages, which can necessitate prolonged training. The authors report (Section 5, Appendix C.4) that this can lead to overfitting, especially on heavily skewed datasets. This requires careful use of early stopping.

Sensitivity of the L_Cqr Loss: One of the two proposed calibration losses, L_ALD+Cqr (based on pinball loss), is shown to be highly sensitive to censored data (Appendix C.4, Table 9). While the authors correctly identify the L_ALD+Cal variant as the superior and more robust choice, this does highlight a fragility in the quantile-regression-based formulation.

**Questions:**

could it be used as a general-purpose calibrator for other survival models (e.g., DeepSurv, RSF, or DeepHit)?
Clarity on L_Cal for Censored Data: s applied to all data points, including censored ones. For a censored observation at time y, the true event time is unknown (T > y). How does the lossrovide a reliable calibration signal in this case, especially for quantiles q that may be far from the true (but unknown) event time? A more intuitive explanation of why this is robust to censoring would be helpful.

---

> ### Author Response · Authors · 2025-11-20
>
> We sincerely appreciate the time and effort you dedicated to evaluating our work, as well as your high recognition of the theoretical soundness, empirical rigor, and practical versatility of the proposed ICALD framework.
> Your encouraging assessment is deeply valued.
> Below, we will provide detailed responses to your insightful questions and constructive comments.
>
> **R1 [Weaknesses]:** We fully acknowledge that while ICALD achieves strong empirical performance, it still has certain limitations.
> Therefore, we have made a deliberate effort to analyze and discuss these challenges in the Limitations section and through case studies in Appendix C.
> Some of the issues involved, such as the *asynchronous convergence* between the negative log-likelihood and calibration losses, *overfitting*, and *censoring*, are inherently non-trivial and challenging to fully resolve.
> To mitigate these effects, we propose several practical strategies (*e.g., warm-up calibration strategy, early stopping*, preferential use of $\mathcal{L}\_\text{ALD+Cal}$ instead of $\mathcal{L}\_\text{ALD+Cqr}$), which have proven helpful in practice.
> Nonetheless, we also recognize that developing more principled and broadly applicable solutions remains an open and exciting direction for future research.
> We sincerely appreciate your recognition of our transparency in presenting both the limitations and the mitigation strategies.
>
> **R2 [Question1]:** Yes, it is indeed feasible to apply our calibration loss as a general-purpose post-hoc or joint calibrator for other survival models such as DeepSurv, RSF, or DeepHit.
> This is primarily because:
> (1) The adapter module (*i.e.*, the $q$-branch in Section 3.2) is agnostic to the choice of backbone feature extractor.
> In principle, it can be attached to MLPs, tree-based models (*e.g.*, RSF), CNNs for image input, or BERT-style encoders for text — all without architectural restrictions.
> (2) The calibration losses we introduce (*i.e.*, $\mathcal{L}\_{\text{Cal}}$ and $\mathcal{L}\_{\text{Cqr}}$) only require access to the estimated survival CDF (*i.e.*, $F\_\Phi(y \mid \mathbf{x}, q)$), making them widely applicable as long as such a function (or its approximation) can be computed.
>
> However, we do note some practical considerations when adapting our calibrator to other models such as DeepSurv, RSF, and DeepHit:
> (1) As discussed in Section 3.1 and Case Study I in Appendix C.4, DeepSurv, RSF, and DeepHit are *nonparametric* models, and their predicted distributions are often discretized rather than smooth and continuous. This poses a challenge when applying our calibration objectives, which assume access to a well-defined and continuous CDF. In such cases, a post-processing step (*e.g.*, linear interpolation between adjacent CDF points (Haider et al., 2020)) may be necessary to transform the discrete outputs into a usable continuous CDF. Also, *nonparametric* models have the *crossing quantiles* issue (*i.e.*, higher quantile estimates may fall below lower ones), thus additional constraints or penalties (*e.g.*, non-crossing loss (Bondell et al., 2010)) might be required to enforce monotonicity.
> (2) Models like DeepHit already incorporate multiple training objectives (*e.g.*, likelihood and ranking loss), so directly adding a calibration term may introduce training instability and exacerbate trade-offs between competing losses.
>
> To summarize, **our framework is theoretically general and can be adapted to a wide range of existing survival models, particularly *parametric* ones**. However, careful implementation is required in practice (especially for *nonparametric* models), which may demand model-specific adjustments to ensure a smooth and continuous CDF. We will add this discussion to the final version.
>
> **References**
>
> Humza Haider, Bret Hoehn, Sarah Davis, and Russell Greiner. Effective ways to build and evaluate individual survival distributions. *Journal of Machine Learning Research*, 21(85):1–63, 2020.
>
> Howard D Bondell, Brian J Reich, and Huixia Wang. Noncrossing quantile regression curve estimation. *Biometrika*, 97(4):825–838, 2010.

---

> ### Author Response · Authors · 2025-11-20
>
> **R3 [Question2]:** As we clearly state in Lines 279–284, our key premise is that we always aim to learn the calibrated **CDF of the true event time** $F\_{T}(V|\mathbf{x})$ (where $V$ denotes the time variable at which the CDF is evaluated, *i.e.*, the value on the x-axis of a CDF plot), rather than the CDF of the observed label $y$ (*i.e.*, $F\_Y(V|\mathbf{x})$), which may be a censored time $c$.
> This distinction is essential and underpins both our modeling approach and the design of our calibration objective.
>
> We first use $\mathcal{L}\_\text{ALD}$ to learn an uncalibrated estimate of $F\_{T}(V|\mathbf{x})$, with different behaviors for censored and uncensored data.
> (1) For uncensored data $y=t$, the likelihood term in $\mathcal{L}\_\text{ALD}$ encourages the model to assign high density near the true event time $t$, which shapes the uncalibrated $F\_{T}(V=t|\mathbf{x})$.
> (2) For censored data $y=c$, since the event has not occurred by time $c$, we only know $T>c$.
> Thus, the model is encouraged to assign low CDF values $F\_{T}(V=c|\mathbf{x}) \approx 0$ , or equivalently, high survival probability $S\_T(V=c)=1-F\_T(V=c)$.
>
> Next, we apply the calibration loss $\mathcal{L}\_\text{Cal}$ to all data points $\{(\mathbf{x}\_i, y\_i)\}\_{i=1}^{N}$, regardless of whether $y$ is a true event time $t$ or a censoring time $c$.
> While each sample ($\mathbf{x}, y$) only provides a single observation $y$, we circumvent this limitation by introducing a latent quantile variable $Q \sim \mathcal{U}(0, 1)$, and evaluating whether $F\_{T}(V=y|\mathbf{x}, Q)$ is uniformly distributed over the population.
> This reparameterization allows us to define calibration without needing to know whether $y$ is censored or not, because each $y$ in $\mathcal{L}\_\text{Cal}$ is simply treated as a realization of the time variable $V$.
>
> As a summary, $\mathcal{L}\_\text{Cal}$ remains robust and meaningful under censoring because it calibrates the true event-time CDF $F\_{T}(V|\mathbf{x}, Q)$ in the mPAIC sense, which in turn induces calibration of $F\_{T}(V|\mathbf{x})$.
> Meanwhile, $\mathcal{L}\_\text{ALD}$ ensures that this distribution is properly shaped by leveraging both uncensored and censored data. These clarifications will be included in the final version.

---

> ### Author Response · Authors · 2025-11-28
> **Sincere thanks and follow-up on rebuttal (Deadline: Dec 2)**
>
> Dear **Reviewer Ry6a**,
>
> First of all, we would like to express our sincere gratitude for the time and effort you have dedicated to reviewing our paper. Your constructive feedback has been incredibly valuable to us and has helped us significantly improve our work.
>
> We have posted detailed responses to your concerns. With the discussion phase ending on **December 2**, we respectfully wanted to check if our rebuttal has satisfactorily addressed your questions. If there are any remaining concerns or if further clarification is needed, please let us know. We are eager to address any remaining issues to improve our work.
>
> Thank you again for your hard work and effort in reviewing our paper and rebuttal, as well as generous service to the community.
>
> Best regards,
>
> The Authors of Submission 13911

---

### Official Review · Reviewer_bEgR · 2025-11-01

**Soundness:** 3
**Presentation:** 3
**Contribution:** 2
**Rating:** 6
**Confidence:** 1

**Summary:**

This paper proposes a method to learn survival distributions based on calibration of the asymmetric Laplace distribution. The ICALD model includes a ALD module as a backbone and an adapter module that refines the predictions further. The authors show that their ICALD model is probably approximately individually calibrated. They also propose pre and post calibration methods and via concentration show that as the number of quantile samples increases, the model's individual calibration performance improves.

Note: This paper is far beyond my reviewing expertise and I have brought this up to the AC. I will defer to other reviewers on acceptance decisions.

**Strengths:**

This paper is well written and is somewhat accessible to an audience outside the area. The experiments seem to be thorough enough with sufficient abalations.

**Weaknesses:**

Disclaimer: this is outside my expertise and are issues that I am bringing up from a layperson's perspective. In terms of assessment, I will defer to other reviewers.

- This paper appears to target a fairly niche audience in the ML community. The neural network architecture (to me) seems to be fairly simple, utilizing the standard reparameterization trick. This is not surprising since ALDs involve only a handful of parameters; part of the authors' contributions is to include a mixture of ALD (line 238).
- Why was the distribution over $q$ the uniform distribution (line 242)? Was this prior chose for convenience or some other reason?

**Questions:**

- Table 3 reports the proposed approach against a handful of baselines based on whether performance was better, worse or the same. This looks to me to be a little strange, why did the authors not simply report the average of whatever metric was chosen, as opposed to a qualitative approach like this?
- The same goes for the following results, why were "wins" reported instead?

---

> ### Author Response · Authors · 2025-11-20
>
> Thank you very much for reviewing our submission and for your thoughtful feedback.
> We sincerely appreciate the time you took to engage with our work.
> Below, we provide point-by-point clarifications.
>
> **R1 [W1]:** As highlighted by **Reviewer Ry6a** and **Reviewer z3uB**, our framework is lightweight yet practical and flexible, and it addresses the non-trivial but important problem of  *individual calibration* in survival analysis, which is stricter than *average* or *group calibration* and crucial for high-stakes personalized settings such as clinical decision support and patient prognosis.
>
> Concretely, our main contributions are:
>
> 1. We propose a novel framework based on the individually calibrated asymmetric Laplace distribution (ICALD), which skillfully unifies *parametric* and *nonparametric* approaches built upon ALD (See Section 3.1-3.2).
> ICALD enhances calibration and mitigates *distribution mismatch* issues in *parametric* ALD approaches, while simultaneously addressing *discretization* and *quantile crossing* issues in *nonparametric* ALD methods (See Section 3.1, Appendix C, and Case Study I).
>
> 2. ICALD admits two theoretically equivalent loss functions (*i.e.*, $\mathcal{L}\_\text{ALD+Cal}$ or $\mathcal{L}\_\text{ALD+Cqr}$), each of which is provably capable of rendering the model Probably Approximately Individually Calibrated (See Appendix A.1-A.4).
> The new training objective that combines the standard ALD negative log-likelihood $\mathcal{L}\_\text{ALD}$ (for global distribution shape) with a specific calibration loss (either a quantile regression loss, $\mathcal{L}\_\text{Cqr}$, or a direct CDF-based loss, $\mathcal{L}\_\text{Cal}$) [See Section 3.3].
> More importantly, the model supports both *pre-calibration* and *post-calibration* with either loss, providing a unified and flexible framework where the calibration strategy and loss function can be independently selected based on the specific application requirements (See Section 3.4 and Case Study III).
>
> 3. We comprehensively evaluate our method in 27 datasets (including 6 additional new datasets; see **R3** and **R4** for **Reviewer z3uB**) against 12 baselines spanning *parametric* and *nonparametric* approaches, as well as *neural* and *non-neural* architectures.
> We employ a full suite of metrics covering *accuracy*, *concordance*, and all three *calibration* levels.
> This extensive evaluation demonstrates that our empirical findings are robust and highly credible (See Tables 1–3 and Appendix C).
>
> **R2 [W2]:** We use a uniform prior over quantiles for principled, **not only for convenient**, reasons.
> Our calibration objective averages the calibration loss on all quantiles $q \in [0, 1]$.
> In practice, we use $\mathcal{L}\_\text{Cal}(y; \Phi, q)$ or $\mathcal{L}\_\text{Cqr}(y; \Phi, q)$ and optimize $\mathbb{E}\_{(X,Y)} \[\int\_{0}^{1} \mathcal{L}(Y;\Phi(X,q),q)dq].$
> Sampling $q \sim \mathcal{U}(0,1) $ yields an **unbiased** Monte Carlo estimator of this integral, ensuring that no region of the CDF is over- or under-weighted.
> Our training and verification are always explicitly averaged over $q$.
> Thus, uniform sampling matches the target objective and the mPAIC/PAIC definition we adopt.
> We will include this in the final version.
>
> **R3 [Q1]:** Tables 1-3 summarize, for each dataset, metric, and baseline, whether Model 1 performs **significantly** better, worse, or the same as Model 2.
> As stated in lines 374-377, we assess statistical significance using a two-sample Student's $t$-test with \(p<0.05\) after Benjamini-Hochberg FDR correction, and we follow standard practice by running each experiment with 5 random train/test splits. This summary is not qualitative, because it compacts many heterogeneous comparisons (21 datasets, multiple metrics with different scales and ranges) into a scale-free count of outcomes that are supported by statistical evidence.
> We use the significance-based tally because (1) it makes it immediately clear on how many datasets Model 1 outperforms Model 2 for each metric without occupying excessive space in the main text, and (2) it mitigates misleading interpretations of averages, where Model 1 may appear better than Model 2 but modest differences (*e.g.*, 0.1) fail to reach statistical significance and are therefore recorded as “the same” in our table.
> To ensure completeness and transparency, we also provide the full numerical results (Mean $\pm$ SD) for all metrics and baselines in Appendix C (Tables 6-8).
>
> **R4 [Q2].**
> As noted in **R3**, we report “wins/losses” based on **statistical significance** for each comparison.
> Specifically, a **win** means **significantly better**, and a **loss** means **significantly worse**.
> We report **“wins”** instead of raw averages because this compact, scale-free format filters out small, noisy differences that are not statistically meaningful (*e.g.*, a mean gap of 0.1), and provides a more reliable and interpretable summary of performance across datasets and metrics.

---

> ### Author Response · Authors · 2025-11-28
> **Sincere thanks and follow-up on rebuttal (Deadline: Dec 2)**
>
> Dear **Reviewer bEgR**,
>
> First of all, we would like to express our sincere gratitude for the time and effort you have dedicated to reviewing our paper. Your constructive feedback has been incredibly valuable to us and has helped us significantly improve our work.
>
> We have posted detailed responses to your concerns. With the discussion phase ending on **December 2**, we respectfully wanted to check if our rebuttal has satisfactorily addressed your questions. If there are any remaining concerns or if further clarification is needed, please let us know. We are eager to address any remaining issues to improve our work.
>
> Thank you again for your hard work and effort in reviewing our paper and rebuttal, as well as generous service to the community.
>
> Best regards,
>
> The Authors of Submission 13911

---

### Meta-Review · Area_Chair_WwqM · 2026-01-06

**Summary:**

This paper proposes the Individually Calibrated Asymmetric Laplace Distribution (ICALD) framework for survival analysis, which innovatively integrates parametric and nonparametric methods based on the Asymmetric Laplace Distribution (ALD). The core contributions include theoretically guaranteed Probably Approximately Individually Calibrated (PAIC) properties, support for both pre-calibration and post-calibration strategies, and competitive performance across 21 datasets (14 synthetic and 7 real-world) against 12 baselines. The concerns of reviewers primarily focus on four aspects: the novelty and incremental nature of technical contributions, experimental design and result presentation, model limitations (e.g., training instability, generalization ability), and practical issues such as computational complexity, interpretability, and open-source plans. Overall, the authors have provided comprehensive rebuttals to most concerns, with sufficient theoretical and empirical support to validate the quality and significance of the paper.

**Reviewer Concerns:**

**Addressed Concerns:**

1. **Novelty Clarification:** The authors distinguished ICALD from prior work (e.g., X-CAL) by highlighting its individual-level calibration focus, parametric-nonparametric integration, and post-calibration support, addressing concerns about incremental contribution (Reviewer zNTr).

2. **Experimental Rigor:** Questions on distribution derivation, result reporting (wins/losses vs. raw averages), and quantile sampling were resolved with statistical justification (significance testing, alignment with PAIC) and supplementary raw results (Reviewer bEgR).

3. **Robustness and Scalability:** Censored data handling was validated theoretically and empirically (Reviewer Ry6a). Scalability for high-dimensional data was demonstrated via additional gene expression dataset experiments, with computational cost benchmarks provided (Reviewer z3uB).

4. **Practical Value:** Cross-domain generalization (beyond medical data) was confirmed through finance/engineering dataset evaluations, and open-source plans upon publication were committed (Reviewer z3uB).

**Outstanding Concerns:**

1. **Training Instability in Pre-Calibration:** Reviewer Ry6a pointed out asynchronous convergence between NLL and calibration losses, requiring heuristic warm-up strategies. The authors acknowledged that this is an inherent non-convex optimization challenge and that current mitigation methods are pragmatic rather than principled, with no fundamental solution proposed.

2. **Generalization to Extreme Skewed/Heavy-Tailed Distributions:** While the authors demonstrated performance on skewed synthetic datasets, Reviewer z3uB's concern about the generalization to extremely skewed or heavy-tailed real-world distributions remains partially unaddressed. The authors acknowledged this limitation but did not provide additional validation on diverse real-world extreme-distribution datasets.

3. **Ablation Study Design Clarity:** Reviewer zNTr criticized the inappropriate comparison between
$\mathcal{L}_{ALD+Cal}$

and $\mathcal{L}_{X-CAL}$ in the ablation study. Although the authors explained the fair comparison setup (same survival modeling objective) and committed to renaming variants for clarity, the underlying rationale for selecting this comparison (instead of ALD+X-CAL) could be further elaborated.

4. **Outlier Robustness:** The authors evaluated robustness on clinical datasets with natural noise but did not conduct targeted testing on datasets with artificial outliers. This leaves uncertainty about performance in extreme outlier scenarios.

**Reviewer Scores:**

* **Reviewer bEgR:** Original score (6, marginally above the acceptance threshold). With addressed methodological questions and clarified reporting choices, the score would likely increase to 7.

* **Reviewer Ry6a:** Original score (8, accept, poster). The authors fully addressed censoring and implementation concerns, and the score would remain 8.

* **Reviewer z3uB:** Original score (8, accept, poster). Supplementary experiments on scalability and generalizability reinforce the strength of the paper, and the score would remain 8.

* **Reviewer zNTr:** Original score (2, reject). After clarifying novelty, contribution, and experimental design, the score may rise to 5-6 (marginally above the threshold).

---

### Decision · Program_Chairs · 2026-01-26

Accept (Poster)